# Annelid functional genomics reveal the origins of bilaterian life cycles

Francisco M. Martín-Zamora[1,8], Yan Liang[1,8], Kero Guynes[1], Allan M. Carrillo-Baltodano[1], Billie E. Davies[1], Rory D. Donnellan[1], Yongkai Tan[2], Giacomo Moggioli[1], Océane Seudre[1], Martin Tran[1,3], Kate Mortimer[4], Nicholas M. Luscombe[2], Andreas Hejnol[5,6], Ferdinand Marlétaz[7✉] & José M. Martín-Durán[1✉]

Indirect development with an intermediate larva exists in all major animal lineages[1], which makes larvae central to most scenarios of animal evolution[2–11]. Yet how larvae evolved remains disputed. Here we show that temporal shifts (that is, heterochronies) in trunk formation underpin the diversification of larvae and bilaterian life cycles. We performed chromosome-scale genome sequencing in the annelid *Owenia fusiformis* with transcriptomic and epigenomic profiling during the life cycles of this and two other annelids. We found that trunk development is deferred to pre-metamorphic stages in the feeding larva of *O. fusiformis* but starts after gastrulation in the non-feeding larva with gradual metamorphosis of *Capitella teleta* and the direct developing embryo of *Dimorphilus gyrociliatus*. Accordingly, the embryos of *O. fusiformis* develop first into an enlarged anterior domain that forms larval tissues and the adult head[12]. Notably, this also occurs in the so-called 'head larvae' of other bilaterians[13–17], with which the *O. fusiformis* larva shows extensive transcriptomic similarities. Together, our findings suggest that the temporal decoupling of head and trunk formation, as maximally observed in head larvae, facilitated larval evolution in Bilateria. This diverges from prevailing scenarios that propose either co-option[9,10] or innovation[11] of gene regulatory programmes to explain larva and adult origins.

Many animal embryos develop into a larva that metamorphoses into a sexually competent adult[1]. Larvae are morphologically and ecologically diverse, and given their broad phylogenetic distribution, they are central to major scenarios of animal evolution[2–11]. However, these scenarios dissent on whether larvae are ancestral[2–6] or secondarily evolved[9,10], and on the mechanisms that facilitated the evolution of larvae[2,9–11]. Therefore, larval origins—and their importance to explain animal evolution—are still contentious.

The trochophore is a widespread larval type characterized by an apical sensory organ and a pre-oral locomotive ciliary band[18] that is typically assigned to Annelida and Mollusca. Annelids, however, show diverse life cycles and larval morphologies, including species with direct and indirect development and either planktotrophic or lecithotrophic larvae[19]. Notably, the groups Oweniidae and Magelonidae—which form Oweniida, the sister taxon to all other annelids[20]—have distinctive planktotrophic larvae (Fig. 1a and Extended Data Fig. 1a). In particular, the larva of Oweniidae, referred to as 'mitraria'[12], has an enlarged pre-oral region and a bundle of posterior chaetae, as well as a pair of nephridia and a long monociliated ciliary band similar to those of phylogenetically distant larvae of echinoderms and hemichordates[21,22]. Yet oweniids show many developmental characteristics that are considered ancestral to Annelida and even Spiralia as a whole[23,24], including

similarities in larval molecular patterns with other trochophore and bilaterian larvae[22,23,25,26]. Therefore, the diversity of life cycles and larval forms but generally conserved early embryogenesis and adult body plans of Annelida is an excellent model to investigate how larval traits evolve. It is also an ideal model to formulate and assess hypotheses on the origin of larvae and animal life cycles.

## *O. fusiformis* has a conserved genome

To investigate how larvae evolved in Annelida, we first generated a chromosome-scale reference assembly for the oweniid *O. fusiformis* (Fig. 1b, inset). The haploid assembly spans 505.8 Mb and has 12 chromosome-scale scaffolds (Supplementary Fig. 1). Almost half of the assembly (43.02%) consists of repeats (Extended Data Fig. 1b,c), and we annotated 26,966 protein-coding genes and 31,903 transcripts, which represent a nearly complete (97.5%) set of metazoan BUSCO genes (Supplementary Fig. 1). Gene family reconstruction and gene content analysis nested *O. fusiformis* with other non-annelid spiralians and taxa with slow-evolving genomes (Fig. 1b and Extended Data Fig. 1d,e). This result provides evidence that *O. fusiformis* has fewer gene family gains and losses and retains more ancestral metazoan orthogroups than other annelid taxa (Fig. 1c and Extended Data Fig. 1f,g). Indeed,

[1]School of Biological and Behavioural Sciences, Queen Mary University of London, London, UK. [2]Genomics and Regulatory Systems Unit, Okinawa Institute of Science and Technology Graduate University, Okinawa, Japan. [3]Department of Infectious Disease, Imperial College London, London, UK. [4]Department of Natural Sciences, Amgueddfa Cymru–Museum Wales, Cardiff, UK. [5]Department of Biological Sciences, University of Bergen, Bergen, Norway. [6]Institute of Zoology and Evolutionary Research, Faculty of Biological Sciences, Friedrich Schiller University Jena, Jena, Germany. [7]Department of Genetics, Evolution and Environment, University College London, London, UK. [8]These authors contributed equally: Francisco M. Martín-Zamora, Yan Liang. ✉e-mail: f.marletaz@ucl.ac.uk; chema.martin@qmul.ac.uk

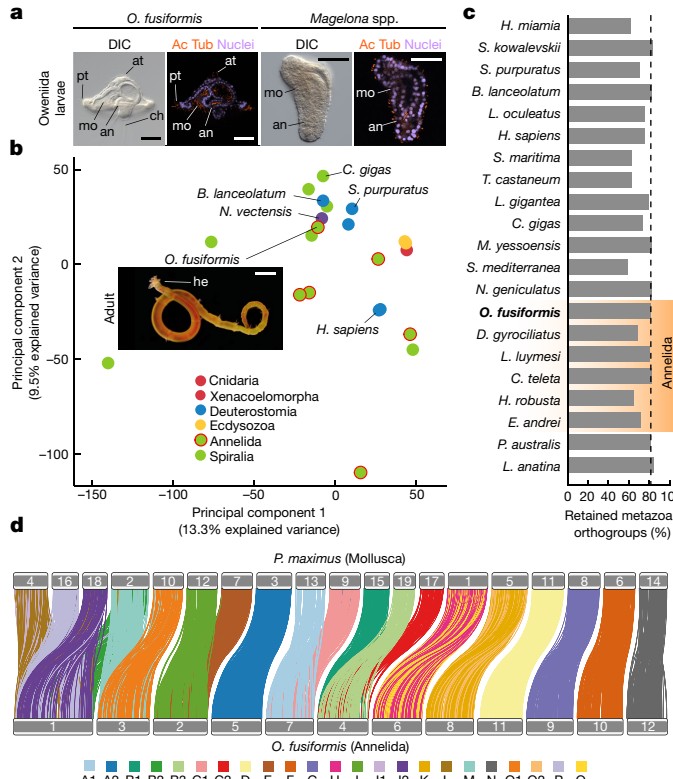

**Fig. 1 | *O. fusiformis* has a distinct larva and a conservatively evolving genome. a**, The larvae of oweniids and magelonids are unlike other annelid larvae. Differential interface contrast (DIC) images and *z*-stack confocal laser scanning views of a *O. fusiformis* mitraria and a *Magelona* spp. larva stained for nuclei using DAPI and acetylated α-tubulin (Ac Tub). **b**, Principal component analysis of metazoan gene complements demonstrates that *O. fusiformis* clusters with other lineages with conservatively evolving gene complements. See Extended Data Fig. 1e for a fully labelled graph. Inset, image of an adult *O. fusiformis*. **c**, Percentage of retained pre-metazoan and metazoan orthogroups per species. Dotted vertical line represents the value for *O. fusiformis*. A list of species names are provided in Supplementary Table 2. **d**, Karyotypic correspondence between *O. fusiformis* and *Pecten maximus*, which exemplifies the ancestral spiralian chromosome complement. Each colour represents an ancestral bilaterian linkage group. Schematic drawings are not to scale. an, anus; at, apical tuft; ch, chaetae; he, head; mo, mouth; pt, prototroch; tt: telotroch. Scale bars, 50 μm (**a**) or 2.5 mm (**b**).

*O. fusiformis* has a *chordin* orthologue, a bone morphogenetic protein inhibitor involved in dorsoventral patterning thought to be lost in annelids[27] and is asymmetrically expressed around the blastopore of the gastrula and larval mouth in *O. fusiformis* (Extended Data Fig. 2). Moreover, *O. fusiformis* has globally retained the ancestral bilaterian linkage, exhibiting chromosomal fusions that are present in molluscs and even nemerteans, and fewer lineage-specific chromosomal rearrangements than other annelids (Fig. 1d and Extended Data Fig. 1h,i). Therefore, *O. fusiformis* shows a more complete gene repertoire and ancestral syntenic chromosomal organization than other annelids. Together with its phylogenetic position and conserved early embryogenesis[23,24], *O. fusiformis* is a key lineage to reconstruct the evolution of Annelida, and of Spiralia generally.

## Heterochronies in gene expression

Next, we sought to identify transcriptomic changes that underpin the distinct life cycles in Annelida. We compared temporal series of embryonic, larval and competent and juvenile transcriptomes of *O. fusiformis* and *C. teleta*, two indirect developers with planktotrophic and lecithotrophic[28] larvae, respectively, and *D. gyrociliatus*, a direct developer[29,30]

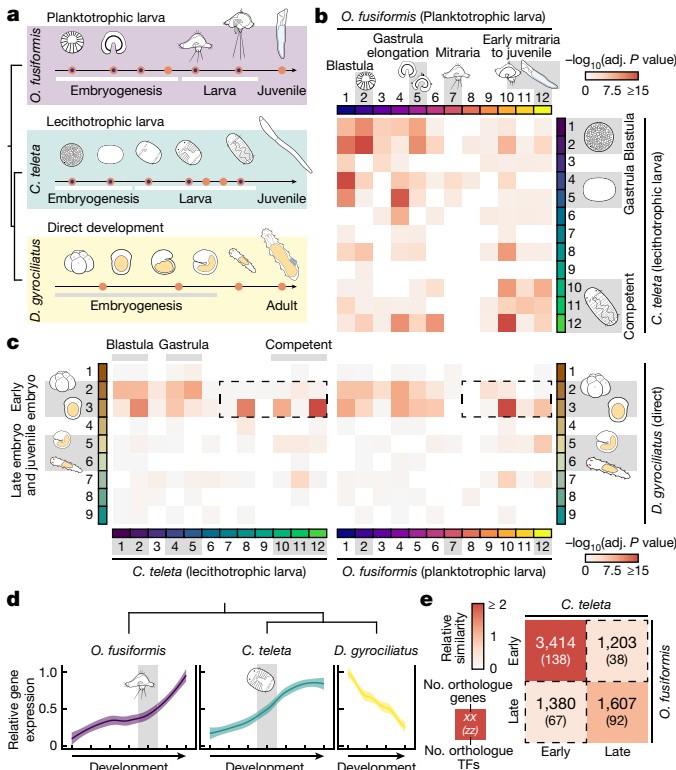

**Fig. 2 | Heterochronies in gene regulatory programmes underpin annelid life cycle diversification. a**, Experimental design of the comparative developmental RNA sequencing (RNA-seq) and assay for transposase-accessible chromatin with sequencing (ATAC-seq) time courses. Orange circles highlight stages of *O. fusiformis*, *C. teleta* and *D. gyrociliatus* development sampled for bulk RNA-seq. Orange circles with a blue inner dot highlight developmental stages sampled for ATAC-seq. **b**,**c**, Similarity heatmaps showcasing the orthogroup overlap between the transcription factors contained in clusters of co-regulated genes obtained by soft *k*-means clustering between all three studied annelid taxa: 12 clusters for *O. fusiformis* and *C. teleta*, and 9 clusters for *D. gyrociliatus*. Time points associated to key clusters are shown for all three species. Dotted black lines in **c** highlight the distinct timing expression differences of a significant number of transcription factors shifted from post-larval expression in indirect developers to early embryogenesis in *D. gyrociliatus*. *P* values were derived from upper-tail hypergeometric tests and Benjamini–Hochberg-adjusted (adj. *P* value). **d**, Average expression dynamics of the 28 single-copy orthologue transcription factors shifted from late expression in both *O. fusiformis* and *C. teleta* to early expression in *D. gyrociliatus*. Curves are locally estimated scatterplot smoothings, coloured shaded areas represent standard error of the mean. **e**, Heatmap of relative similarity based on whole-genome orthogroup overlap analysis by quadrants between pre-larval (early) and post-larval (late) clusters in *O. fusiformis* and *C. teleta*. Dotted black lines denote the groups of genes and transcription factors under heterochronies between both species.

(Fig. 2a). Transcriptional dynamics during early embryogenesis were overall similar among these species (Supplementary Fig. 3). *C. teleta* and *D. gyrociliatus* showed increasing transcriptomic divergence with each other as they develop into adult stages; however, the maximal transcriptomic divergence between these annelids and *O. fusiformis* occurred at the mitraria stage (Extended Data Fig. 3a,b). Soft clustering of all expressed transcripts produced 12 distinct groups of temporally co-regulated genes in *O. fusiformis* and *C. teleta*, and 9 clusters in *D. gyrociliatus* (Extended Data Fig. 3c–e), which were expressed gradually along the life cycle of all three species. Only one cluster in each species showed a bimodal activation at early embryogenesis and in the competent larva (juvenile or adult forms), consistently involving genes enriched for core cellular processes (Extended Data Fig. 3f).

Indeed, translation and metabolism predominated in clusters of early development in the three annelids, whereas cell communication and signalling, morphogenesis and organogenesis were enriched in later stages of development (Extended Data Fig. 3f). Therefore, regardless of the life cycle, transcriptional dynamics are generally conserved during annelid development, yet adults and the planktotrophic larva are the most transcriptionally distinct stages.

To identify the genes that mediate the transcriptional differences at larval and adult stages, we performed pairwise inter-species comparisons of gene and transcription factor composition among clusters of temporally co-regulated genes (Fig. 2b,c and Extended Data Fig. 4a,b). Early clusters followed by late clusters were the most conserved in the three comparisons when all genes were considered (Extended Data Fig. 4c,d). However, transcription factors used in post-larval stages in indirect development were consistently shifted towards early embryogenesis in direct development (Fig. 2c and Extended Data Fig. 4c,e). In both *O. fusiformis* and *C. teleta*, this shift involved 28 transcription factors that function in various developmental processes, from nervous system (for example, *pax6* (ref. [31])) and mesoderm (for example, *foxF* (ref. [26])) formation to axial patterning (for example, *Hox1* and *Hox4* (ref. [32])) (Supplementary Fig. 12). Notably, the overall expression of these 28 genes was also temporally shifted between indirect developing annelids, with the maximum level of expression occurring earlier in *C. teleta* than in *O. fusiformis* (Fig. 2d). Additionally, 2,583 genes also exhibited temporal shifts between the larvae of *O. fusiformis* and *C. teleta* (Fig. 2e), including 105 transcription factors, but mostly enzymes and structural genes that probably reflect the different biology of these two larvae (Extended Data Fig. 4f,g and Supplementary Figs. 13–16). Therefore, temporal shifts (that is, heterochronies) in the use of shared genetic programmes and regulatory genes correlate with and might account for life cycle and larval differences in Annelida.

## Different timings of trunk development

Homeodomain transcription factors were the largest class among the 28 transcription factors with temporal expression shifts between direct and indirect developing annelids (Supplementary Fig. 12). Indeed, homeodomain genes were enriched in the competent larva in *O. fusiformis* but were prevalent from stage 5 larva onwards in *C. teleta* (Extended Data Fig. 4h). Accordingly, Hox genes, which regionalize the bilaterian trunk along the anteroposterior axis[33], were strongly upregulated in the competent mitraria larva (Extended Data Fig. 5a,b). *O. fusiformis* had a conserved complement of 11 Hox genes—similar to *C. teleta*[32]—arranged as a compact, ordered cluster in chromosome 1, except for *Post1*, which was located downstream of this chromosome (Extended Data Fig. 5c,d). *C. teleta* and *D. gyrociliatus* started expressing Hox genes along their trunks[30,32] during or soon after gastrulation (Extended Data Fig. 5e). *O. fusiformis*, however, did not express Hox genes during embryogenesis but in the trunk rudiment during larval growth, already in an anteroposterior staggered pattern, as later observed in the juvenile (Fig. 3a and Extended Data Fig. 5e–h). This late activation of Hox genes is not specific to *O. fusiformis*, as it also occurs for most Hox genes in the planktotrophic trochophore of the echiuran annelid *Urechis unicinctus*[34] (Extended Data Fig. 5e). Therefore, the spatially collinear Hox code along the trunk is established at distinct developmental stages depending on the life cycle mode in Annelida.

To determine whether the difference in timings of trunk patterning is limited to the expression of Hox genes, we used tissue-specific adult transcriptomes to define a set of 1,655 anterior and 407 posterior and trunk genes in *O. fusiformis* (Extended Data Fig. 6a–d). Anterior genes were significantly more expressed during embryogenesis, whereas posterior and trunk genes were upregulated at the mitraria stage and significantly outweighed the expression dynamics of anterior genes from that stage onwards (Fig. 3b and Extended Data Fig. 6e,f). Moreover, anterior, trunk and posterior genes with spatially resolved expression

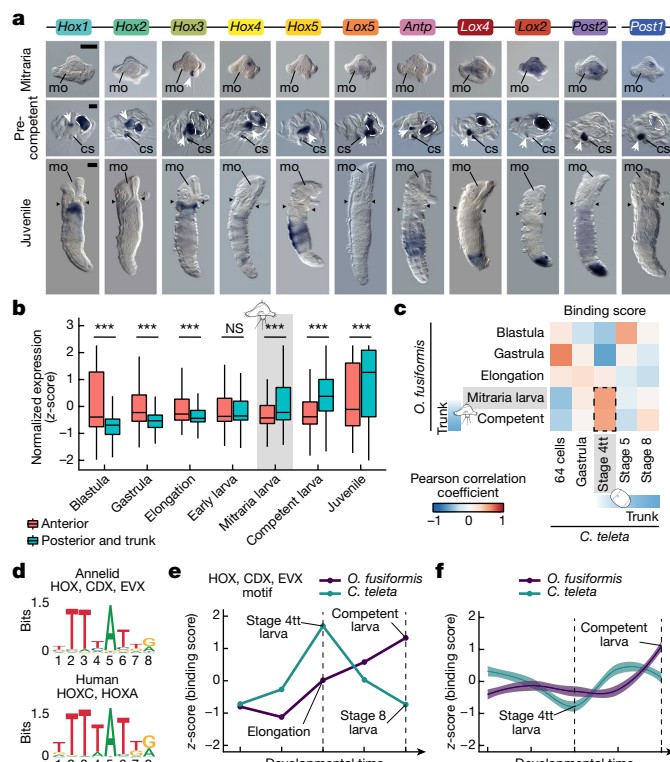

**Fig. 3 | Trunk development is delayed to pre-metamorphosis in *O. fusiformis*.** **a**, Representative images from three independent analyses of Hox gene expression by whole-mount in situ hybridization at the mitraria larva, pre-competent larva, and juvenile stages of *O. fusiformis*. Only *Hox3* is expressed at the mitraria stage (white arrowhead). Hox genes show spatial collinearity along the anteroposterior axis at the developing trunk of the pre-competent larva (white arrowheads) and in the juvenile. Dotted lines in the competent larva panels indicate background from the midgut. Black arrowheads in the juvenile panels indicate head to trunk boundary. cs, chaetal sack; mg, mid gut. Scale bar, 50 μm (larval stages) or 100 μm (juvenile stage). **b**, Average expression dynamics of anterior ($n = 1,655$), and posterior and trunk genes ($n = 407$) expressed in corresponding adult tissues during *O. fusiformis* development. *P* values were derived from two-tailed Student's *t*-tests and adjusted using the Bonferroni method for multiple testing correction. ***, $P < 0.001$; NS, not significant. Centre lines in boxplots are the median, box is the interquartile range (IQR), and whiskers are the first or third quartile ± 1.5× IQR. **c**, Correlation matrices of transcription factor binding score (TFBSs). The dotted black line highlights the high TFBS correlation and heterochrony between the mitraria and competent larvae of *O. fusiformis* and the stage 4tt larva of *C. teleta*. **d**, Sequence logo of the annelid archetype (top) shows substantial similarity to the human homologue (bottom). **e**, TFBS dynamics for the annelid HOX, CDX, and EVX motif during *O. fusiformis* and *C. teleta* development. **f**, Average TFBS dynamics of all motifs in the peaks of the Hox cluster. Curves are locally estimated scatterplot smoothings, coloured shaded areas represent standard error of the mean.

followed different temporal dynamics in *O. fusiformis*, *C. teleta* and *D. gyrociliatus*. In *O. fusiformis*, trunk[25] and posterior[24,26] genes were concentrated in a small ventral area and around the anal opening of the larva and increased in spatial range and expression levels as the trunk formed (Extended Data Fig. 6g,h). By contrast, anterior genes[26,35] patterned most of the mitraria, and their expression remained stable during development (Extended Data Fig. 6g,h). Posterior and anterior genes followed similar dynamics in *C. teleta*, and trunk genes were upregulated already post-gastrula in both *C. teleta* and *D. gyrociliatus* (Extended Data Fig. 6i–l). Therefore, trunk development, which initially occurs from lateral growth of the trunk rudiment[12,28], is deferred to pre-metamorphic stages in planktotrophic annelid trochophores compared with annelids with lecithotrophic larvae and direct developers.

## Heterochronies in Hox regulation

To investigate the genomic regulatory basis for the heterochronies in trunk development among annelid larvae, we profiled open chromatin regions at five equivalent developmental stages in *O. fusiformis* and *C. teleta* (Fig. 2a). This analysis identified 63,726 and 44,368 consensus regulatory regions, respectively. In both species, open chromatin was more abundant within gene bodies (Extended Data Fig. 7a). There was, however, a general increase in promoter peaks in *O. fusiformis* and distant intergenic regulatory elements in both species during development (Extended Data Fig. 7b). Moreover, the largest changes in peak accessibility occurred in the mitraria in *O. fusiformis* and stage 5 larva in *C. teleta* (Supplementary Fig. 18). In *O. fusiformis*, most regulatory regions acted before the start of trunk formation, whereas the numbers of accessible regions with a maximum of accessibility before and after the onset of trunk development were comparable in *C. teleta* (Extended Data Fig. 7c). Accordingly, the regulation of genes involved in morphogenesis and organogenesis, as well as neurogenesis, was concentrated in late clusters in *O. fusiformis* but unfolded more continuously in *C. teleta* (Supplementary Fig. 23). Therefore, different dynamics of chromatin accessibility occur during development and larva formation in these two annelids.

To investigate the regulatory programmes controlling larva development in *O. fusiformis* and *C. teleta*, we predicted transcription factor-binding motifs on peaks obtained from ATAC-seq data. This analysis identified 33 motifs common to both species that were strongly assigned to a known transcription factor class (Supplementary Fig. 29). Notably, the binding dynamics of these 33 motifs revealed a temporal shift in regulatory motifs acting between the mitraria and competent larva in *O. fusiformis* to the early post-gastrula (stage 4tt) larva of *C. teleta* (Fig. 3c and Extended Data Fig. 7d–f). Seven motifs followed this pattern (Extended Data Fig. 7g and Supplementary Fig. 29), including one with high similarity to the human HOX, CDX and EVX motif archetype (Fig. 3d,e) that is overrepresented and upregulated on the basis of its binding score at the competent stage in *O. fusiformis* (Extended Data Fig. 7h and Supplementary Fig. 30). Indeed, motif-binding dynamics in regulatory elements assigned to Hox genes supported a change in global regulation of the Hox cluster at the competent and early larval stages in *O. fusiformis* and *C. teleta*, respectively (Fig. 3f and Supplementary Fig. 31), which mirrored the transcriptional onset of these genes and the start of trunk development in the two species[32]. Motifs assigned to NKX and GATA factors, which are expressed in the developing trunk in both species[25,36], were among the most abundant bound motifs in the Hox cluster in both species (Extended Data Fig. 7i). However, only 39 one-to-one orthologues with bound HOX, CDX and EVX motifs at the maximum of motif binding were common to *O. fusiformis* and *C. teleta* (Extended Data Fig. 7j). Therefore, different regulatory dynamics of the Hox cluster—possibly triggered by a reduced common set of upstream regulators—underpin temporal variability in Hox activity and downstream targets. These shifts probably promoted the developmental and morphological differences in trunk formation between planktotrophic and lecithotrophic annelid larvae.

## Different dynamics of new genes

New, species-specific genes, which account for a significant proportion of some larval transcriptomes[6,37], could also contribute to and explain the transcriptomic differences among annelid larvae. In *O. fusiformis*, *C. teleta* and *D. gyrociliatus*, genes of metazoan and pre-metazoan origin tended to peak, dominate and be enriched at early development, whereas younger genes were more highly expressed in competent and juvenile stages (Extended Data Fig. 8a–e). By contrast, species-specific genes followed lineage-specific dynamics (Supplementary Fig. 32). These genes, for instance, were more expressed in the juveniles of *O. fusiformis* and *D. gyrociliatus*, but in the blastula and gastrula of *C. teleta* (and to some extent also at the blastula stage in *O. fusiformis*; Extended

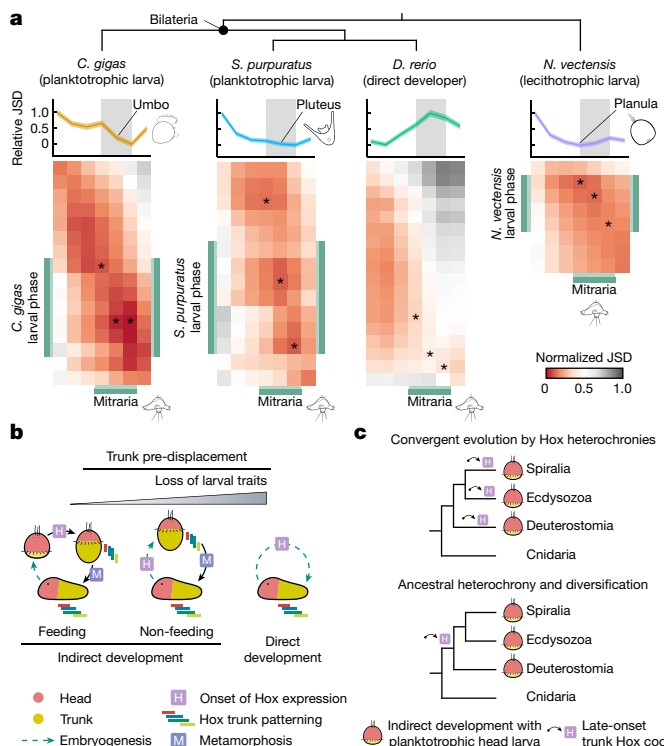

**Fig. 4 | The evolution of life cycles in Annelida and Bilateria. a**, Heatmaps of pairwise normalized Jensen–Shannon divergence (JSD) between *O. fusiformis* and *Crassostrea gigas*, *Strongylocentrotus purpuratus*, *D. rerio* and *Nematostella vectensis*. Asterisks indicate the stages of minimal JSD of each species to the larval phase of *O. fusiformis*. Larval phases are highlighted in green. Average relative JSD of the stages of minimal divergence to each *O. fusiformis* stage is shown on top of each heatmap. Confidence intervals represent the standard deviation from 250 bootstrap resamplings of the orthologue sets. See Extended Data Fig. 9a and Extended data Fig. 10a for fully labelled heatmaps. **b**, Schematics of the three main types of life cycles and the timing of Hox gene expression in bilaterians. Compared to indirect development with feeding larvae, lineages with non-feeding larvae and direct development pre-displace (that is, initiate earlier) trunk differentiation and Hox gene expression. Larval organs are reduced in non-feeding larvae and absent in direct development. **c**, Proposed alternative scenarios for the evolution of maximal indirect development with head larvae in Bilateria. Top, head larvae evolved convergently by repeatedly shifting trunk development (as seen by Hox gene expression) to pre-metamorphic stages. Bottom, head and trunk development were ancestrally temporally decoupled, which could have facilitated the evolution of head larvae in different bilaterian lineages.

Data Fig. 8a,c,d). Species-specific genes were only enriched and over-represented at larval stages in *C. teleta* (Extended Data Fig. 8f–h). Therefore, genes of different evolutionary origins contribute to the development of annelid larvae. This result suggests that the increased use of new genes in some lophotrochozoan larvae[6,37] might be due to the evolution of lineage-specific larval traits.

## Similarities between bilaterian larvae

To assess whether the transcriptional dynamics found in annelids are also observed in other metazoans, we extended our comparative transcriptomic approach to nine other animal lineages. In relative terms, global transcriptional dynamics between *O. fusiformis* and other animals tended to be more dissimilar at early development than at juvenile and adult stages (Fig. 4a and Extended Data Figs. 9a,b and 10a). The exception was the direct developer *Danio rerio*, for which the mitraria larva was the most dissimilar stage (Fig. 4a). This was also the case when comparing *O. fusiformis* with the direct-developing annelid *D. gyrociliatus* (Extended Data Fig. 3b). Notably, *O. fusiformis* shared

maximal transcriptomic similarities during larval phases with bilaterian species with planktotrophic ciliated larvae and even cnidarian planulae (Fig. 4a and Extended Data Fig. 9a–e). Genes involved in core cellular processes directly contributed to these similarities, which probably reflects common structural and ecological needs of metazoan larvae (Extended Data Fig. 9f,g). However, transcription factor expression levels were also maximally similar between those species at larval phases (Extended Data Fig. 9a,b,e). Therefore, adult development is generally more similar[9] than early embryogenesis across major animal lineages, but phylogenetically distant animal larvae also exhibit unexpected genome-wide transcriptional—and potentially regulative—similarities.

## Discussion

Our study provides a perspective on life cycle evolution in Bilateria. The planktotrophic larva of *O. fusiformis* defers trunk differentiation to late pre-metamorphic stages and largely develops from anterior ectodermal domains. This occurs in other feeding annelid larvae[38] (Extended Data Fig. 5f), and probably in Chaetopteriformia[39,40], and thus the late differentiation of the adult trunk might be an ancestral trait to Annelida (Extended Data Fig. 10b). Delaying trunk development to post-larval stages also occurs in phylogenetically distant clades within Spiralia[16,17], Ecdysozoa[14,41] and Deuterostomia[15,42,43], the larvae of which are generally referred to as head larvae[13,14]. By contrast, non-feeding larvae[32,44] and direct developers[30] in both Annelida and other bilaterian taxa[45,46] start to pattern their trunks with or immediately after the onset of anterior or head patterning, which always takes place before gastrulation in bilaterians[47,48]. Therefore, heterochronies in trunk development correlate with, and possibly account for, the evolution of different life cycles in animals (Fig. 4b). This differs from previously proposed mechanisms to explain the origins of animal life cycles, namely co-option of adult genes into larval-specific regulatory programmes[9,10] and independent evolution of adult gene regulatory modules[2,49].

Bilaterian head larvae could be lineage-specific innovations associated with the evolution of maximal indirect development[13,14,16] that evolved convergently by delaying trunk differentiation and Hox patterning (Fig. 4c). The similarities in larval molecular patterns[5,15,16] would then reflect ancient gene regulatory modules that were independently co-opted to develop analogous cell types and larval organs. Alternatively, the post-embryonic onset of trunk differentiation and Hox expression might be the most parsimonious ancestral state for Bilateria (Extended Data Fig. 10c,d and Supplementary Table 93). This could have facilitated the evolution of larvae, which would then originally share anterior genetic modules for their development (Fig. 4c). Regardless of the scenario and despite their limitations, our datasets highlight the importance of heterochronic changes for the diversification of bilaterian life cycles. The data also uncover a reduced set of candidate genes and regulatory motifs that might influence life cycle differences in Annelida and perhaps even Bilateria. In the future, comparative functional studies of these and other genes will reveal how temporal changes in gene expression and regulation have shaped the evolution of animal larvae and adults.

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

## Methods

### Adult culture, spawning and in vitro fertilization

Sexually mature *O. fusiformis* adults were collected from subtidal waters near the Station Biologique de Roscoff and cultured in the laboratory as previously described[23]. In vitro fertilization and collection of embryonic and larval stages were performed as previously described[23]. *C. teleta* Blake, Grassle & Eckelbarger, 2009 was cultured, grown and sifted, and its embryos and larvae were collected following established protocols[28]. *Magelona* spp. were collected in muddy sand from the intertidal of Berwick-upon-Tweed, Northumberland, NE England (around 55° 46′00.4″ N, 1° 59′04.5″ W) and kept initially in aquaria at the Museum Wales before their transfer to Queen Mary University of London, where they were kept in aquaria with artificial sea water.

### Genome size measurements

To estimate the haploid DNA nuclear content of *O. fusiformis*, we used a flow cytometer Partex CyFlow Space fitted with a Cobalt Samba green laser (532 nm, 100 mW) and the built-in software Flo-Max (v.2.82) as described for the annelid *D. gyrociliatus*[23], with adult individuals of *Drosophila melanogaster* as reference. Additionally, we used Jellyfish (v.2.3)[50] to count and generate a 31-mer histogram from adaptor-cleaned, short-read Illumina reads (see section below) and GenomeScope (v.2.0)[51] to obtain an in silico estimation of the genome size and heterozygosity of *O. fusiformis*.

### Genome sequencing, assembly and quality check

Ultra-high molecular weight (UHMW) gDNA was extracted following the Bionano genomics IrysPrep agar-based, animal tissue protocol using sperm from a single *O. fusiformis* male. UHMW gDNA was cleaned up using a salt–chloroform wash following PacBio's recommendations before long-read sequencing using PacBio (v.3.0) chemistry at the University of California Berkeley. A total of 16 SMRT cells of PacBio Sequel were used for sequencing with 600 min movie time, producing a total of 170.07 Gb of data (10.72 million reads, N50 read length between 25.75 kb and 30.75 kb). In addition, we used UHMW gDNA of that same individual to generate a 10x Genomics linked reads library, which we sequenced in an Illumina HiSeq4000 at the Okinawa Institute of Science and Technology to produce 28.62 Gb of data (141.66 million read pairs). PacBio reads were assembled with CANU (v.8.3rc2)[52] assuming 'batOptions="−dg 3 −db 3 −dr 1 −ca 500 −cp 50' and 'correctedError-Rate = 0.065'. Pacbio reads were remapped using pbalign (v.0.3.2) and the assembly polished once using Arrow (genomicconsensus, v.2.3.2). Then Illumina paired-end reads generated with the 10x Genomics linked reads were extracted, remapped using bwa mem (v.0.7.17)[53] and used for polishing with Racon (v.1.16)[54]. Bionano Genomics optical mapping data were used to scaffold the PacBio-based assembly, which was de-haploidized with purge_haplotigs (v.1.0.4)[55] setting cut-off values at 35, 85 and 70× coverage to reconstruct a high-quality haploid reference assembly. HiC-based chromosome scaffolding was performed as described below. Merqury (v.1.1)[56] and BUSCO (v.5)[57] were used to assess genome completeness and to evaluate the quality of the assembly (Supplementary Fig. 1).

### Transcriptome sequencing

Fourteen samples spanning key developmental time points of the *O. fusiformis* life cycle, including active oocyte, zygote, 2-cell, 4-cell and 8-cell stages, 3 h post-fertilization (h.p.f.), 4 h.p.f., coeloblastula (5 h.p.f.), gastrula (9 h.p.f.), axial elongation (13 h.p.f.), early larva (18 h.p.f.), mitraria larva (27 h.p.f.), pre-metamorphic competent larva (3 weeks post-fertilization) and post-metamorphic juvenile were collected in duplicates (except for the latter), flash frozen in liquid nitrogen and stored at −80 °C for total RNA extraction. Samples within replicates were paired, with each one containing around 300 embryos or 150 larvae coming from the same in vitro fertilization process. Nine further samples from adult tissues and body regions (blood vessel, body wall, midgut, prostomium, head, ovary, retractor muscle, tail and testes) were also collected as described above. Likewise, an additional five samples spanning post-cleavage time points of *C. teleta*, including 64 cells and gastrula stages, and stage 4tt, stage 5 and stage 7 larval stages, were collected in duplicates. Total RNA was isolated using a Monarch Total RNA Miniprep kit (New England Biolabs) following the supplier's recommendations. Total RNA samples from developmental stages from both *O. fusiformis* and *C. teleta* were used to prepare strand-specific mRNA Illumina libraries that were sequenced at the Oxford Genomics Centre (University of Oxford, UK) over three lanes of an Illumina NovaSeq6000 system in 2 × 150 bp mode to a depth of around 50 million reads (Supplementary Tables 13 and 16). Adult tissue samples were sequenced at BGI on a BGISeq-500 platform in 2 × 100 bp mode to a depth of about 25 million reads (Supplementary Table 49).

### Annotation of repeats and transposable elements

RepeatModeler (v.2.0.1)[58] and RepBase were used to construct a de novo repeat library for *O. fusiformis*, which was then filtered for bona fide genes using the predicted proteome of *C. teleta*. In brief, we used DIAMOND (v.0.9.22)[59] with an *e*-value cut-off of $1 \times 10^{-10}$ to identify sequences in the de novo repeat library with significant similarity to protein-coding genes in *C. teleta* that are not transposable elements (TEs). Sequences with a significant hit were manually inspected to verify they were not TEs; if they were, they were manually removed from the de novo repeat library. The filtered consensus repeat predictions were then used to annotate the genome assembly of *O. fusiformis* with RepeatMasker open-4.0. We next used LTR_finder (v.1.07)[60], a structural search algorithm, to identify and annotate long tandem repeats (LTRs). Finally, we generated a consensus set of repeats by merging RepeatMasker and LTR_finder predictions with RepeatCraft[61], using default parameters but a maximum LTR size of 25 kb (as derived from the LTR_finder annotation) (Supplementary Table 1). The general feature format (GFF) and fasta files with the annotation of TEs and repeats are available in the GitHub repository (see Data availability section).

### Gene prediction and functional annotation

We used SAMtools (v.1.9)[62] and the annotation of repeats to soft mask *O. fusiformis* genome assembly before gene prediction. We then mapped all embryonic and adult transcriptomes and a publicly available dataset[63] (Sequence Read Archive (SRA) identifier: SRR1222288) with STAR (v.2.5.3a)[64] after removing low-quality read pairs and read pairs containing Illumina sequencing adapters with trimmomatic (v.0.39)[65]. StringTie (v.1.3.6)[66] was used to convert STAR alignments into gene transfer format (GTF) files and Portcullis (v.1.1.2)[67] to generate a curated set of splice junctions. Additionally, we generated de novo transcriptome assemblies for all samples using Trinity (v.2.5.1)[68] with default parameters, which were thereafter mapped to the soft-masked assembly with GMAP (v.2020-04-08)[69]. We then ran the default Mikado (v.2.1) pipeline[70] to merge all transcriptomic evidence and reliable splice junctions into a single set of best-supported transcripts and gene models. From this merged dataset, we filtered full-length, non-redundant transcripts with a BLAST hit on at least 50% of their length and at least two exons to obtain a gene set that we used to train Augustus (v.3.2.3)[71]. Simultaneously, we used the Mikado gene annotation and Portcullis splice junctions to generate confident sets of exon and intron hints, respectively. We also ran Exonerate (v.2.4.0)[72] to generate spliced alignments of the proteome of *C. teleta* proteome on *O. fusiformis* soft-masked genome assembly to obtain further gene hints. We then merged all exon and intron hints into a single dataset, which we passed into Augustus (v.3.2.3)[71] for ab initio gene prediction. Finally, PASA (v.2.3.3)[73] was used to combine RNA-seq and ab initio gene models into a final gene set, from which spurious predictions with in-frame stop codons (228 gene models), predictions that overlapped with repeats (5,779 gene models) and that had high similarity

to TEs in the RepeatPeps.lib database (2,450 models) were removed. This filtered gene set included 26,966 genes, encompassing 31,903 different transcripts (Supplementary Fig. 1). To assess the completeness of this annotation, we ran BUSCO (v.5)[57] in proteome mode, which resulted in 97.7% of the core genes present. Moreover, 31,678 out of the 31,903 (99.29%) filtered transcripts were supported by RNA-seq data, and 80.69% of the transcripts had a significant BLAST match (e-value cut-off < 0.001) to a previously annotated annelid gene (database containing non-redundant proteomes of the high-quality annelid genomes of *C. teleta*, *D. gyrociliatus*, *Eisenia andrei*, *Lamellibrachia luymesi*, *Paraescarpia echinospica*, *Riftia pachyptila* and *Streblospio benedicti*). A similar functional annotation approach was followed to re-annotate the genome of *C. teleta* with the new RNA-seq data, using as starting assembly the soft-masked version available at Ensembl Metazoa. This resulted in 41,221 transcripts, 39,814 of which had RNA-seq support (96.59%). Additionally, 80.47% of the transcripts had a significant BLAST match (e-value cut-off < 0.001) to other well-annotated annelid genomes (see above).

Protein homologies for the filtered transcripts of *O. fusiformis* and *C. teleta* were annotated using BLAST (v.2.2.31+)[74] with the UniProt/SwissProt database provided with Trinotate (v.3.0)[75]. We used HMMER (v.2.3.2)[76] to identify protein domains using Trinotate's PFAM-A database and signalP (v.4.1)[77] to predict signal peptides. These functional annotations were integrated into a Trinotate database, which retrieved Gene Ontology (GO), eggNOG and Kyoto Encyclopedia of Genes and Genomes (KEGG) terms for each transcript. In addition, we ran the PANTHER HMM scoring tool to assign a PantherDB[78] orthology identifier to each transcript. In total, we retrieved a functional annotation for 22,516 transcripts (63.86%). Functional annotation reports are provided in the GitHub repository (see Data Availability section).

## Chromosome-scale scaffolding

Sperm from a single *O. fusiformis* worm and an entire sexually mature male were used as input material to construct two Omni-C Dovetail libraries following the manufacturer's recommendations for marine invertebrates. These libraries were sequenced in an Illumina NovaSeq6000 at the Okinawa Institute of Science and Technology to a depth of 229 and 247 million reads. HiC reads were processed using the Juicer pipeline (r.e0d1bb7)[79] to generate a list of curated contracts ('merged no dups') that was subsequently used to scaffold the assembly using 3d-dna (v.180419)[80]. The resulting assembly and contact map were visually inspected and curated using Juicebox (v.1.11.08)[79], and adjustments were submitted for a subsequent run of optimization using 3d-dna. Finally, repeats and TEs were re-annotated in this chromosome-scale assembly as described above, and the annotation obtained for the PacBio-based assembly was lifted over with Liftoff (v.1.6.1)[81] (Supplementary Fig. 1). All gene models but two were successfully re-annotated in the chromosome-scale assembly.

## Gene family evolution analyses

We used the AGAT suite of scripts to generate non-redundant proteomes with only the longest isoform for a set of 21 metazoan proteomes (Supplementary Table 2). To reconstruct gene families, we used OrthoFinder (v.2.2.7)[82] using MMSeqs2 (ref. [83]) to calculate sequence similarity scores and an inflation value of 2. OrthoFinder gene families were parsed and mapped onto a reference species phylogeny to infer gene family gains and losses at different nodes and tips using the ETE 3 library[84], as well as to estimate the node of origin for each gene family. Gene expansions were computed for each species using a hypergeometric test against the median gene number per species for a given family using previously published code[30] (Supplementary Tables 3–7). Principal component analysis was performed on the orthogroups matrix by metazoan lineage, given that orthogroups were present in at least three of the 22 analysed species, to eliminate taxonomically restricted genes. All single copy orthologue files derived from this analysis used

throughout the study are available in the GitHub repository (see Data Availability section).

## Macrosynteny analyses

Single-copy orthologues obtained using the mutual best hit approach generated using MMseqs2 (ref. [83]) using the annotations of *Branchiostoma floridae*[85], *P. maximus*[86], *S. benedictii*[87] and *Lineus longissimus*[88,89] were used to generate Oxford synteny plots comparing sequentially indexed orthologue positions. Plotting order was determined by hierarchical clustering of the shared orthologue content using the complete linkage method as originally proposed. Comparison of the karyotype of all four species was performed using the Rideogram package by colouring pairwise orthologues according to the ALG assignment in comparisons with *P. maximus* and *B. floridae*.

## Evolutionary analysis of *chordin* in annelids

The identification of *chordin* (*chrd*) and *chordin-like* (*chrdl*) genes in *O. fusiformis* was based on the genome functional annotation (see above). To mine *chrd* orthologues, 81 annelid transcriptomic datasets were downloaded from the SRA (Supplementary Table 8) and assembled using Trinity (v.2.5.1)[68] to create BLAST local nucleotide databases. We also created a nucleotide database for *C. teleta* using its annotated genome[90] (European Nucleotide Archive (ENA) accession number: GCA_000328365.1). Human and *O. fusiformis* CHRD proteins were used as queries to find *chrd* orthologues following the mutual best hit approach (e-value ≤ 10⁻³), obtaining 103 distinct candidate *chrd* transcripts that were then translated (Supplementary Table 9). A single candidate CHRD protein for *Themiste lageniformis* (M. J. Boyle, unpublished data) was included ad hoc at this step. In addition, 15 curated CHRD and CHRDL protein sequences (and an outgroup) were obtained from various sources (Supplementary Table 10) and aligned together with *O. fusiformis* CHRD and CHRDL sequences in MAFFT (v.7)[91] with the G-INS-I iterative refinement method and default scoring parameters. From this mother alignment, further daughter alignments were obtained using "mafft --addfragments"[92], the accurate "--multipair" method, and default scoring parameters. For orthology assignment, two phylogenetic analyses were performed on selected candidate sequences, which included the longest isoform for each species–gene combination, given that it included a 10-residue or longer properly aligned fragment in either the CHRD domains or the von Willebrand factor type C (VWFC) domains. vWFC and CHRD domains were trimmed and concatenated using domain boundaries defined by ProSITE domain annotation for the human chordin precursor protein (UniProt: Q9H2X0). Either all domains or the VWFC domains only were used for phylogenetic inference (Extended Data Fig. 2c,d and Supplementary Tables 11 and 12) with a WAG amino acid replacement matrix[93] to account for transition rates, the FreeRate heterogeneity model (R4)[94] to describe sites evolution rates, and an optimization of amino acid frequencies using maximum likelihood using IQ-TREE (v.2.0.3)[95]. 1,000 ultrafast bootstraps[96] were used to extract branch support values. Bayesian reconstructions in MrBayes (v.3.2.7a)[97] were also performed using the same WAG matrix but substituting the R4 model for the discrete gamma model[98], with 4 rate categories (G4). All trees were composed in FigTree (v.1.4.4). Alignment files are available in the GitHub repository (see Data availability section).

## Gene expression profiling

We profiled gene expression dynamics from blastula to juvenile stages for *O. fusiformis*, from 64-cell to competent larva stages for *C. teleta* (Supplementary Fig. 2), from early development to female adult stages for *D. gyrociliatus*, and across the 9 adult tissues samples of *O. fusiformis*. Sequencing adaptors were removed from raw reads using trimmomatic (v.0.39)[65]. Cleaned reads were pseudo-aligned to the filtered gene models using kallisto (v.0.46.2)[99], and genes with an expression level above an empirically defined threshold of 2 transcripts per million

(TPM) were deemed expressed. For each species, the DESeq2 (v.1.30.1) package[100] was used to normalize read counts across developmental stages (Supplementary Tables 13–21) and adult tissues (Supplementary Tables 49–51) and to perform pairwise differential gene expression analyses between consecutive developmental stages. $P$ values were adjusted using the Benjamini–Hochberg method for multiple testing correction. We defined a gene as significantly upregulated for a $\log_2$(fold-change) (LFC) > 1 or downregulated for a LFC < 1, given an adjusted $P$ value < 0.05. Principal component analyses were performed on the variance stabilizing-transformed matrices of the normalized DESeq2 matrices. For the *O. fusiformis* adult tissues samples, genes specifically expressed (TPM > 2) only in both the head and head plus two anterior-most segment samples were classified as adult anterior genes, and those expressed only in both the tail and the body wall were classified as adult trunk and posterior genes (Supplementary Tables 52 and 53). For all three annelid taxa, anterior, trunk and posterior markers were defined as genes for which their spatial expression pattern has been validated through in situ hybridization in the literature (Supplementary Tables 54–56). TPM and DESeq2 gene expression matrices of developmental and adult tissue samples are also available in the GitHub repository (see Data availability section).

### Gene clustering and co-expression network analyses
Transcripts were clustered according to their normalized DESeq2 expression dynamics through soft $k$-means clustering (or soft clustering) using the mfuzz (v.2.52) package[101] (Supplementary Tables 23–26). Out of the total number of transcripts, we discarded those that were not expressed at any developmental stage (225 out of 31,903 for *O. fusiformis*, 1,407 out of 41,221 for *C. teleta*, and 200 out of 17,388 for *D. gyrociliatus*). We then determined an optimal number of 12 clusters (*O. fusiformis* and *C. teleta*) and 9 clusters (*D. gyrociliatus*) for our datasets by applying the elbow method to the minimum centroid distance as a function of the number of clusters. For construction of the gene co-expression networks for *O. fusiformis* and *C. teleta*, we used the WGCNA package (v.1.70-3)[102]. All transcripts expressed at any developmental stage were used to build a signed network with a minimum module size of 300 genes and an optimized soft-thresholding power of 16 and 8 for *O. fusiformis* and *C. teleta*, respectively. Block-wise network construction returned 15 gene modules for *O. fusiformis*, from which 1 module was dropped owing to poor intramodular connectivity, and 19 gene modules for *C. teleta* (Supplementary Tables 23 and 24). The remaining 14 gene modules of *O. fusiformis* (A–N) and 19 gene modules of *C. teleta* (A–O, W–Z) were labelled with distinct colours, with unassigned genes labelled in grey. Random subsets consisting of the nodes and edges of 30% of the transcripts were fed into Cytoscape (v.3.8.2)[103] for network visualization (Supplementary Fig. 9). Module eigengenes were chosen to summarize the gene expression profiles of gene modules. GO enrichment analysis of each gene cluster and gene module was performed using the topGO (v.2.44) package. We performed a Fisher's exact test and listed the top 30 (soft $k$-means clusters) or top 15 (WGCNA modules) significantly enriched GO terms of the class biological process (Supplementary Tables 27–31, Supplementary Figs. 4–6, 10 and 11). To ease visualization, all 486 non-redundant enriched GO terms from the 33 soft $k$-means clusters from all 3 species were clustered through $k$-means clustering by semantic similarity using the simplifyEnrichment (v.1.2.0) package[104] (Supplementary Figs. 7 and 8). Full network nodes and edges files and the random 30% network subset files are available in the GitHub repository (see Data availability section).

### Transcription factor repertoire analysis
We selected a custom set of 36 transcription factor classes from all 9 transcription factor superclasses from the TFClass database[105]. Transcripts in *O. fusiformis*, *C. teleta* and *D. gyrociliatus* were deemed transcription factors and classified into one or more of the 36 classes if they were a match for any of the corresponding PANTHER identifiers (Supplementary

Tables 32–33 and Supplementary Fig. 3). Over-representation and under-representation of the different transcription factor classes in the gene expression clusters was tested through pairwise two-tailed Fisher's exact tests, for which we then adjusted the $P$ values using Benjamini–Hochberg correction for multiple testing.

### Orthogroup overlap analysis
We performed pairwise comparisons between each possible combination of soft $k$-means clusters of all three annelid taxa. The numbers of overlapped orthogroups between either the full clusters or the transcription factors belonging only to each cluster were subjected to upper-tail hypergeometric tests. $P$ values were then adjusted using the Benjamini–Hochberg method for multiple testing correction. For the simplified analyses by quadrants, clusters were classed as early/pre-larval (*O. fusiformis*: 1–6; *C. teleta*: 1–5; *D. gyrociliatus*: 1–3) or late/pre-larval (*O. fusiformis*: 8–12; *C. teleta*: 7–12; *D. gyrociliatus*: 5–7), thus rendering 4 different quadrants for each species pairwise comparison: $early_{species A}$–$early_{species B}$, $early_{species A}$–$late_{species B}$, $late_{species A}$–$early_{species B}$ and $late_{species A}$–$late_{species B}$. Clusters corresponding to female adult expression in *D. gyrociliatus* (8 and 9) were discarded for comparison purposes. Relative similarity (RS) values for each of the four quadrants were computed as the following ratio:

$$RS = \frac{mean(-\log_{10}(adjusted\ P\ value)_{quadrant})}{mean(-\log_{10}(adjusted\ P\ value)_{total})}$$

Values above 1 indicate a higher orthogroup overlap than average, whereas values below 1 represent a lower overlap than average. For genes under heterochronic shifts—that is, with distinct temporal expression dynamics—between indirect and direct development, a gene set was constructed with the genes with a single-copy orthologue in both *O. fusiformis* and *C. teleta*, for which expression was shifted from post-larval clusters (*O. fusiformis*: 7–12; *C. teleta*: 8–12) to early clusters 2 and 3 in *D. gyrociliatus* (Fig. 2b, Supplementary Tables 34 and 35 and Supplementary Fig. 12). For the characterization of genes under heterochronic shifts between planktotrophic and lecithotrophic larvae, two gene sets were generated with the genes with $early_{O. fusiformis}$–$late_{C. teleta}$ and $late_{O. fusiformis}$–$early_{C. teleta}$ dynamics, as described above (Supplementary Tables 36–39 and Supplementary Figs. 13 and 14). GO enrichment analysis of both gene sets was performed using the topGO (v.2.44) package. We performed a Fisher's exact test and listed the top 15 significantly enriched GO terms of the class biological process (Supplementary Table 40). BlastKOALA[106] server was used to assign a KEGG orthology number to one-to-one orthologues showing heterochronic sifts and KEGG mapper[107] to analyse the annotations (Supplementary Tables 41 and 42).

### Pathway analyses
Human genes involved in the animal autophagy pathway (map04140) were obtained from the KEGG pathway database[108]. *D. melanogaster* and *Saccharomyces cerevisiae* genes involved in the chitin synthesis pathway were fetched from FlyBase[109] and SGD[110], respectively, based on the enzyme nomenclature numbers of the pathway enzymatic activities[111]. Orthology in *O. fusiformis* and *C. teleta* for the autophagy pathway genes was determined from the single-copy orthologue sets to the human genes, for which one for both species existed (Supplementary Tables 43 and 44). For the chitin synthesis pathway, and owing to the high number of paralogues and expansions and losses of enzymatic activities of the chitin synthesis pathway, orthology was inferred from PANTHER family and subfamily identifiers to the corresponding enzymatic activities (Supplementary Tables 45 and 46). We then used this orthology to reconstruct the chitin synthesis pathway in annelids. Timing across both species and the presence or lack thereof of heterochronic shifts between *O. fusiformis* and *C. teleta* were determined as described above (Supplementary Figs. 15 and 16).

## Hox genes orthology assignment

A total of 129 curated Hox sequences were retrieved from various databases (Supplementary Table 47) and aligned with *O. fusiformis* HOX proteins with MAFFT (v.7) in automatic mode. Poorly aligned regions were removed with gBlocks (v.0.91b)[112] to produce the final alignments. Maximum likelihood trees were constructed using RAxML (v.8.2.11.9)[113] with an LG substitution matrix[114] and 1,000 ultrafast bootstraps. All trees were composed in FigTree (v.1.4.4). Alignment files are available in the GitHub repository (see Data availability section).

## Whole-mount in situ hybridization and immunohistochemistry

Fragments of *chordin* and Hox genes were isolated as previously described[24] using gene-specific oligonucleotides and a T7 adaptor. Riboprobes were synthesized using a T7 MEGAscript kit (ThermoFisher, AM1334) and stored at a concentration of 50 ng μl⁻¹ in hybridization buffer at –20 °C. Whole-mount in situ hybridization in embryonic, larval and juvenile stages were conducted as described elsewhere[24,26]. Antibody staining in larval stages of *O. fusiformis*, *Magelona* spp. and *C. teleta* was carried out as previously described[23,115] using the following antibodies: mouse anti-acetyl-α-tubulin antibody, clone 6-11B-1, 1:800 dilution (Sigma-Aldrich, MABT868, RRID: AB_2819178) and goat anti-mouse IgG (H+L) cross-adsorbed secondary antibody, Alexa Fluor 647, 1:800 dilution (Thermo Fisher Scientific, A-21235, RRID: AB_2535804). Differential interface contrast images of the colorimetric in situ were obtained using a Leica 560 DMRA2 upright microscope equipped with an Infinity5 camera (Lumenera). Fluorescently stained samples were scanned using a Nikon CSU-W1 spinning disk confocal microscope.

## ATAC-seq

We performed two replicates of ATAC-seq from samples containing around 50,000 cells at the blastula (about 900 embryos), gastrula (around 500), elongation (about 300), mitraria larva (around 150 larvae) and competent larva (about 40) stages for *O. fusiformis*, and the 64-cells stage (about 500 embryos), gastrula (around 200), stage 4tt larva (about 120 larvae), stage 5 larva (around 90) and stage 8 larva (around 50) for *C. teleta* following the omniATAC protocol[116], but gently homogenizing the samples with a pestle in lysis buffer and incubating them on ice for 3 min. Tagmentation was performed for 30 min at 37 °C with an in-house purified Tn5 enzyme[117]. After DNA clean-up, ATAC-seq libraries were amplified as previously described[116]. Primers used for both PCR and quantitative PCR are listed in Supplementary Tables 57 and 59. Amplified libraries were purified using ClentMag PCR Clean Up beads as indicated by the supplier and quantified and quality checked on a Qubit 4 fluorometer (ThermoFisher) and an Agilent 2200 TapeStation system before pooling at equal molecular weight. Sequencing was performed on an Illumina HiSeq4000 platform in 2 × 75 bp mode at the Oxford Genomics Centre (blastula, elongation and mitraria larva stages, and one replicate of the gastrula sample of *O. fusiformis*, as well as the 64-cells, gastrula and stage 4tt larva stages of *C. teleta*) and on an Illumina NovoSeq6000 in 2 × 150 bp mode at Novogene (one replicate of gastrula and the two replicates of competent larva stages of *O. fusiformis* and the two replicates of stage 5 and stage 8 larva of *C. teleta*).

## Chromatin accessibility profiling

We used cutadapt (v.2.5)[118] to remove sequencing adaptors and trim reads from libraries sequenced in 2 × 150 bp mode to 75 bp reads. Quality filtered reads were mapped using NextGenMap (v.0.5.5)[119] in paired-end mode, duplicates were removed using samtools (v.1.9)[120] and mapped reads were shifted using deepTools (v.3.4.3)[121] (Supplementary Tables 58 and 60). Fragment size distribution was estimated from resulting BAM files and transcription start site enrichment analysis was computed using computeMatrix and plotHeatmap commands in deepTools (v.3.4.3). Peak calling was done using MACS2 (v.2.2.7.1)[122,123] (-f BAMPE --min-length 100 --max-gap 75 and -q 0.01). Reproducible peaks were identified by irreproducible discovery rates (values <0.05) (v.2.0.4) at each developmental stage. Peaks from repetitive regions were filtered using BEDtools (v.2.28.0)[124] at each developmental stage. Next we used DiffBind (v.3.0.14)[125] to generate a final consensus peak set of 63,732 peaks in *O. fusiformis* and 46,409 peaks in *C. teleta*, which were normalized using DESeq2 (Supplementary Fig. 17). Peak clustering according to accessibility dynamics was performed as described above for RNA-seq, using the same number of 12 clusters to make both profiling techniques comparable. Principal component analysis and differential accessibility analyses between consecutive developmental stages were also performed as described above. An LFC > 0 and a LFC < 0 indicates whether a peak opens or closes, respectively, given an adjusted *P* value < 0.05. Stage-specific and constitutive peaks were determined using UpSetR (v.1.4.0)[126], and both the consensus peak set and the stage-specific peak sets were classified by genomic region using HOMER (v.4.11)[127] and further curated. Visualization of peak tracks and gene structures was conducted using pyGenomeTracks (v.2.1)[128] and deepTools (v.3.4.3)[121]. To correlate chromatin accessibility and gene expression, this genomic region annotation was used to assign peaks to their closest gene (63,726 peaks were assigned to 23,025 genes in *O. fusiformis* and 44,368 peaks were assigned to 23,382 genes in *C. teleta*). Pearson correlation coefficient between chromatin accessibility and gene expression was computed individually by peak using two-sided tests (Supplementary Fig. 18). GO enrichment analysis of the gene sets regulated by peak clusters was performed using the topGO (v.2.44) package. We performed Fisher's exact test and listed the top 30 significantly enriched GO terms of the class biological process (Supplementary Figs. 19 and 20). To ease visualization, all 242 non-redundant enriched GO terms were clustered through *k*-means clustering by semantic similarity using the simplifyEnrichment (v.1.2.0) package[104] (Supplementary Tables 61–71 and Supplementary Figs. 21–23). Coverage files and peak set files are available in the GitHub repository (see Data availability section).

## Motif identification, clustering, matching and curation

To identify transcription-factor-binding motifs in chromatin accessible regions in the two species, we first used HOMER[127] (v.4.1) to identify known and de novo motifs in the consensus peak sets, which produced 456 motifs for *O. fusiformis* and 364 motifs for *C. teleta* (Supplementary Tables 72 and 73). Significance of motifs was derived from binomial tests from cumulative binomial distributions. We then used GimmeMotifs (v.0.16.1)[129] with a 90% similarity cut-off to cluster the motifs predicted in *O. fusiformis* and *C. teleta* into 141 consensus motifs, which we matched against four motif databases to assign their putative identity (Gimme vertebrate (5.0)[129], HOMER[127], CIS-BP[130] and a custom JASPAR2022 (ref. [131]) core motifs without plant and fungi motifs; Supplementary Fig. 24). We then used the human non-redundant TF motif database (https://resources.altius.org/~jvierstra/projects/motif-clustering-v2.0beta/) to manually curate the annotation. After removing motifs that probably represented sequence biases, we finally obtained 95 motif archetypes for *O. fusiformis* and 91 for *C. teleta* (Supplementary Table 74), which we then used to perform motif counts in peaks (Supplementary Tables 75 and 76) and motif accessibility estimation (Supplementary Tables 77 and 78) with GimmeMotifs (v.0.16.1)[129]. Data clustering was performed with mfuzz (v.2.52)[101] (Supplementary Figs. 25 and 27). Over-representation and under-representation of counts of the common curated motif archetypes in the peak accessibility soft clusters (see above) was tested through pairwise two-tailed Fisher's exact tests, for which we then adjusted the *P* values using the Bonferroni correction for multiple testing.

## Transcription factor footprinting and Hox gene regulatory network exploration

To predict transcription factor binding, as a proxy of activity, we conducted footprinting analysis using TOBIAS (v.0.12.0)[132] during development in the 95 and 91 motif archetypes for *O. fusiformis* and *C. teleta*,

respectively (Supplementary Tables 79 and 80). Bound and unbound sites were first estimated by fitting a two-component Gaussian-mixture model, and significance was then tested using a one-tail test from the right-most normal distribution. Transcription factor binding scores (TFBSs) were clustered using mfuzz (v.2.52)[101]. Pearson correlation coefficients of motif accessibility and TFBSs were calculated by stage and by motif separately on the basis of 33 common, curated motif archetypes (Supplementary Figs. 26 and 28–30). To reconstruct potential upstream regulators and downstream effectors of the Hox genes, we first subset ATAC-seq peaks annotated to the Hox genes in the Hox cluster (that is, all except *Post1*) in *O. fusiformis* and *C. teleta* and extracted the bound motifs on those peaks (Supplementary Tables 81 and 82). TFBSs were summed for each motif to obtain global dynamics, and their temporal dynamics were then clustered using mfuzz (v.2.52)[101] (Supplementary Fig. 31). For the downstream genes regulated by Hox, we obtained genes annotated to ATAC-seq peaks with a bound HOX, EVX and CDX motif at the competent stage in *O. fusiformis* and stage 4tt larva in *C. teleta* (Supplementary Tables 83 and 84). One-to-one orthologues were used to identified shared targets and PANTHER identifiers to obtain their functional annotation.

## Phylostratigraphy

To evaluate gene expression dynamics by phylostratum and developmental stage in all three annelid lineages, we used the OrthoFinder gene families and their inferred origins. We deemed all genes originating before and with the Cnidarian–Bilaterian ancestor of pre-metazoan and metazoan origin (Supplementary Tables 85–87). We then applied a quantile normalization onto the DESeq2-normalized matrices of gene expression. The 75th percentile of the quantile-normalized gene expression levels was used as the summarizing measure of the gene expression distribution by developmental stage. Over-representation and under-representation of the different phylostrata in the gene expression clusters were tested through pairwise two-tailed Fisher's exact tests, for which we then adjusted the *P* values using Bonferroni correction for multiple testing. Gene expression dynamics of new genes and genes of pre-metazoan and metazoan origin across selected metazoan lineages (see 'Comparative transcriptomics' section below) were also evaluated as described above (Supplementary Fig. 32).

## Comparative transcriptomics

Publicly available RNA-seq developmental time courses for the development of *Amphimedon queenslandica*, *Clytia hemisphaerica*, *N. vectensis*, *S. purpuratus*, *Branchiostoma lanceolatum*, *D. rerio*, *D. melanogaster*, *Caenorhabditis elegans*, *C. gigas*, *D. gyrociliatus*, and two stages of *C. teleta* were downloaded from the SRA using SRA-Toolkit (v.2.11.3) (Supplementary Table 88), cleaned for adaptors and low-quality reads with trimmomatic (v.0.39)[65] and pseudo-aligned to their respective non-redundant genome-based gene repertoires—that is, with a single transcript isoform, the longest, per gene model—using kallisto (v.0.46.2)[99]. We then performed a quantile transformation of TPM values using scikit-learn (v.1.0.2)[133] and calculated the Jensen–Shannon divergence (JSD) value from (1) all single-copy orthologues, (2) the set single-copy transcription factor orthologues and (3) the set of common single-copy orthologues across all lineages, either between all possible one-to-one species comparisons (1) or between all species and *O. fusiformis* (2 and 3), using the philentropy (v.0.5.0) package[134] as follows:

$$\mathrm{JSD}_{\mathrm{raw}}\left(P\|Q\right)=\frac{1}{2}\sum_{i=0}^{n}p_i\times\log_2\!\left(\frac{p_i}{\frac{1}{2}\left(p_i+q_i\right)}\right)+\frac{1}{2}\sum_{i=0}^{n}q_i\times\log_2\!\left(\frac{q_i}{\frac{1}{2}\left(p_i+q_i\right)}\right)$$

Transcriptomic divergences were calculated on the basis of 250 bootstrap replicates, from which statistically robust mean values and standard deviations were obtained. Raw mean JSD values ($\mathrm{JSD}_{\mathrm{raw}}$) were adjusted ($\mathrm{JSD}_{\mathrm{adj}}$) by dividing by the number of single-copy orthologues (1), single-copy transcription factor orthologues (2) or common

single-copy orthologues (3) of each comparison (Supplementary Tables 22, 89 and 90) and normalized using the minimum and maximum adjusted JSD values from all one-to-one species comparisons as follows:

$$\mathrm{JSD}_{\mathrm{norm}}\left(P\|Q\right)=\frac{\mathrm{JSD}_{\mathrm{adj}}\left(P\|Q\right)-\min\mathrm{JSD}_{\mathrm{adj}}}{\max\mathrm{JSD}_{\mathrm{adj}}-\min\mathrm{JSD}_{\mathrm{adj}}};\mathrm{JSD}_{\mathrm{norm}}\in[0,1]$$

Relative JSD values were obtained equally, using minimum and maximum adjusted JSD values from each one-to-one species comparison instead. Gene-wise JSD (gwJSD) between five key one-to-one larval stages comparisons was computed as follows:

$$\mathrm{gwJSD}\left(P\|Q\right)=\frac{1}{2}\times p_i\times\log_2\!\left(\frac{p_i}{\frac{1}{2}\left(p_i+q_i\right)}\right)+\frac{1}{2}\times q_i\times\log_2\!\left(\frac{q_i}{\frac{1}{2}\left(p_i+q_i\right)}\right)$$

Similarity-driving genes—that is, those with very low gwJSD—were subset as those below the threshold defined as 25% of the point of highest probability density of the gwJSD distributions. GO enrichment analysis of the similarity-driving gene sets was performed using the topGO (v.2.44) package. We performed Fisher's exact test and listed the top 30 significantly enriched GO terms of the class biological process (Supplementary Table 91). To ease visualization, all 51 non-redundant enriched GO terms from the 5 gene sets were clustered through *k*-means clustering by semantic similarity using the simplifyEnrichment (v.1.2.0) package[104]. The subsets of similarity-driven transcription factors of each pairwise comparison are listed in Supplementary Table 92. For comparative Hox gene expression dynamics profiling in metazoan lineages, the same non-redundant gene expression matrices were normalized using the DESeq2 (v.1.30.1) package[100] (Supplementary Fig. 33), unless Hox gene models were missing, in which case they were manually added ad hoc to the non-redundant genome-based gene repertoires (Supplementary Table 94). Hox gene expression profiling in *U. unicinctus* was performed as described for the rest of taxa but using the available reference transcriptome[135] instead (Supplementary Table 48). All gene expression matrices are available in the GitHub repository (see Data availability section).

## Reporting summary

Further information on research design is available in the Nature Portfolio Reporting Summary linked to this article.

## Data availability

Accession codes and unique identifiers to previously publicly available datasets we used for this study are listed in Supplementary Table 2 (genome files used in gene family evolution analyses), Supplementary Table 8 (transcriptomes used in the evolutionary analysis of *chordin* in annelids), Supplementary Tables 41 and 43 (gene identifiers used in pathway analyses), Supplementary Table 47 (sequence identifiers used in the orthology assignment of Hox genes), Supplementary Table 48 (RNA-seq datasets used for Hox gene expression profiling in *U. unicinctus*) and Supplementary Table 88 (RNA-seq datasets used for comparative annelid and metazoan transcriptomics and Hox gene expression profiling). Repetitive elements database RepBase can be accessed at https://www.girinst.org/repbase/. Transcription factor public database TFClass can be found at http://tfclass.bioinf.med.uni-goettingen.de/. All sequence data associated with this project are available at the European Nucleotide Archive (project PRJEB38497) and Gene Expression Omnibus (accession numbers GSE184126, GSE202283, GSE192478, GSE210813 and GSE210814). Genome assemblies, TE annotations, genome annotation files used for RNA-seq and ATAC-seq analyses, WGCNA nodes and edges files, alignment files used in orthology assignment and other additional files are publicly available at GitHub (https://github.com/ChemaMD/OweniaGenome).

## Code availability

All code used in this study is available at GitHub (https://github.com/ChemaMD/OweniaGenome).

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

**Acknowledgements** We thank A. de Mendoza, D. Gavriouchkina and S. Rossiter for their support and valuable comments on the manuscript; staff at Station Biologique de Roscoff for their help with collection and animal supplies; staff at the Oxford Genomics Centre at the Wellcome Centre for Human Genetics (funded by Wellcome Trust grant reference 203141/Z/16/Z) for the generation and initial processing of RNA-seq and ATAC-seq sequencing data; M. J. Boyle for providing the *chordin* sequence for *T. lageniformis*; J. Deane for initial help with Hox gene characterization in *O. fusiformis*; and core technical staff at the Department of Biology at Queen Mary University of London for their constant support. This work used computing resources from Queen Mary University of London's Apocrita HPC facilities. This work was funded by the Horizon 2020 Framework Programme to J.M.M.-D. (European Research Council Starting Grant action number 801669) and A.H. (European Research Council Consolidator Grant action number 648861), a Royal Society University Research Fellowship (URF\R1\191161) and a Japan Society for the Promotion of Science Kakenhi grant (JP 19K06620) to F.M., and a Biotechnology and Biological Sciences Research Council LIDo iCASE PhD studentship (BB/T008709/1) to J.M.M.-D. and B.E.D.

**Author contributions** J.M.M.-D., F.M., Y.L. and F.M.M.-Z. conceived and designed the study. Y.L. collected RNA-seq samples for *O. fusiformis* and *C. teleta*, performed ATAC-seq experiments and contributed to all data analyses. F.M.M.-Z. performed *chordin* orthology studies and contributed to all data analyses. K.G. conducted in situ hybridization analyses of Hox genes. A.M.C.-B. collected RNA-seq samples for *C. teleta*, performed immunostainings on larvae and gene expression analyses of *chordin*. B.E.D. and R.D.D. contributed to computational analyses. Y.T. generated OMNI-C libraries. G.M. performed repeat annotations and analyses. O.S. identified and performed in silico analyses of Hox genes. M.T. performed genomic extractions and optical mapping. K.M. collected *Magelona* spp. A.H. and N.M.L. contributed to sequencing efforts. F.M. and J.M.M.-D. assembled and annotated the genome and contributed to data analyses. Y.L., F.M.M.-Z. and J.M.M.-D. drafted the manuscript, and all authors critically read and commented on the manuscript.

**Competing interests** The authors declare no competing interests.

**Additional information**
**Correspondence and requests for materials** should be addressed to Ferdinand Marlétaz or José M. Martín-Durán.

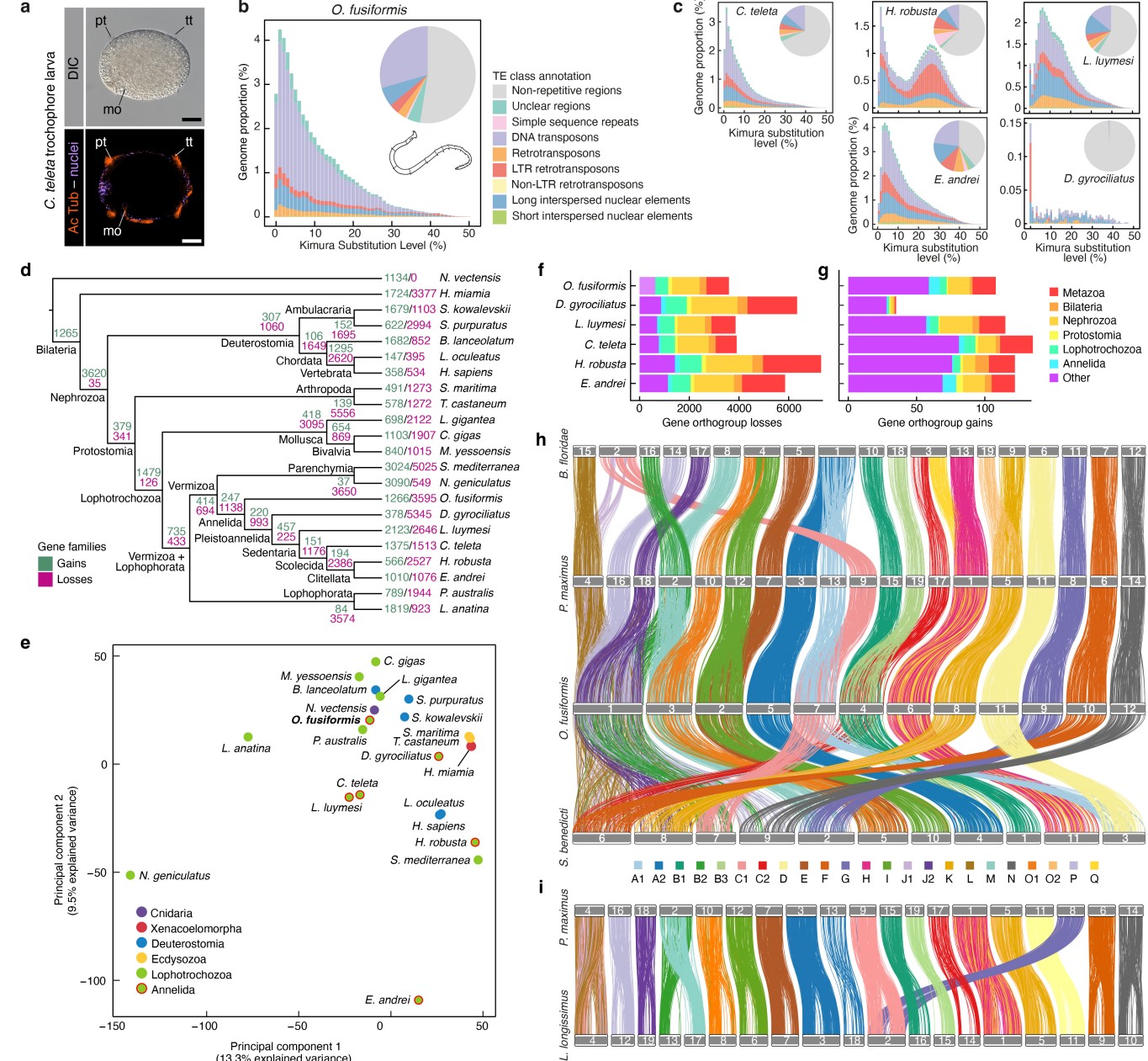

**Extended Data Fig. 1 | The genome of *Owenia fusiformis* is conservatively evolving. a**, Differential interface contrast (DIC) images and z-stack confocal laser scanning views of a *C. teleta* trochophore larva stained for DAPI and acetylated α-tubulin. **b,c**, Pie charts of the transposable element content and Kimura substitution plots of transposable element divergence for *O. fusiformis* and other selected annelid species belonging to different annelid clades as depicted in **c**. Unlike *H. robusta* and *L. luymesi*, which show bursts of transposable elements, *O. fusiformis* shows more steady rates of expansion. **d**, Gene family evolution analysis across 22 metazoan lineages under a consensus tree topology. Gains are shown in green, losses in violet. Gene family losses in *O. fusiformis* are like those of slow-evolving lineages. **e**, Principal component analysis from Fig. 1b, showing the full set of species. **f,g**, *O. fusiformis* has the

lowest number of gene losses of all sampled annelids (**e**), and the least gene expansions (**f**) after the extremely compact genome of *D. gyrociliatus*. **h**, Macrosynteny analysis between *O. fusiformis*, and from top to bottom, the cephalochordate *Branchiostoma floridae*, the bivalve *Pecten maximus*, and the annelid *Streblospio benedicti*. *Owenia fusiformis* retains ancestral linkage groups but also exhibits annelid- and species-specific chromosomal arrangements. However, the karyotype of *O. fusiformis* is more conserved than that of the annelid *S. benedicti*. **i**, Macrosynteny analysis between the bivalve *P. maximus* and the nemertean worm *L. longissimus*. *Lineus longissimus* exhibits conserved ancestral bilaterian linkage groups, including three potential lophotrochozoan-specific chromosomal rearrangements (H+Q, J2+L and K+O2), plus a nemertean-specific fusion (G+C1). Scale bar in **a**, 50 μm.

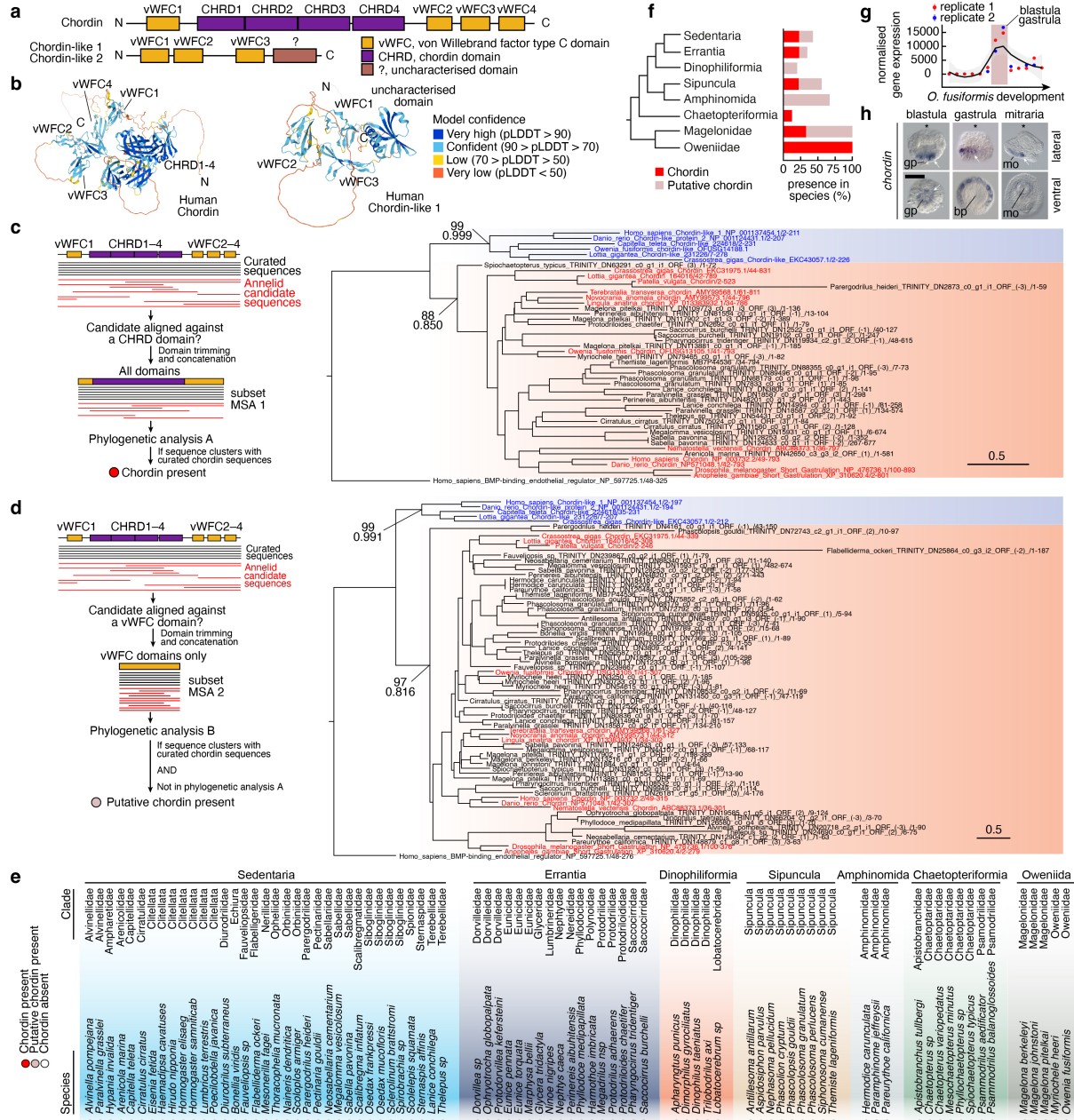

**Extended Data Fig. 2 | *chordin* was lost multiple times in annelids. a**, Domain organisation of Chordin (CHRD) and Chordin-like (CHRDL1/2) proteins, as inferred from human orthologs. **b**, Public AlphaFold protein structure prediction for human Chordin (UniProt: Q9H2X0) and Chordin-like 1 (UniProt: Q9BU40) revealed a previously unknown and uncharacterised domain in CHRDL1 and CHRDL2 (also depicted in **a**). **c**,**d**, Orthology assignment of *chordin* annelid candidates. From the multiple sequence alignment, candidate annelid sequences with a 10-residue or longer fragment aligned against either the CHRD (**c**; i.e., bona fide *chordin* genes) or the vWFC domains (**d**; i.e., putative *chordin* genes) were kept for further analysis. CHRDL cluster is shaded in blue; CHRD cluster, in red. Bootstrap support values (top) and posterior probabilities

(bottom) are shown at both key nodes. Sequences in red and blue are curated CHRD and CHRDL sequences, respectively. **e**,**f**, Summary phylogenetic trees of presence or absence of *chordin* (red) or putative *chordin* (light brown) across Annelida. **g**, RNA-seq expression levels of *chordin* in *O. fusiformis*, which peaks at the blastula and gastrula stages, after the specification and inductive activity of the embryonic organiser. Curve is a locally estimated scatterplot smoothing, coloured shaded area represents standard error of the mean. **h**, Whole mount *in situ* hybridisation of *chordin* at the blastula (5 h post fertilisation, hpf), gastrula (9 hpf), and mitraria larva (27 hpf) stages of *O. fusiformis*. Asterisks mark the animal/anterior pole. gp: gastral plate; bp: blastopore, mo: mouth. Representative results of three independent analyses. Scale bar in **h**, 50 μm.

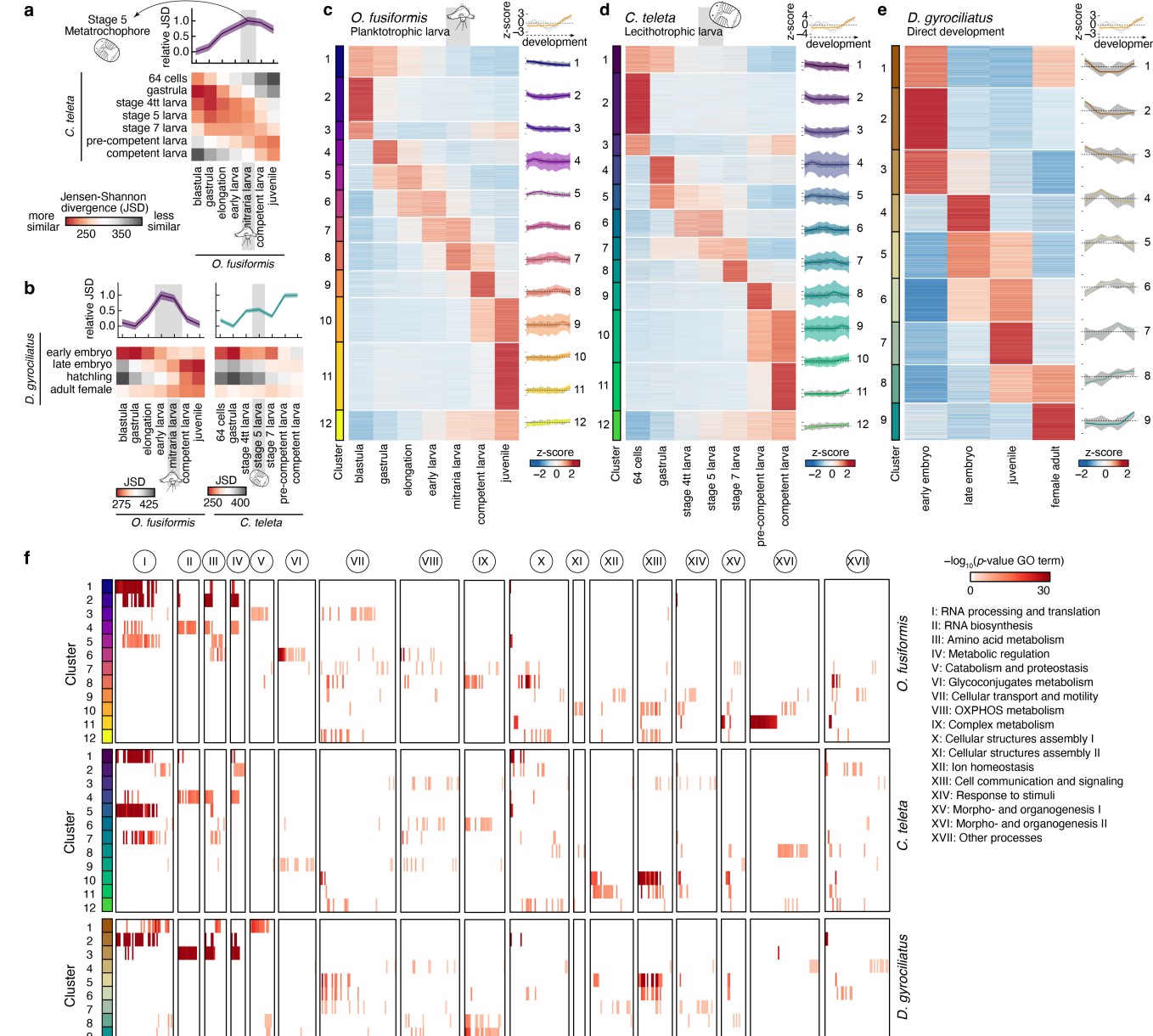

**Extended Data Fig. 3 | Gene expression dynamics during annelid life cycles.**
**a,b**, Heatmaps of average pairwise transcriptomic Jensen–Shannon Divergence
(JSD) between *O. fusiformis* and *C. teleta* (**a**), and between *D. gyrociliatus* and
either *O. fusiformis* (**b**, left) or *C. teleta* (**b**, right). Average relative JSD of the
*C. teleta* or *O. fusiformis* stages of minimal divergence to each corresponding
stage is shown on top. Confidence intervals represent standard deviation from
250 bootstrap resamplings of the ortholog sets. **c**–**e**, Soft *k*-means clustered
heatmap of all transcripts whose expression was not null in at least one
developmental stage into an optimal number of 12 clusters (*O. fusiformis*, **c**; and
*C. teleta*, **d**) and 9 clusters (*D. gyrociliatus*, **e**). Soft clustering considerably
increased temporal resolution for the RNA-seq time course of *D. gyrociliatus*.

On the right of each heatmap, gene-wise expression dynamics (grey lines) and
locally estimated scatterplot smoothing (coloured lines) for each cluster.
Coloured shaded areas represent standard error of the mean. **f**, Enrichment
analysis of biological process gene ontology (GO) terms for RNA-seq clusters.
Each line represents a single GO term, for which the $-\log_{10}(p$-value) for each
RNA-seq cluster is shown in a colour-coded scale. GO terms were clustered into
15 distinct clusters based on semantic similarity (see Supplementary Figs. 7, 8).
Clusters are shown on the bottom of the heatmaps. For the full list of GO terms
and clusters, see Supplementary Figs. 4–6. *P*-values were derived from
upper-tail Fisher's exact tests.

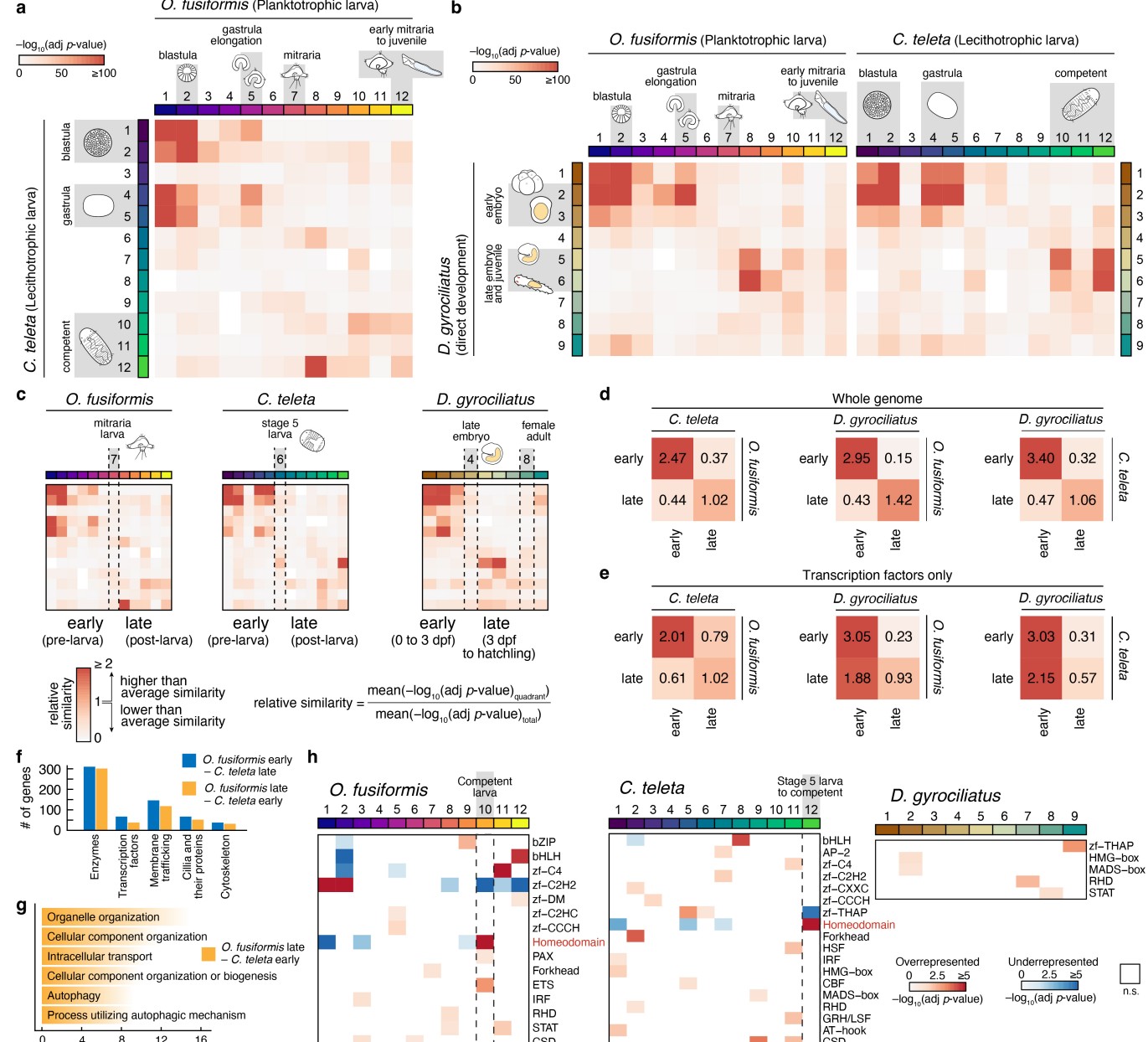

**Extended Data Fig. 4 | Heterochronic shifts in gene regulatory programmes between annelid life cycles. a,b**, Similarity heatmaps showcasing the orthogroup overlap between the clusters of co-regulated genes (see Extended Data Fig. 3c–e), between the three annelids. *P*-values were derived from upper-tail hypergeometric tests and Benjamini-Hochberg-adjusted. **c**, Explanation of the orthogroup overlap analysis by quadrants. Clusters were classed as "early" (before dotted lines) or "late" (after dotted lines). Clusters of the female adult of *D. gyrociliatus* were disregarded. **d,e**, Heatmaps of relative similarity by quadrants of the orthogroup overlap analyses of the whole genomes (**d**) and transcription factors only (**e**). Colour scale in **d** and **e** is the same as in **c**. **f**, KEGGbrite characterisation of the gene sets under heterochronic shifts (surrounded by dotted black lines in Fig. 2e) between *O. fusiformis* and *C. teleta*.

**g**, Bar plots depicting *p*-values of top biological process GO terms of genes shifted from late expression in *O. fusiformis* to early expression in *C. teleta*. *P*-values were derived from upper-tail Fisher's exact tests. Full list is available in Supplementary Fig. 13. **h**, Enrichment analysis of the number of transcription factors per class in clusters of co-transcribed genes of *O. fusiformis* (left), *C. teleta* (centre) and *D. gyrociliatus* (right). For each cluster and class combination, the Bonferroni-adjusted *p*-value from the two-sided Fisher's exact test is shown. Cells in red represent overrepresented classes (odds ratio, OR > 1; adjusted *p*-value < 0.05); cells in blue, underrepresented classes (OR < 1, adjusted *p*-value < 0.05). Dotted lines highlight clusters of maximal enrichment of the homeodomain class. n.s.: not significant.

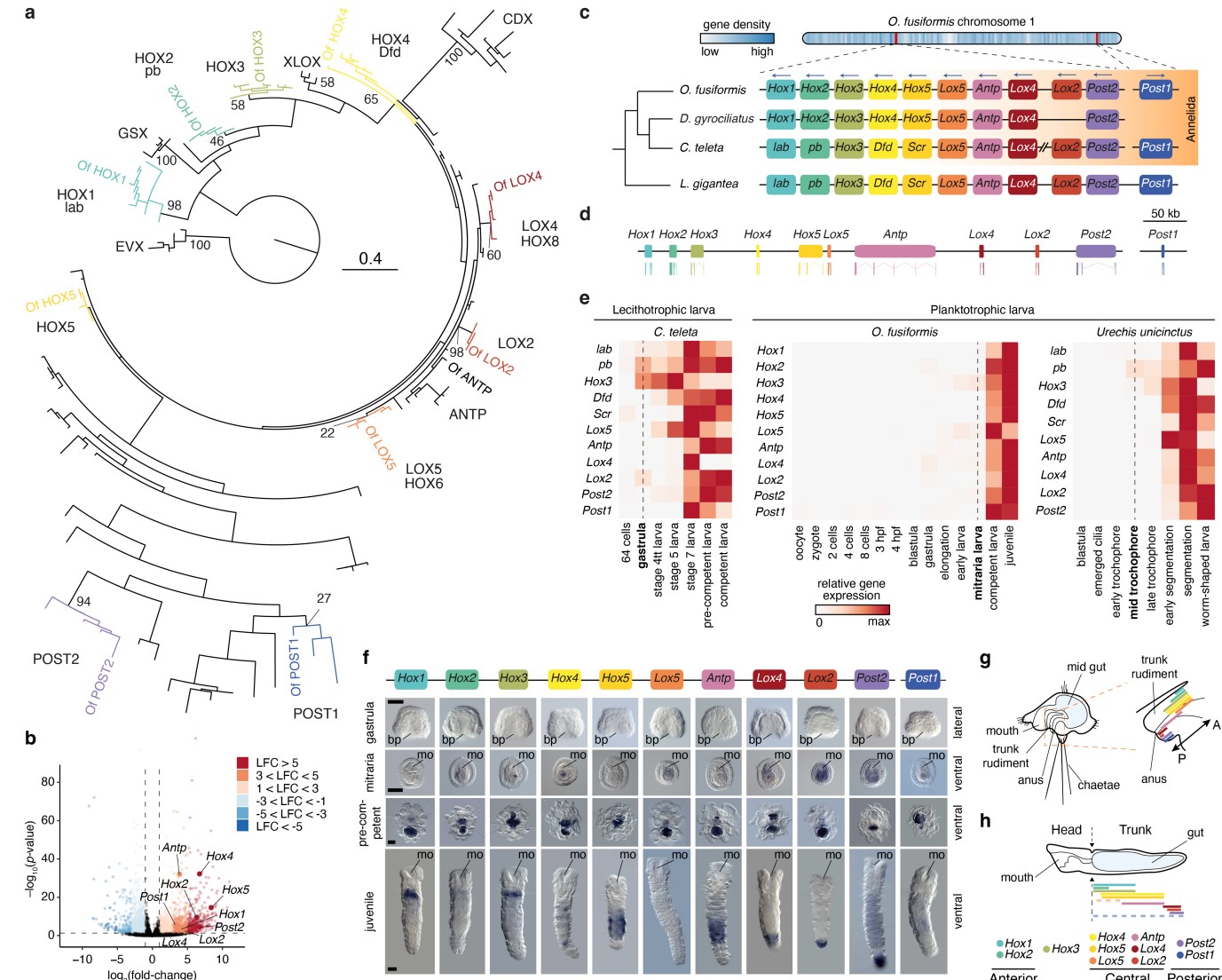

**Extended Data Fig. 5 | The *Hox* gene complement and expression in *O. fusiformis*. a**, Orthology assignment of *O. fusiformis Hox* genes through maximum likelihood phylogenetic inference. Bootstrap support values are shown for major gene groups. Of: *O. fusiformis*. **b**, Volcano plot of the mitraria to competent larva transition, highlighting the marked upregulation of *Hox* genes. LFC: log₂(fold-change). *P*-values were derived from the described DESeq2 pipeline and Benjamini-Hochberg-adjusted. **c**, Chromosomal location of the *Hox* cluster and *Post1* gene in *O. fusiformis* (top) and schematic comparison of *Hox* cluster organisation in annelids and a mollusc (bottom). Arrows denote direction of transcription. **d**, Schematic representation to scale of the genomic loci and intron–exon composition of *Hox* genes in *O. fusiformis*. **e**, Heatmaps of *Hox* gene expression during the development of *C. teleta*, *O. fusiformis* and the echiuran annelid *Urechis unicinctus*. In the two annelid species with planktotrophic larvae, *Hox* genes only become expressed at the larval stage (dotted vertical line), and not during embryogenesis, as observed in *C. teleta*. **f**, Whole mount *in situ* hybridisation of *Hox* genes in the gastrula (lateral views) and in the mitraria larva, pre-competent larva, and juvenile stages of *O. fusiformis* (ventral views). The area encircled by a dotted white line at the pre-competent stage highlights a region of probe trapping from ingested food content. bp: blastopore; mo: mouth. Representative results of three independent analyses. **g,h** Schematic representations of the expression of *Hox* genes in the trunk rudiment of the competent larva (**g**) and juvenile trunk (**h**). A: anterior; P: posterior. Drawings are not to scale, and schematic expression domains are approximate. Scale bars in **f**, 50 μm in gastrulae and larvae, and 100 μm in juvenile.

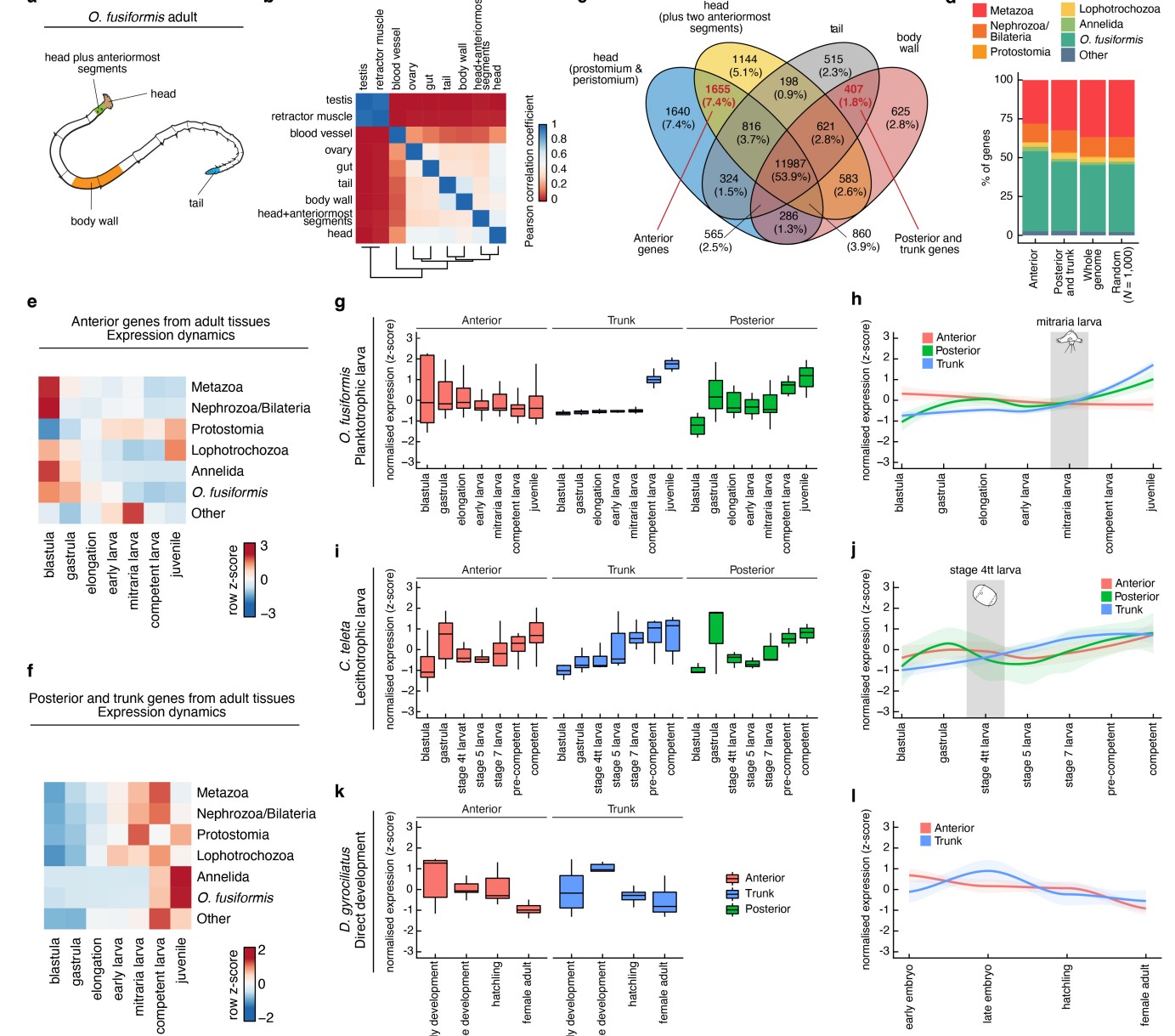

**Extended Data Fig. 6 | Transcriptomic dynamics of anteroposterior genes.**
**a**, Schematic drawing of the adult body regions used to define anterior and posterior and trunk genes. **b**, Correlation matrix of RNA-seq experiments from all nine adult tissues, calculated from a variance stabilising-transformed matrix of the normalised DESeq2 matrix. **c**, Venn diagram showing the number of tissue-specific and shared expressed genes (TPM > 2). Gene sets highlighted with red text were defined as adult anterior, and adult posterior and trunk genes. **d**, Phylostratigraphic classification of adult anterior, and adult posterior and trunk genes, compared to the whole genome and a random subset of 1,000 genes. **e,f**, Expression dynamics of each phylostratum by developmental stage in the adult anterior (**e**), and adult posterior and trunk gene sets (**f**), calculated from the 75 % percentile of a quantile-normalised matrix of gene expression levels. Adult anterior genes of most phylostrata peak at the blastula, while the

maximum expression of adult trunk/posterior genes of most phylostrata peak at post-larval stages. **g–l**, Average expression dynamics of *in situ* hybridisation-validated anterior, trunk, and posterior markers throughout *O. fusiformis* (**g,h**), *C. teleta* (**i,j**), and *D. gyrociliatus* (**k,l**) development. For boxplots in **g,i**, and **k**, centre lines, median; box, interquartile range (IQR); whiskers, first or third quartile ± 1.5 × IQR. Lower whiskers are sometimes not apparent due to the distribution skewness towards zero. Curves in **h,j**, and **l** are locally estimated scatterplot smoothings. Coloured shaded areas represent standard error of the mean. *n* = 23, 8, and 17 anterior markers, 10 and 3 posterior markers, and 15, 10, and 8 trunk markers, for *O. fusiformis*, *C. teleta*, and *D. gyrociliatus*, respectively. Key stages where expression of trunk markers is incipient are shown for both *O. fusiformis* and *C. teleta*.

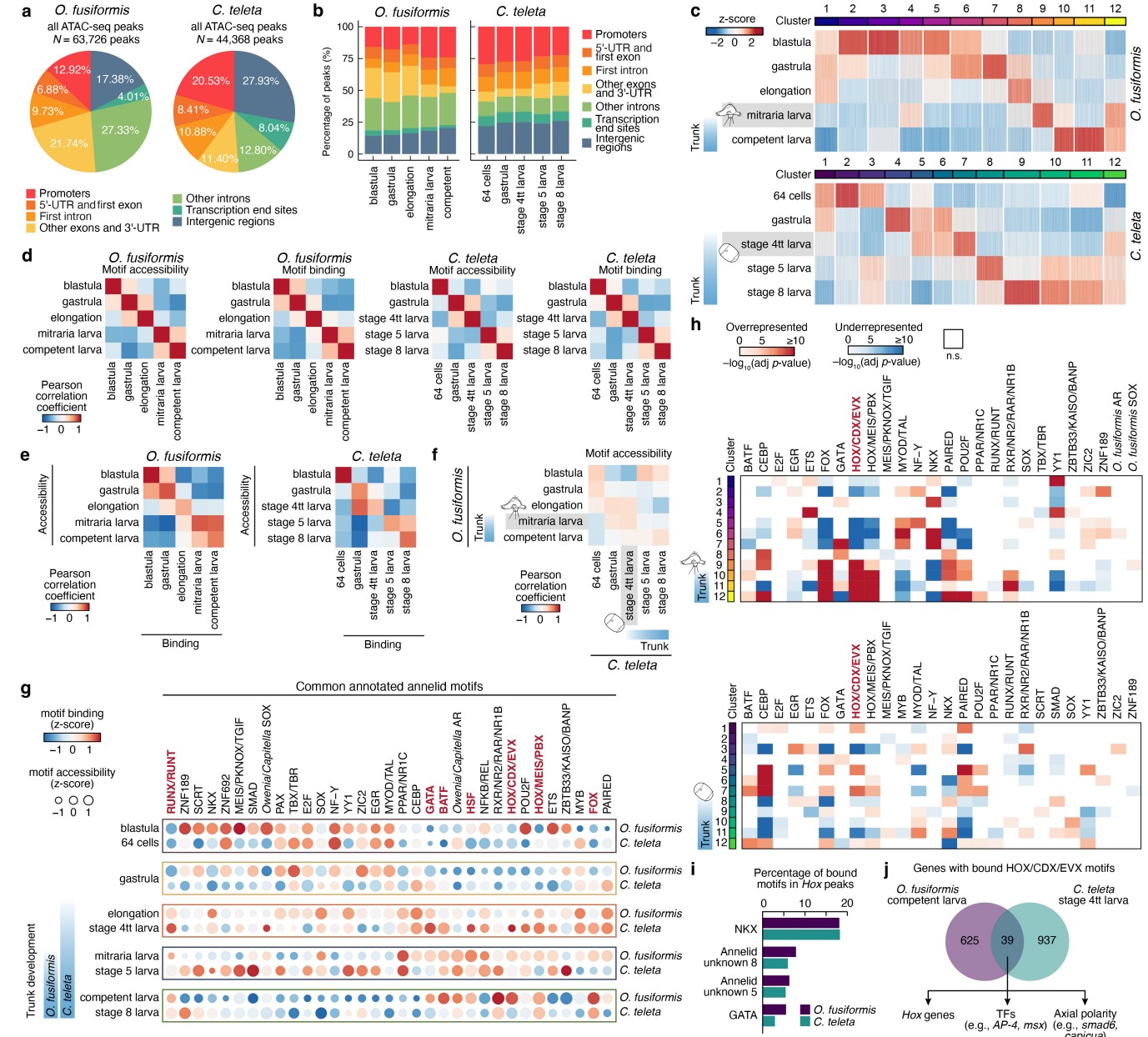

**Extended Data Fig. 7 | Chromatin dynamics during annelid development.**
**a**, Genomic feature annotation of the consensus ATAC-seq peaks. **b**, Stacked bar plots showing the proportion of called peaks per developmental stage classified by genomic feature. **c**, Heatmap of normalised peak accessibility of the soft clustered consensus ATAC-seq peak sets. **d**, Self-correlation matrices of normalised motif accessibility and transcription factor binding score, revealing distinct chromatin regulatory dynamics throughout development. **e**, Correlation matrices of normalised motif accessibility to transcription factor binding score during annelid development. **f**, Correlation matrix of normalised motif accessibility between both species. **d**–**f** further validate the non-triviality of the results obtained in Fig. 3c. Pearson correlation coefficients in **d**–**f** were derived from two-tailed tests. **g**, Heatmap of normalised motif accessibility and transcription factor binding dynamics for each of the common annotated annelid motif archetypes during *O. fusiformis* and *C. teleta*

development. Colour scale denotes transcription factor binding score dynamics, bubble size represents motif accessibility dynamics, both in a z-score scale. Motif archetypes highlighted in red are representative examples of the heterochronic shifts shown in bulk in Fig. 3c. **h**, Enrichment analysis of the number of occurrences of the common annotated annelid motif archetypes in the peak clusters inferred through soft *k*-means clustering and shown in **c**, for *O. fusiformis* (top) and *C. teleta* (bottom). For each cluster and motif combination, the Bonferroni-adjusted *p*–value of the two-tailed Fisher's exact test is shown. Red cells represent significantly overrepresented lineages (odds ratio, OR > 1, adjusted *p*–value < 0.05). Blue cells denote significantly underrepresented lineages (OR < 1, adjusted *p*–value < 0.05). **i**, Most abundant bound motifs in peaks of the *Hox* clusters. **j**, Downstream regulated genes by transcription factors bound to the HOX/CDX/EVX motif archetype.

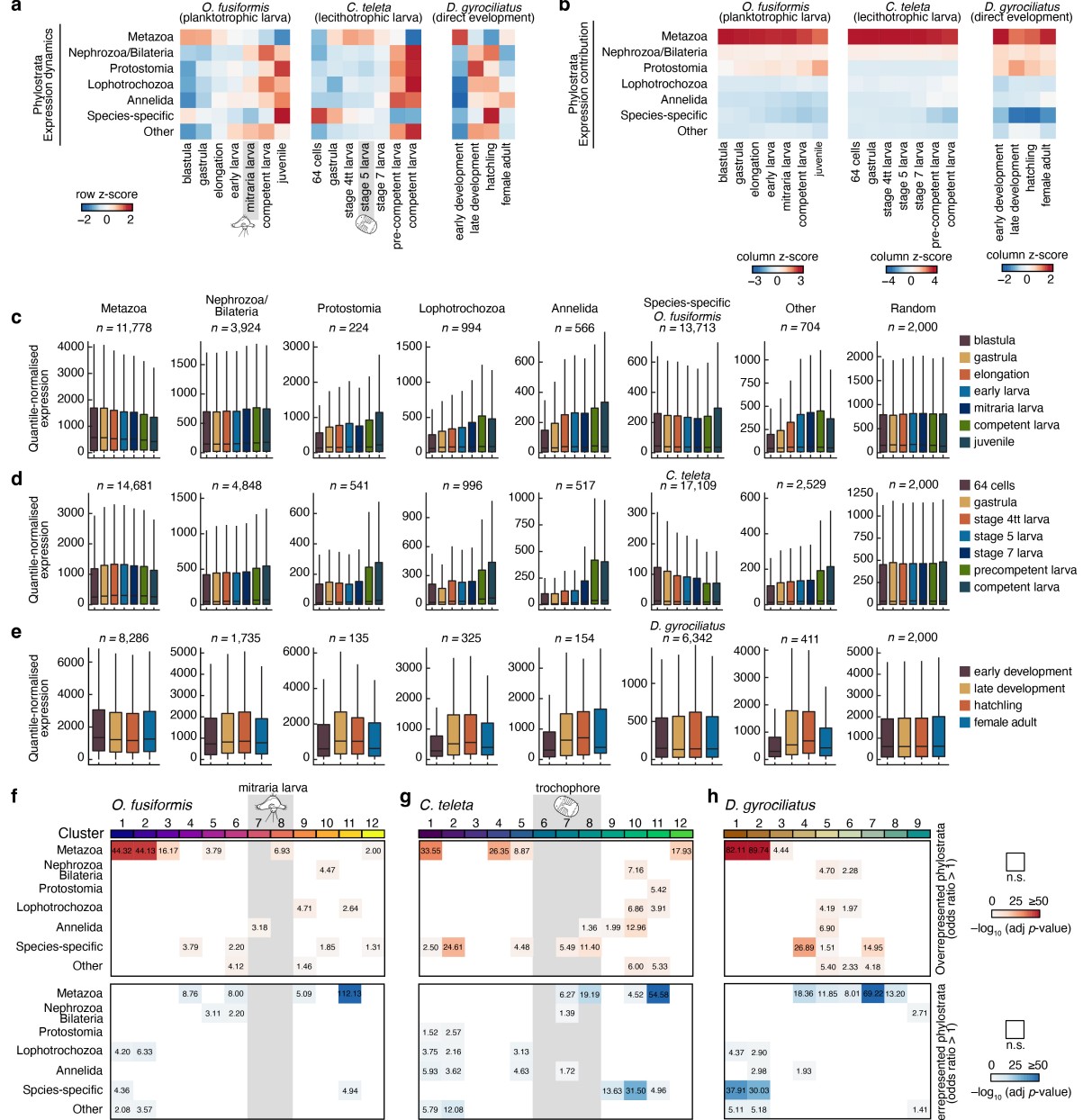

**Extended Data Fig. 8 | Phylostratigraphy analyses in annelid life cycles.**
**a**,**b** Expression dynamics (**a**) and expression contribution (**b**) of each phylostratum by developmental stage in all three annelids, calculated from the 75% percentile of a quantile-normalised matrix of gene expression levels. Older genes are expressed at the highest levels across annelid development. **c**–**e**, Boxplots of quantile-normalised expression levels of genes classified by phylostratum across *O. fusiformis* (**c**), *C. teleta* (**d**), and *D. gyrociliatus* (**e**) development. A random subset of 2,000 genes is shown as a negative control. *n* denotes number of genes per phylostratum. **f**–**h**, Enrichment analysis of the number of genes per phylostratum in clusters of co-transcribed genes as

inferred through soft *k*-means clustering and shown in Extended Data Fig. 3c–e, for *O. fusiformis* (**f**), *C. teleta* (**g**), and *D. gyrociliatus* (**h**). For each cluster and phylostratum combination, the Bonferroni-adjusted *p*–value of the two-tailed Fisher's exact test is shown. Upper tables include significantly overrepresented lineages (odds ratio, OR > 1, adjusted *p*–value < 0.05). Lower tables include significantly underrepresented lineages (OR < 1, adjusted *p*–value < 0.05). Shaded grey areas indicate clusters of genes with peak expression at the mitraria larva, for *O. fusiformis*; and stage 4tt through stage 7 larval stages, for *C. teleta*.

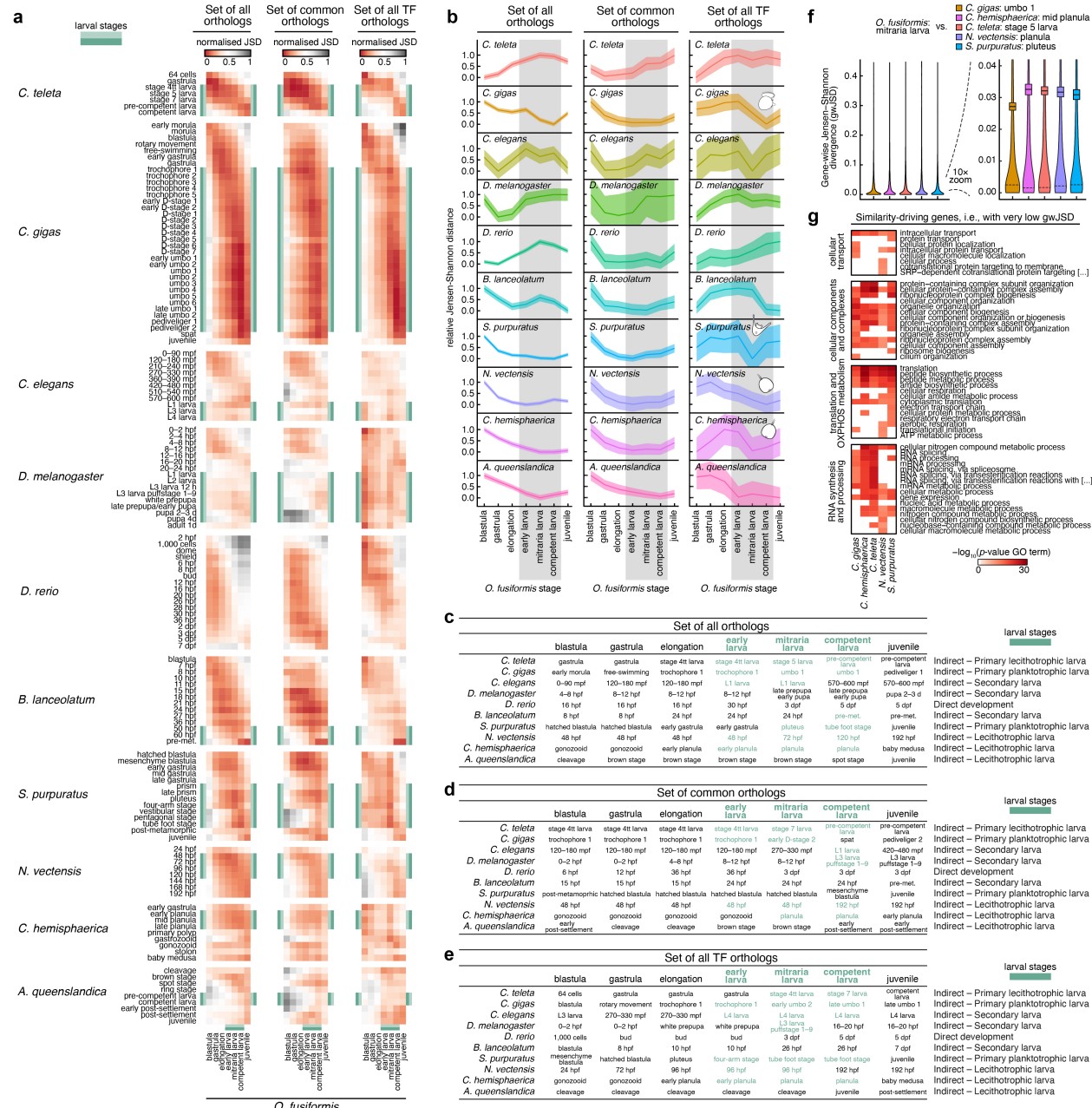

**Extended Data Fig. 9 | Bilaterian planktotrophic larvae and cnidarian larvae share maximal transcriptional similarity. a**, Heatmaps of normalised transcriptomic Jensen–Shannon divergence (JSD) from pairwise comparisons of all single copy one-to-one orthologs (left), the set of common orthologs to all species (centre), and all single copy one-to-one transcription factor orthologs (right), between *O. fusiformis* and ten other metazoan lineages with different life cycles. Larval stages are highlighted in green. **b**, Average relative JSD for the datasets shown in **a**, from stages of minimal JSD to each *O. fusiformis* stage. Confidence intervals represent the standard deviation from 250 bootstrap resamplings of the ortholog sets. **c**–**e**, Stages of minimal JSD to each *O. fusiformis* stage, calculated from the one-to-one ortholog set (**c**), the common ortholog set (**d**), and the one-to-one transcription factor ortholog set (**e**). Larval stages

are highlighted in green. **f**, Violin plots of the gene-wise Jensen Shannon divergence (gwJSD) distributions for the pairwise comparisons of the one-to-one ortholog sets between the mitraria larva of *O. fusiformis* and the stages of minimal transcriptomic divergence as in **c**. for *C. gigas* (*n* = 6,737 single copy orthologs), *C. hemisphaerica* (*n* = 4,691), *C. teleta* (*n* = 7,651), *N. vectensis* (*n* = 5,254), and *S. purpuratus* (*n* = 5,015). Boxes represent mean estimate ± standard deviation. Dotted lines mark the point of highest probability density. Genes below ¼ of this point were subset as similarity-driving genes. **g**, Biological process GO terms enrichment of the five similarity-driving gene sets. GO terms were clustered by semantic similarity into 4 clusters. Each row represents a single GO term, for which the −log₁₀(*p*−value) for each gene set is shown in a colour-coded scale.

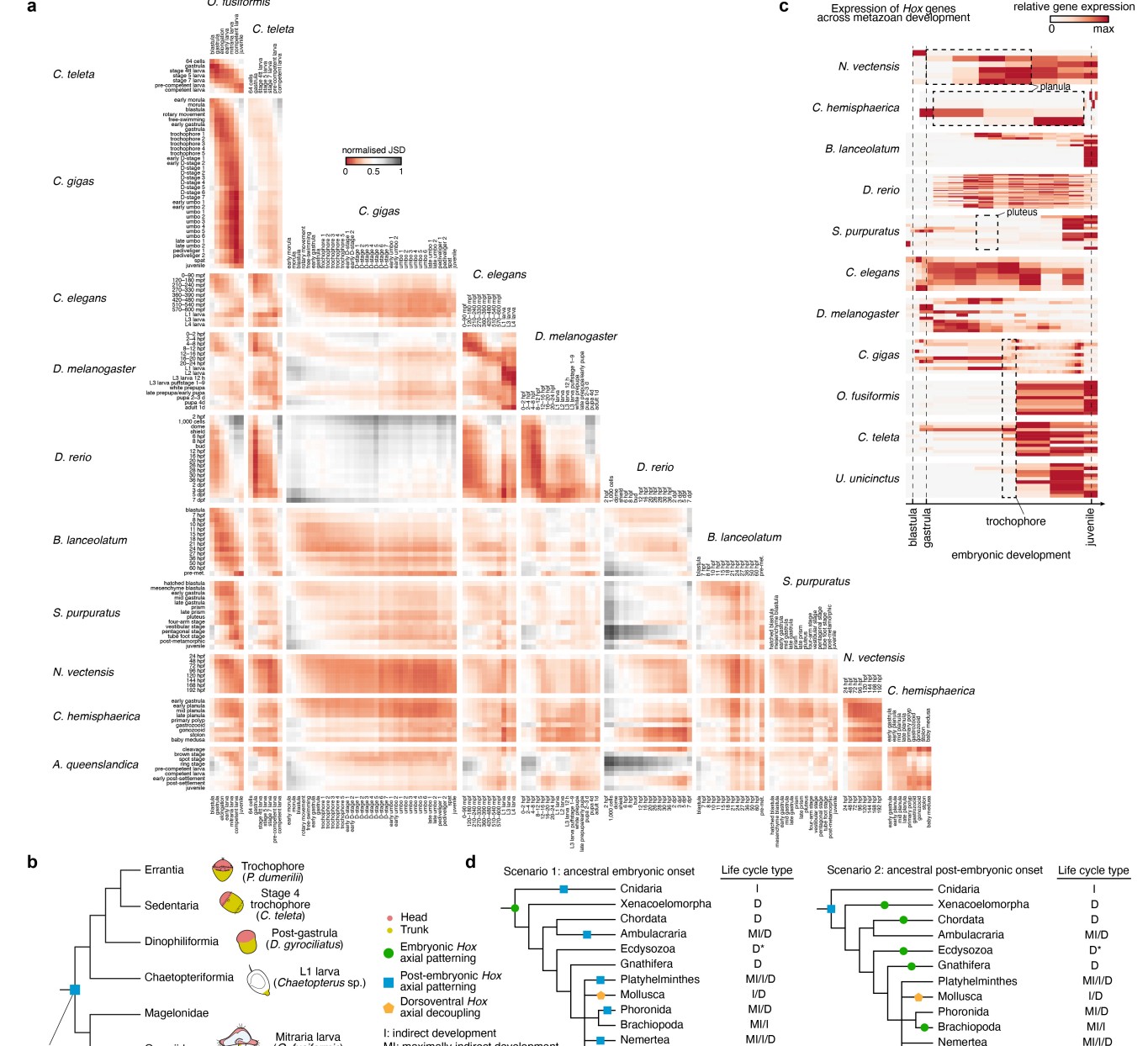

**Extended Data Fig. 10 | Comparative transcriptomic analysis of metazoan life cycles. a**, Matrix of heatmaps of normalised transcriptomic Jensen–Shannon divergence (JSD) from pairwise comparisons of all single copy one-to-one orthologs between all eleven metazoan lineages. From top to bottom and left to right: the annelids *O. fusiformis* and *C. teleta*, the bivalve *C. gigas*, the nematode *C. elegans*, the insect *D. melanogaster*, the vertebrate *D. rerio*, the cephalochordate *B. lanceolatum*, the sea urchin *S. purpuratus*, the cnidarians *N. vectensis* and *C. hemisphaerica*, and the poriferan *A. queenslandica*. **b**, Proposed evolutionary scenario for larval and life cycle evolution in Annelida. Post-embryonic trunk patterning is likely an ancestral condition with the convergent pre-displacement of trunk differentiation to embryogenesis concurring with the evolution of indirect development with feeding larva and direct development. Drawings are not to scale. **c**, Expression dynamics of *Hox* genes across the developmental RNA-seq time courses of all eleven species from **a** and the echiuran annelid

*U. unicinctus*. Heatmaps were vertically aligned at the blastula, gastrula, and juvenile stages for all species. Lophotrochozoan lineages with trochophore larvae were also vertically aligned at the trochophore stage. Dotted lines encompass the larval stages of species with ciliated larvae. See Extended Data Fig. 5e and Supplementary Fig. 33 for the fully labelled and non-deformed heatmaps. **d**, Alternative evolutionary scenarios for the deployment of *Hox* genes (as proxy for trunk patterning and assuming the staggered expression along the directive axis of cnidarians and anteroposterior axis of bilaterians is homologous, which does not necessarily imply homology of the two axes). Given our current understanding of *Hox* gene deployment in cnidarian and bilaterian taxa, a late post-embryonic *Hox* patterning ancestral to Bilateria and Cnidaria, as seen in extant lineages with maximal indirect development, is a more parsimonious scenario (on the right).

# Reporting Summary

## Statistics

For all statistical analyses, confirm that the following items are present in the figure legend, table legend, main text, or Methods section.

| n/a | Confirmed | |
|---|---|---|
| ☐ | ☒ | The exact sample size (*n*) for each experimental group/condition, given as a discrete number and unit of measurement |
| ☐ | ☒ | A statement on whether measurements were taken from distinct samples or whether the same sample was measured repeatedly |
| ☐ | ☒ | The statistical test(s) used AND whether they are one- or two-sided *Only common tests should be described solely by name; describe more complex techniques in the Methods section.* |
| ☐ | ☒ | A description of all covariates tested |
| ☐ | ☒ | A description of any assumptions or corrections, such as tests of normality and adjustment for multiple comparisons |
| ☐ | ☒ | A full description of the statistical parameters including central tendency (e.g. means) or other basic estimates (e.g. regression coefficient) AND variation (e.g. standard deviation) or associated estimates of uncertainty (e.g. confidence intervals) |
| ☐ | ☒ | For null hypothesis testing, the test statistic (e.g. *F*, *t*, *r*) with confidence intervals, effect sizes, degrees of freedom and *P* value noted *Give P values as exact values whenever suitable.* |
| ☐ | ☒ | For Bayesian analysis, information on the choice of priors and Markov chain Monte Carlo settings |
| ☒ | ☐ | For hierarchical and complex designs, identification of the appropriate level for tests and full reporting of outcomes |
| ☐ | ☒ | Estimates of effect sizes (e.g. Cohen's *d*, Pearson's *r*), indicating how they were calculated |

*Our web collection on statistics for biologists contains articles on many of the points above.*

## Software and code

Policy information about availability of computer code

| | |
|---|---|
| Data collection | SRA-Toolkit v2.11.3 |
| Data analysis | Jellyfish v.2.3, FloMax v.2.82, GenomeScope 2.0, CANU v.8.3rc2, bwa mem v.0.7.17, Racon v.1.16, purge_haplotigs v.1.0.4, Merqury v.1.1, BUSCO v.5, RepeatModeler v.2.0.1, RepBase, DIAMOND v.0.9.22, RepeatMasker "open-4.0", LTR_finder v.1.07, RepeatCraft v.0.1.1, SAMtools v.1.9, STAR v. 2.5.3a, trimmomatic v.0.39, StringTie v.1.3.6, Portcullis v.1.1.2, Trinity v.2.5.1, GMAP v.2020-04-08, Mikado v.2.1 pipeline, BLAST v.2.2.31+, Augustus v.3.2.3, Exonerate v.2.4.0, Augustus v.3.2.3, PASA v.2.3.3, Trinotate v.3.0, HMMER v.2.3.2, signalP v.4.1, PANTHER HMM, Juicer pipeline r.e0d1bb7, 3d-dna v.180419, Juicebox v.1.11.08, Liftoff v.1.6.1, AGAT suite of scripts v0.8.1, OrthoFinder v.2.2.7, MMSeqs2, ETE 3 library, kallisto v.0.46.2, ESeq2 v.1.30.1 package, mfuzz v.2.52 package, WGCNA package v.1.70–3, Cytoscape v.3.8.2, topGO v.2.44, simplifyEnrichment v.1.2.0 package, MAFFT v.7, IQ-TREE v.2.0.3, MrBayes v.3.2.7a, BlastKOALA server, gBlocks v.0.91b, RAxML v.8.2.11.9, FigTree v.1.4.4, cutadapt v.2.5, NextGenMap v.0.5.5, deepTools v.3.4.3, MACS2 v.2.2.7.1, BEDtools v.2.28.0, IDR v.2.0.4.2, DiffBind v.3.0.14, UpSetR v.1.4.0, HOMER v.4.1, pyGenomeTracks v.2.1, GimmeMotifs v.0.16.1, TOBIAS v.0.12.0, scikit-learn v1.0.2, philentropy v.0.5.0 package, R version 3.5.1, R version 4.1.2, Python 3.8.10, Visual Studio Code v.1.70.2, Inkscape 1.0.1, Adobe Photoshop 2021 22.4.3. release, Adobe Illustrator 2021 25.4.1 release

All custom code not previously published and key files relevant for the reproducibility of this study are available in our GitHub repository: https://github.com/ChemaMD/OweniaGenome. |

For manuscripts utilizing custom algorithms or software that are central to the research but not yet described in published literature, software must be made available to editors and reviewers. We strongly encourage code deposition in a community repository (e.g. GitHub). See the Nature Portfolio guidelines for submitting code & software for further information.

## Data

Policy information about availability of data

All manuscripts must include a data availability statement. This statement should provide the following information, where applicable:
- Accession codes, unique identifiers, or web links for publicly available datasets
- A description of any restrictions on data availability
- For clinical datasets or third party data, please ensure that the statement adheres to our policy

Accession codes and unique identifiers to previously publicly available datasets we used for this study are listed in Supplementary Table 2 (genome files used in gene family evolution analyses), Supplementary Table 8 (transcriptomes used in the evolutionary analysis of chordin in annelids), Supplementary Tables 41 and 43 (gene identifiers used in pathway analyses), Supplementary Table 47 (sequence identifiers used in Hox genes orthology assignment), Supplementary Table 48 (RNA-seq datasets used for Hox gene expression profiling in U. unicinctus) and Supplementary Table 88 (RNA-seq datasets used for comparative annelid and metazoan transcriptomics and Hox gene expression profiling). Repetitive elements database RepBase can be accessed at https://www.girinst.org/repbase/. Transcription factor public database TFClass can be found at http://tfclass.bioinf.med.uni-goettingen.de/. All sequence data associated with this project are available at the European Nucleotide Archive (project PRJEB38497) and Gene Expression Omnibus (accession numbers GSE184126, GSE202283, GSE192478, GSE210813 and GSE210814). Genome assemblies, transposable element annotations, genome annotation files used for RNA-seq and ATAC-seq analyses, WGCNA nodes and edges files, alignment files used in orthology assignment, and other additional files are publicly available in https://github.com/ChemaMD/OweniaGenome.

# Field-specific reporting

Please select the one below that is the best fit for your research. If you are not sure, read the appropriate sections before making your selection.

☒ Life sciences  ☐ Behavioural & social sciences  ☐ Ecological, evolutionary & environmental sciences

For a reference copy of the document with all sections, see nature.com/documents/nr-reporting-summary-flat.pdf

# Life sciences study design

All studies must disclose on these points even when the disclosure is negative.

| | |
|---|---|
| Sample size | Sample sizes for genomic and transcriptomic analyses were estimated based on the amount of genomic DNA and total RNA obtained per individual. For ATAC-seq analyses, sample size per library was that such that there was a final number of 50,000 cells for subsequent tagmentation. |
| Data exclusions | No data was excluded. |
| Replication | Two biological replicates were collected for RNA-seq and ATAC-seq datasets, which is a commonly accepted standard in the field. A high correlation between biological replicates was observed. Other experimental techniques (e.g. in situ hybridisation, immunohistochemistry) were performed at least three times to verify observed results. |
| Randomization | All Capitella teleta animal cultures were set up from randomly selected late larval stages from distinct larval broods to ensure genetic variability in subsequent generations. Embryos and larvae for experiments were collected from either spontaneous broods collected during weekly siftings, or from mating dishes specifically set up for embryonic and larval stages collection between a randomly selected male and a randomly selected female. All Owenia fusiformis animal collections were also performed randomly for in vitro fertilisations to ensure genetic variability in the progeny. Unlabelled and unidentified animals were randomly dissected to obtain either oocytes or sperm. |
| Blinding | All animal collections were allocated blindly to any of the replicates of study. Investigators did not know during the data analysis stage about the origin of each biological replicate for transcriptomic and epigenomic studies. |

# Reporting for specific materials, systems and methods

We require information from authors about some types of materials, experimental systems and methods used in many studies. Here, indicate whether each material, system or method listed is relevant to your study. If you are not sure if a list item applies to your research, read the appropriate section before selecting a response.

## Materials & experimental systems

| n/a | Involved in the study |
|---|---|
| ☐ | ☒ Antibodies |
| ☒ | ☐ Eukaryotic cell lines |
| ☒ | ☐ Palaeontology and archaeology |
| ☐ | ☒ Animals and other organisms |
| ☒ | ☐ Human research participants |
| ☒ | ☐ Clinical data |
| ☒ | ☐ Dual use research of concern |

## Methods

| n/a | Involved in the study |
|---|---|
| ☒ | ☐ ChIP-seq |
| ☐ | ☒ Flow cytometry |
| ☒ | ☐ MRI-based neuroimaging |

# Antibodies

| | |
|---|---|
| Antibodies used | Mouse anti-acetyl-alpha tubulin Antibody, clone 6-11B-1, 1:800 dilution (Sigma-Aldrich Cat# MABT868, RRID:AB_2819178), Goat anti-Mouse IgG (H+L) Cross-Adsorbed Secondary Antibody, Alexa Fluor 647, 1:800 dilution (Thermo Fisher Scientific Cat# A-21235, RRID:AB_2535804) |
| Validation | Antibody cross-reactivity against Owenia fusiformis was predicted based on multiple sequence alignments (MSA) of targeted antigens with closely phylogenetically related species (e.g. Capitella teleta, Platynereis dumerilii, Owenia collaris, etc.) for which there was published literature using those antigens. Antibodies were then validated in immunohistochemistry with our own species. |

# Animals and other organisms

Policy information about studies involving animals; ARRIVE guidelines recommended for reporting animal research

| | |
|---|---|
| Laboratory animals | Capitella teleta Blake, Grassle & Eckelbarger, 2009: we kept year round individuals from both sexes and all ages. Adult specimens were only kept until they were 18 weeks old. In this study we only studied embryonic and larval stages. |
| Wild animals | This study did not involve wild animals. |
| Field-collected samples | Owenia fusiformis Delle Chiaje, 1844: sexually mature individuals were collected from subtidal waters near the Station Biologique de Roscoff, sent through post to our home institution, and cultured in the lab as described before (see Methods). Animals were kept until the end of their lives or until they were used for experiments, i.e. spawnings for in vitro fertilisations.<br>Magelona spp.: were collected in muddy sand from the intertidal of Berwick-upon-Tweed, Northumberland, NE England (~55.766781, -1.984587) and kept initially in aquaria at the National Museum Cardiff before their transfer to Queen Mary University of London, where they were kept in aquaria with artificial sea water. Magelona spp. were killed for spawnings for in vitro fertilisations too, in order to get larvae for immunohistochemistry. |
| Ethics oversight | Work on annelids and their embryos are not subject of ethical approvals or restrictions in the United Kingdom. |

Note that full information on the approval of the study protocol must also be provided in the manuscript.

# Flow Cytometry

## Plots

Confirm that:

☒ The axis labels state the marker and fluorochrome used (e.g. CD4-FITC).

☒ The axis scales are clearly visible. Include numbers along axes only for bottom left plot of group (a 'group' is an analysis of identical markers).

☒ All plots are contour plots with outliers or pseudocolor plots.

☒ A numerical value for number of cells or percentage (with statistics) is provided.

## Methodology

| | |
|---|---|
| Sample preparation | Owenia adults were removed from their tubes before flow cytometry analysis. Worms were transferred into a petri dish, washed well in seawater to remove any contaminant, and finely chopped with a razor blade in 2 ml of General-Purpose Buffer to generate a suspension of nuclei. This suspension was filtered through a 30 μm nylon mesh and stained with propidium iodide (Sigma; 1 mg/mL) on ice. |
| Instrument | We used a flow cytometer Partec CyFlow Space fitted with a Cobalt Samba green laser (532nm, 100mW) |
| Software | We used the built-in instrument software FloMax v.2.82. |
| Cell population abundance | We used flow cytometry to estimate genome size using D. melanogaster as reference and thus we did not sort any cell populations. To estimate genome size from propidium iodide staining, we did three independent runs for each species analysing at least 1,000 nuclei per run. |
| Gating strategy | We considered all cell populations for genome size estimation, and thus no gating strategy was implemented. |

☒ Tick this box to confirm that a figure exemplifying the gating strategy is provided in the Supplementary Information.

