## [Peer Review File · Nature]

Manuscript Title: Annelid functional genomics reveal the origins of bilaterian life cycles

Reviewer Comments & Author Rebuttals

Reviewer Reports on the Initial Version:

Referees' comments:

Referee #1 (Remarks to the Author):

In this manuscript, Liang and Martin-Zamora et al. study the genome and developmental transcriptome of the annelid *Owenia fusiformis* to make inferences about the evolution of larvae across annelids, and also across the entire bilateral clade. First, they generated a chromosome-scale genome assembly, which has high contiguity and high completeness in terms of protein-coding gene content. Bulk transcriptome data across development revealed two major phases of change in gene expression, mirrored in the ATAC-seq data, and sets of co-regulated genes. Because only one cluster contains genes with expression both in larvae and in juveniles, the authors suggest that this means larvae could not have evolved via co-option of “adult” genetic modules, as proposed in the intercalation hypothesis. The authors then compared the *O. fusiformis* data to newly generated developmental transcriptomic data from another annelid, *Capitella telata*, and to published datasets from other bilaterians and a cnidarian. This analysis showed increasing similarity over developmental time, particularly at juvenile/adult stages, which challenges the “terminal addition” hypothesis that proposes that larvae are homologous and adult body plans evolved via addition of adult-specific genetic modules. Based on the authors’ studies of Hox gene expression, which appear late in development in *O. fusiformis*, and on published similar data from other species, the authors suggest that heterochronic shifts in trunk patterning modules underlie the evolution of larvae across bilaterians.

Overall, the genome, transcriptome, and ATAC-seq data generated for this manuscript are well-executed and the analyses are by-and-large sound (with a few minor exceptions). These will serve as valuable resources to the evo-devo community. The authors’ efforts in trying to glean biological insight from the genome are commendable. However, many of the claims in the paper are not supported by the data, as the authors offer interpretations but don’t consider alternative explanations. I suspect this is because the study is simply not designed to offer rigorous testing of different hypotheses. The paper also relies heavily on the framework of two existing hypotheses about larval evolution, which were postulated long before we knew much about gene regulation during development and evolution, which makes them pretty easy targets to take down. The central claim, that heterochronic shifts in trunk patterning modules may have shaped larval evolution, is reasonable, but doesn’t rely on any of the genome-scale work that is the focus of the paper. The evolution of larvae and of bilaterian life histories is an important question, however meaningful and field-advancing statements about these processes will need more work, particularly functional studies.

Major Comments

Failure to consider null hypotheses and alternatives, and unsupported claims

There's a considerable list of claims that are not substantiated by the data:

1) The authors should take a step back and clarify what their null expectations from the bulk transcriptome data might be. Given we know that the *O. fusiformis* larva is unique, and across annelids larvae can look different but produce adults that are recognizably "annelid", an a priori expectation would be that larvae might have very distinct transcriptomes. I think this is what underlies the "biphasic" pattern in Figure 2b. PC 2 is saying that the larva is quite different, as we might expect.

2) The statement that the "*O. fusiformis* larva does not co-opt genes expressed in the adult" cannot be made based on these data. There is nothing about the intercalation hypothesis that specifies how many genes or what proportion of genes one would expect to be shared between larvae and adults. The authors report 1,426 genes that are indeed expressed in both larvae and adults. How do we know that there aren't key, important genes in this list that could be causal to the formation of larvae? I don't necessarily believe that the intercalation hypothesis is correct, but the authors are not providing compelling data to disprove it.

3) In line 189, the authors state "our findings support that axial and body patterning, as well as anterior and gut differentiation contribute to the first transcriptional phase, while adult trunk formation largely drives the second post-embryonic transcriptional phase during *O. fusiformis* life cycle." The gut differentiation claim comes from the GO analysis mentioned in line 158, but the data in this paper do not allow any quantitative statements about anterior patterning or trunk formation. Yes, these genes turn on at specific times, but how large or small an effect they are driving can only be determined either by quantitatively assessing the significance of their contribution relative to that of other genes, or by doing functional studies.

4) line 241 states "older rather than younger genes contribute to the development of the mitraria larva, suggesting that the increased use of novel genes in other lophotrochozoan larvae^{7,42} might be due to lineage-specific traits found on those larvae,". There's an alternative interpretation here. Given that the *O. fusiformis* larva is a "head larva" to an extreme, and anterior patterning genes are highly conserved, it may be that this aspect of the biology is driving the observation that older genes are dominating the mitraria. So it is less about other larvae having lineage specific traits, but about *Owenia* having expanded its pre-oral domain.

5) That adult body plans in bilaterians share a ground plan and patterning genes has been stated by others (e.g. Raff 2008, Phil. Trans. R. Soc. B 363, 1473–1479). The finding in this paper that at a large transcriptome-wide scale (focusing on single copy genes) there is similarity among bilaterian adults could be explained by the fact that adult bodies operate on many similar cell biological functions - digestion, excretion, muscle contraction, synaptic transmission, and so on. Thus, given what we know about animal biology in 2022, this could be our null expectation. I am not a proponent of the terminal addition scenario, but technically, the terminal addition scenario does not specify how many genes would one need to observe being different across adults of different animal species for the hypothesis to be correct. The terminal addition hypothesis also does not say that homologous

larvae don't evolve and diverge across species, so how can we specify what level of transcriptome similarity we should expect? The data in this paper cannot rule out that a few key genes were the ones needed to activate genetic modules for making adult bodies work, and this occurred differently in different species. Additionally, the apparent similarity between adult stages could just be a byproduct of the fact that species with extremely divergent larvae are being compared. I don't think there is any reason, a priori, to expect large scale similarity between drosophila, nematode, and annelid larvae, it is like comparing apples to oranges.

6) The authors use orthologous single-copy genes for the analysis in Figure 4. This analysis does not accommodate species-specific genes, so it cannot assess whether novel genetic modules were needed to make adult body plans, which could be an important aspect for the terminal addition scenario. This consideration is not mentioned in the paper.

7) The maximal similarity between *O. fusiformis* larva and other plaktotrophic larvae could simply be driven by convergently-evolved functional properties, such as having to make cilia and to swim around to eat. So the pattern of similarity and differences in transcriptome profiles across species does not directly enable an assessment of homology. Furthermore, the heterochronic shifts in trunk patterning across annelid larvae could easily be driven by key upstream regulators that are different between these species. Thus the observations in this paper do not clearly support homology or convergence of annelid (or bilaterian) larvae. The authors do admit this possibility, so I recommend that they lead with the idea of being open to either scenario rather than showing preference to one (e.g. the claim "our study supports that the late deployment of trunk differentiation programmes is likely ancestral to Annelida").

8) The authors also should consider homology with all of the complexity that is associated with this term. "larvae, which would then be homologous on the grounds of being largely anteriorly derived transitory structures" could mean that there is process homology in terms of anterior genetic modules being used across larvae; this is not the same as larvae being homologous. But stating that larvae themselves are homologous is saying that the ancestor of all bilaterians had a larval stage that became modified in descendant taxa. It is unclear what the authors mean by homology here.

ATAC-seq analysis and interpretations

The biphasic nature of ATAC-seq peaks and specifically the finding that the largest changes in peak accessibility occur in the larval stages is completely consistent with what was observed in the transcriptome data, and would be the null expectation for this work. Given that transcriptionally the larvae are the most different, corresponding differences in chromatin dynamics would be expected to underlie that. Other than yielding what is expected, and providing a dataset for future work, the ATACseq data are not adding to the biological inferences in this manuscript.

The authors should explain if peak widening has been observed in other species/contexts and whether it is known to have any particular biological relevance. Additionally, the specificity of the TF motifs needs to be explained if the authors want to claim biological functions. For example, stating that "Consistent with our transcriptomic dataset, motifs related to transcription factors involved in patterning anterior territories (e.g., PAX2/5/8 and PAX4/6/27), muscle and gut development (GATA28, FOXC28) and early neurogenesis (ATOH41) are amongst the most differentially accessible

in open chromatin regions during embryogenesis, whilst ciliary band genes (OTX28), trunk related genes (e.g., NKX2.127) and most notably Hox genes appear in regulatory regions during larval competence and trunk patterning” requires that we know that the binding sites annotated as specific to FOXC or NKX2.1 in *O. fusiformis* are indeed likely to bind the specific orthologs that have roles in anterior and trunk patterning respectively. Usually, annotating binding sites in new model systems using vertebrate databases will yield specific names of motifs, but this does not mean that the sites will be regulated by the particular TF. For example, sites annotated could be bound by any number of other Fox TFs, most of which are NOT specific to the biological functions that FoxC performs. NKX2.1 sites could be bound by other Nkx homologs, which aren’t trunk patterning genes. Thus, these claims are a big leap. The authors have to either show functional associations with specific orthologs and sites.

Central claim

The Hox expression data convincingly show that trunk patterning genes are expressed late in development, lending support to the idea of a heterochronic shift in *O. fusiformis*. This inference does not rest on the genome scale work that the majority of the manuscript focuses on, and this shift was already known from another annelid, *Urechis unicinctus*. So the data in this paper add to the story, but don’t on their own, introduce a wholly new idea.

Minor comments:

line 168: “in line with the vastly enlarged pre-oral region of the mitraria that forms early on in the adult head” - I am not sure what it means for the pre-oral region of the mitraria to form in the adult head. Do you mean this region makes the adult head?

line 172: “Hox genes, a conserved family of transcription factors involved in anterior-posterior trunk regionalisation in Bilateria³⁶, are among the most upregulated genes at these stages (Fig. 2g).” - it was not easy to find the ranks of all of the Hox genes in the list of DE genes. Only a small subset of these genes are highlighted in the volcano plot in Figure 2.

line 180: “*O. fusiformis* does not express Hox genes during embryogenesis” - the levels look low in heat maps, but were in situ hybridizations done in stages earlier than mitraria to confirm this? Transcription factors can have low but biologically meaningful expression despite overall low levels in transcriptome data.

Figure 2 legend: Given that Figure 2f is a schematic, the legend should make clear where the in situ hybridization data for these genes can be found. It should clarify “based on references XYZ”.

Referee #2 (Remarks to the Author):

The authors introduce the genome of the annelid *Owenia fusiformis*, as well as a comprehensive catalog of gene expression and chromatin accessibility profiles at different developmental time

points. This is a high-quality genome, which will constitute a must-use in any future comparative genomics efforts and one that immediately places *Owenia* at the top of annelid model systems. I want to congratulate the authors for the effort.

The functional genomics datasets are also high quality and I have no concerns regarding data generation and analysis. These datasets are presented in the context of a discussion about the evolution of larval stages, proposing an intermediate solution to the intercalation (independent larval origins) versus terminal addition (ancient larval form, parallel evolution of adult forms). This represents my main concern with the manuscript: I find this interpretation confusing and even contradictory at times.

For example, I don't think the observed temporal gene expression dynamics is incompatible with an intercalation scenario. First, maybe the temporal sampling hasn't enough resolution to detect "co-opted genes" from adult to larval development, or simply these genes are few and not recovered in the gene module analysis (looking at extended data fig.4a it seems there are a number of such genes). The situation is similar with the ATAC-seq analysis. In fact, this data does suggest that many motifs are shared between *Capitella* larva and adult (Fig 3e). Overall, it does seem that adult and larval stage share features, both at the gene expression as well as the accessibility level, and they are the result of a continuous gradient, rather than distinctive phases.

Moving to cross-species analysis, the comparison between *Capitella* and *Owenia* seems to support the intercalation scenario, showing maximal divergence at the larval stages (line 258). But further comparison with other species points at opposite patterns, suggesting that these comparisons are confounded by differences in the cellular composition, tissue heterochronies, etc.

In summary, I do not think the temporal gene expression and accessibility data provide much insight into the question of larval homology and the origin of bilaterian life cycles. In any case, the authors should adequately problematize the data in the context of the major intrinsic limitation of such bulk analyses: the presence of heterochronies and the absence of spatial/tissue resolution make it very difficult to interpret temporal expression/accessibility in terms of high or low similarity between adult and larval stages, as well as between species.

In contrast, I do find the results of the Hox expression pattern analysis very interesting and the interpretation of these results convincing, both in supporting the homology of adult bilaterian bodyplans, as well as in suggesting an important role for heterochronies in the evolution of larval structures.

Minor points

- I'm surprised the authors don't use the adult tissue transcriptomes in any of the analyses.
- The authors should provide the code for reproducing the analyses presented in the manuscript.

Referee #3 (Remarks to the Author):

Liang et al. have undertaken a thorough analysis of the *Owenia fusiformis* genome, and its developmental regulation and expression. *O. fusiformis* is a member of a basal-branching annelid

lineage. Overall, the results and analyses are impressive and high-quality, and the data appear to be robust. Together, they contribute to our understanding of the biology and evolution of annelids and lophotrochozoans, recognising the important caveat that *O. fusiformis* possesses a derived larval type restricted to members of family Oweniidae (the mitraria). This is in contrast to the trochophore larva that is present in most annelids and multiple closely-related phyla, including molluscs.

Despite having a derived larva and some odd developmental features – some oweniids were thought to undergo deuterostomous development, which was shown to be wrong by the corresponding author and colleagues (Martin-Duran et al. 2016 *Nature Ecol Evol*) - the analysis *O. fusiformis* development provides some important insights (e.g. Hox expression and gene regulation).

However, these results do not necessarily contribute to the main proposition of the paper about bilaterian body plan evolution. The authors posit that changes in the timing of trunk formation in annelids, which occurs by the addition of new segments at the posterior end by teloblast growth, sheds light on the evolution of larvae and bilaterian life cycles. This study is built on the premise that analysis of one clade of bilaterians can address long-standing, conflicting hypotheses about the origin of bilaterian larvae and body plans – “intercalation” (larvae are an elaboration of embryonic development of a direct developer, promoting dispersal and habitat selection) vs “terminal addition” (sexually competent larval-like adults are ancestral to which a second life cycle phase was added that then became the sexually-competent adult).

The authors state that in (1) the “intercalation” scenario the larval body plan evolves by co-opting adult gene regulatory networks that lead to adult-enriched gene expression profiles and (2) “terminal addition” scenario where that adult evolved secondarily by incorporating new genetic programmes. This statement sets-up the comparative analyses they undertake and is used to support their argument, which is the main premise of this manuscript. This construct however does not reflect current thinking about how gene regulatory networks and gene pleiotropy might manifest in these two scenarios. It is highly likely not to be as black-and-white as set-out in the manuscript. There is not a single way in which gene expression can manifest under these two scenarios. For instance, why is gene co-option invoked in one scenario and not the other? The addition of life cycle stages does not require the invention of new genes or gene networks (e.g. complex life cycles of parasitic platyhelminths or triphasic cnidarians with a medusa). The final paragraph of the paper recognising some of these caveats and thus undermines the proposition framework the authors established in the introduction and that runs through the manuscript.

As mentioned above, this is an impressive characterisation of an annelid genome and its developmental regulation. There are some interesting correlations between planktotrophy and lecithotrophy in relation to the expression of the Hox code. It would be interesting to see if this stood up after comparisons if other phyla, including molluscs. However, the argument that this study provides critical insights into the evolution of body plans is less compelling. The author’s case is not clearly articulated and the choice of taxon is questionable. It seems to answer such questions one needs to look at a deeper nodes. Cnidarians and even sponges have biphasic life cycles with ciliated larval forms suggesting that both “intercalation” and “terminal addition” theories have little relevance when trying to understand the evolution of bilaterian larval and adult body plans. Put simply, different selective forces at different stages of the life cycle – coupled with developmental

biases and contingencies – is enough to explain modern developmental and body plan diversity.

Referee #4 (Remarks to the Author):

A. Summary of the key results

This paper presents the chromosome-scale genome of *Owenia fusiformis* along with transcriptomic and epigenomic profiling of various life history stages of the species. By comparing with 22 other animal genomes they show that *Owenia* has fewer gene family gains and losses, and retains more ancestral metazoan orthogroups than found in other annelid genomes known to date. The analysis of transcriptomes and regulatory genome also allowed them to assess gene regulatory events for the formation and development of the trochophore larva.

Owenia is part of a clade with low species richness that is the sister group to the rest of Annelida. *Owenia* also has unusual larvae among annelids and does not have a 'classical' trochophore stage. *Owenia*'s phylogenetic position makes it a potentially important group to understand larval and life cycle evolution in Annelida and more inclusive animal groups such as Bilateria as a whole. It is known for *Owenia* that the embryos develop into an enlarged anterior domain that forms larval tissues and the adult head, with the posterior segmented region is developing closer to metamorphosis. The authors show that different genes and genomic regulatory elements control the development of its feeding larva and adult stage in *Owenia*. By using comparative transcriptomics they show that *Owenia* shares most transcriptomic similarities at larval stages with bilaterian species with planktotrophic ciliated larvae and even the larvae of the cnidarian *Nematostella*, rather than to its closest relative (in the study) the annelid *Capitella teleta*, which has direct development.

B. Originality and significance.

The importance of this study is that the authors use their results to assess the currently dominant hypotheses concerning animal life cycle evolution; 1. The "intercalation" (co-option) hypothesis where larval stages were independently added to animal life cycles versus 2. the "terminal addition" (innovation) hypothesis where ancestral Bilateria resembled larvae. They reject them both, instead emphasizing the homology concept of 'head larvae' and the decoupling of head and trunk genetic programs. They suggest that this late deployment of trunk differentiation is likely ancestral for Annelida, and not necessarily related to maximal indirect development. They infer the decoupling of head and trunk genetic programs may have facilitated the evolution of larvae, which would then be homologous.

C. Data & methodology: validity of approach, quality of data, quality of presentation.

D. Data & methodology: validity of approach, quality of data, quality of presentation

All seems to be excellent. A huge amount of work drawn together to argue for a novel perspective on animal evolution

E. Conclusions: robustness, validity, reliability

F Suggested improvements: experiments, data for possible revision

The conclusions draw to gather a range of evidence, underlain by the novel work here on *Owenia*. I think it deserves publication as is. While it would be ideal to have 1. The sister group to Annelida and 2. more data for a range of annelids, especially in the paraphyletic grade with respect to the Sedentaria/Errantia clade, they do present results for *Chaetopterus* and argue you it to be a form of

head larvae. This may be a bit of a stretch since the anterior region then becomes a series of segments (See Irvine et al. 1999). Nevertheless it is not a classical trochophore and so there may well be the underlying homology there they argue for.

G. References: appropriate credit to previous work.

H. Clarity and context: lucidity of abstract/summary, appropriateness of abstract, introduction and conclusions

Citations are fine. It is a complex paper but is very well written.

Greg Rouse

Referee #5 (Remarks to the Author):

In this study, Liang et al. characterize the chromosome-level genome of *Owenia fusiformis* and explore stage-specific gene expression through transcriptomics and in situ hybridization, gene regulation through ATAC-seq, and gene family evolution. Their gene expression findings support that axial and body patterning (among some other processes) take place during an early developmental transcriptional phase, while adult trunk formation largely occurs in a second post-embryonic transcriptional phase during *O. fusiformis* life cycle. In light of these results, the case is made that heterochronic shifts in trunk development rather than co-option and innovation in genetic programs have driven bilaterian life cycle evolution. The work also provides insights into evolution of bilaterian genome organization and gene family gain/loss along evolutionary timescales. The manuscript is well-written (but see some comments about organization below).

I am very excited about this manuscript and its results. In my opinion, the manuscript is thorough, well-written, and will have an important impact on understanding of animal evolution. I am very familiar with all of the bioinformatic methods employed (except WGCNA; I'm also not knowledgeable about in situ hybridization) and the methodologies used here are exemplary (but see my requests for clarification in the "Minor points" below).

Major points:

The second paragraph of the results section is a lengthy report of / discussion on results related to synteny. This is interesting but of little importance to the take-home messages highlighted in the abstract of the paper. I feel that the authors are burying the lead and that this text should be reduced and combined with the first paragraph on more general genome characterization or moved towards the end of the results.

Related to the discussion of novel genes – what proportion of the predicted genes/transcripts have direct evidence from 1) the *Owenia* transcriptomes and 2) proteomes of other annelids and /or public databases like the input proteomes used for the Metzoa odb10 used here in BUSCO?

The code availability statement indicates that no custom code was used in this study, but I would argue that making the series of commands used to conduct the analyses presented herein is code and providing the series of exact commands used as well as key input/output files would improve the reproducibility and clarity about exactly what was done in this paper.

Minor points:

A non-trivial portion of the genome is not encompassed in the pseudomolecules/"chromosomes" – I assume that all of these scaffolds were included in downstream analyses?

The taxonomic authority should be listed after the first use of *Owenia fusiformis* (arguably other scientific names too but given the focus on this species, at least *Owenia* should have the authority mentioned).

Line 90: 10X genomics "read clouds" is vague (and the "g" in genomics should be capitalized).

Line 163: I use "expressed" when referring to mRNA and "localized" when referring to protein. The meaning here is clear from context, but I thought I would note this.

Line 198: Please clarify "mostly abundant within gene bodies" – within introns?

Line 243: I think "on" should be corrected to "in"

Lines 395-396: The methods for filtering the repeat-masked genome to recover genes is not adequately described.

Figure 5C – Amphinomida is clade including Amphinomidae and a second family (Euphrosinidae) – I suggest using this more inclusive name.

Supplementary tables 8-10: I would have liked to see at least 3 replicates per transcriptome.

Author Rebuttals to Initial Comments:

Response to Reviewers

We wish to thank the five referees for their positive appraisals and constructive comments, which we believe have significantly improved our work. We are pleased to provide a revised manuscript that we hope addresses the comments raised. The main changes to our manuscript are summarised as follows:

As recommended (referee #1), we now test explicit hypotheses by extending transcriptomic analyses to two additional annelid species with different life cycles compared to *Owenia fusiformis*: *Capitella teleta* (indirect lecithotrophic development) and *Dimorphilus gyrociliatus* (direct development). We thus now broadly cover the major types of life cycles in annelids, and, by combining inter-species comparisons of temporally co-regulated gene clusters with a comprehensive characterisation of the transcription factor repertoire in these three species, we define and test hypotheses on how gene expression and developmental programmes change according to life cycles in this group (referee #1, #2 and #3). These analyses have allowed us to uncover a small set of 28 transcription factors (incl. *Hox* genes) for which temporal expression covaries with developmental modes and larval types (in line with gene regulatory hypotheses raised by referees #1, #2 and #3).

To strengthen results on different dynamics of anterior and posterior genes during the development of *O. fusiformis* (referee #1), we leverage adult tissue-specific transcriptomes (referee #2 suggestion). These reveal that anterior genes dominate embryogenesis while posterior genes significantly outweigh post-larval stages in *O. fusiformis*. We also now strengthen our epigenomic studies (referee #1) by performing ATAC-seq profiling of developmental stages of *C. teleta*; the results of which provide additional evidence for observed transcriptional heterochronic shifts during life cycle transitions at the genome regulatory and transcription factor binding levels.

We have expanded our comparative transcriptomic analyses to other non-bilaterian taxa (referee #3 suggestion), explored the genes and transcription factors driving transcriptional similarities between metazoan primary larvae (referee #1 and #2 suggestions), and we have characterised the dynamics of species-specific genes in annelid and metazoan lineages (referee #1 comment).

We have extensively rewritten our manuscript to focus on the heterochronic shifts related to larval and life cycle diversification. In doing so, we have shortened the section on synteny evolution (referee #5 suggestion) and adjusted interpretations of our findings in relation to traditional scenarios for larval evolution and claims of homology, as suggested (referees #1, #2 and #3). Finally, we have built a public code repository (requested by referees #2 and #5) and incorporated all other text and figure suggestions, and formatted the manuscript to Nature requirements, with all main findings summarised in 5 main display items, 10 Extended Data figures and 31 Supplementary Figures.

Below we provide a detailed point-by-point response to each of the referees' concerns. We hope we have addressed them all satisfactorily, and we will be happy to address any additional comments and suggestions.

Referee #1 (Remarks to the Author):

*In this manuscript, Liang and Martin-Zamora et al. study the genome and developmental transcriptome of the annelid *Owenia fusiformis* to make inferences about the evolution of larvae across annelids, and also across the entire bilateral clade. First, they generated a chromosome-scale genome assembly, which has high contiguity and high completeness in terms of protein-coding gene content. Bulk transcriptome data across development revealed two major phases of change in gene expression, mirrored in the ATAC-seq data, and sets of co-regulated genes. Because only one cluster contains genes with expression both in larvae and in juveniles, the authors suggest that this means larvae could not have evolved via co-option of “adult” genetic modules, as proposed in the intercalation hypothesis. The authors then compared the *O. fusiformis* data to newly generated developmental transcriptomic data from another annelid, *Capitella telata*, and to published datasets from other bilaterians and a cnidarian. This analysis showed increasing similarity over developmental time, particularly at juvenile/adult stages, which challenges the “terminal addition” hypothesis that proposes that larvae are homologous and adult body plans evolved via addition of adult-specific genetic modules. Based on the authors’ studies of Hox gene expression, which appear late in development in *O. fusiformis*, and on published similar data from other species, the authors suggest that heterochronic shifts in trunk patterning modules underlie the evolution of larvae across bilaterians.*

Overall, the genome, transcriptome, and ATAC-seq data generated for this manuscript are well-executed and the analyses are by-and-large sound (with a few minor exceptions). These will serve as valuable resources to the evo-devo community. The authors’ efforts in trying to glean biological insight from the genome are commendable.

RESPONSE: Many thanks for the positive appraisal.

However, many of the claims in the paper are not supported by the data, as the authors offer interpretations but don’t consider alternative explanations. I suspect this is because the study is simply not designed to offer rigorous testing of different hypotheses. The paper also relies heavily on the framework of two existing hypotheses about larval evolution, which were postulated long before we knew much about gene regulation during development and evolution, which makes them pretty easy targets to take down. The central claim, that heterochronic shifts in trunk patterning modules may have shaped larval evolution, is reasonable, but doesn’t rely on any of the genome-scale work that is the focus of the paper.

ACTION: As detailed below, we have qualified the interpretation of our findings under traditional scenarios of larval evolution (also suggested by referee #2 and #3). Moreover, we now include new genome-scale transcriptomic and epigenomic datasets and analyses that we think allow rigorous testing of null and alternative hypotheses. Together, these new results strengthen our finding that heterochronic shifts in trunk patterning (and other processes) shape larval and life cycle evolution (**new Figures 2, 3 and 4** and associated Extended Data and Supplementary Figures).

The evolution of larvae and of bilaterian life histories is an important question, however meaningful and field-advancing statements about these processes will need more work, particularly functional studies.

ACTION: We agree and state this in the Discussion (**lines 375-378**: “In the future, comparative functional studies of these and other genes are needed to thoroughly dissect the regulatory principles underlying head and trunk development and decipher how temporal changes in gene expression and regulation have shaped the evolution of larval and adult forms in Bilateria.”).

Major Comments

Failure to consider null hypotheses and alternatives, and unsupported claims

There's a considerable list of claims that are not substantiated by the data:

*1) The authors should take a step back and clarify what their null expectations from the bulk transcriptome data might be. Given we know that the *O. fusiformis* larva is unique, and across annelids larvae can look different but produce adults that are recognizably "annelid", an a priori expectation would be that larvae might have very distinct transcriptomes.*

ACTION: We now expand our transcriptomic analyses to two annelid species with different life cycles (*C. teleta*, lecithotrophic larva; *D. gyrociliatus*, a paedomorphic direct developing annelid) and perform pairwise inter-species transcriptomic comparisons of (i) gene expression and (ii) soft clustering composition (based on all genes and only transcription factors) to assess how transcriptional dynamics change across life cycles. This allows us to define and test different expectations (**lines 148–151**), as also reflected in the main figures (**Figure 2e**, and **Figure 4d** for ATAC-seq data). As suggested, our null expectation is that per-cluster gene composition between species will be different at larval stages. Our analyses show that gene composition and expression is similar early in development, but innovation (e.g., at late stages) and temporal shifts in gene expression are common (**Figure 2f, g; Extended Data Figure 4; lines 148-177**) and likely underpin larva and life cycle diversification.

I think this is what underlies the "biphasic" pattern in Figure 2b. PC 2 is saying that the larva is quite different, as we might expect.

ACTION: Following referee #2 comment, we do not refer anymore to a biphasic development in *O. fusiformis*. The PCA is now in Supplementary Figure 2.

*2) The statement that the "*O. fusiformis* larva does not co-opt genes expressed in the adult" cannot be made based on these data. There is nothing about the intercalation hypothesis that specifies how many genes or what proportion of genes one would expect to be shared between larvae and adults.*

ACTION: We agree with the referee and removed the statement.

The authors report 1,426 genes that are indeed expressed in both larvae and adults. How do we know that there aren't key, important genes in this list that could be causal to the formation of larvae? I don't necessarily believe that the intercalation hypothesis is correct, but the authors are not providing compelling data to disprove it.

ACTION: We now include Gene Ontology (GO) enrichment analyses of all bimodal clusters of temporally co-expressed genes in *O. fusiformis* and *C. teleta*, as well as clusters that span three or more stages and the larval time points in these species (**Extended Data Figure 3e–g**). We now refer to these clusters and their associated GO terms, which are largely involved in core cellular process, in the main text (**lines 132-141**). In addition, we modified the interpretation of our findings with respect to the intercalation and terminal addition hypotheses.

*3) In line 189, the authors state "our findings support that axial and body patterning, as well as anterior and gut differentiation contribute to the first transcriptional phase, while adult trunk formation largely drives the second post-embryonic transcriptional phase during *O. fusiformis* life cycle." The gut differentiation claim comes from the GO analysis mentioned in line 158, but the data in this paper do not allow any quantitative statements about anterior patterning or trunk formation. Yes, these genes turn on at specific times, but how large or small an effect they are driving can only be determined either by quantitatively assessing the significance of their contribution relative to that of other genes, or by doing functional studies.*

ACTION: Following a suggestion by referee #2, we used the adult tissue-specific transcriptomes in *O. fusiformis* to define anterior and trunk/posterior genes and statistically quantify and compare their expression dynamics during *O. fusiformis* life cycle (**Figure 3d; Extended Data Figure 6**). Trunk/posterior genes significantly outweigh anterior genes from the mitraria stage onwards when their developmental expression dynamics are compared, while embryogenesis is dominated by anterior genes, which is consistent with the morphological formation of head and trunk. Moreover, by using a candidate gene approach with genes known to be expressed in anterior, trunk and posterior domains by *in situ* hybridisation in *O. fusiformis*, *C. teleta* and *D. gyrotilatus*, we demonstrate that the development of these three body regions follows different quantitative and qualitative temporal dynamics in annelids with different life cycles (**Figure 3e; Extended Data Figure 6**). These new analyses together with the findings from the genome-wide inter-species inter-cluster comparisons (**Figure 2f, g; Extended Data Figure 4**) are three independent and complementary evidence that support our main claim that heterochronic shifts in developmental programmes correlate and might explain life cycle evolution in Annelida.

*4) line 241 states “older rather than younger genes contribute to the development of the mitraria larva, suggesting that the increased use of novel genes in other lophotrochozoan larvae^{7,42} might be due to lineage-specific traits found on those larvae,”. There’s an alternative interpretation here. Given that the *O. fusiformis* larva is a “head larva” to an extreme, and anterior patterning genes are highly conserved, it may be that this aspect of the biology is driving the observation that older genes are dominating the mitraria. So it is less about other larvae having lineage specific traits, but about Owenia having expanded its pre-oral domain.*

ACTION: We have amended this sentence based on the updated comparative phylostratigraphic analyses of the three annelid species with distinct life cycles (**lines 302-306**: “Therefore, genes of different evolutionary origins contribute to the development of annelid larvae, suggesting that the increased use of novel genes in some lophotrochozoan larvae^{7,46} might be due to lineage-specific traits, such as the shell primordium of molluscan trochophores⁷ and perhaps even ciliary bands with multiciliated cells⁴⁸, which are absent in oweniid larvae.”).

5) That adult body plans in bilaterians share a ground plan and patterning genes has been stated by others (e.g. Raff 2008, Phil. Trans. R. Soc. B 363, 1473–1479).

The finding in this paper that at a large transcriptome-wide scale (focusing on single copy genes) there is similarity among bilaterian adults could be explained by the fact that adult bodies operate on many similar cell biological functions - digestion, excretion, muscle contraction, synaptic transmission, and so on. Thus, given what we know about animal biology in 2022, this could be our null expectation.

ACTION: We now include this reference and explanation in the context of adult body plan conservation (**lines 328-330**: “Together, these findings suggest that adult development is generally more similar¹⁰ than early embryogenesis across major animal lineages, maybe because adult bodies operate on many similar cell biological functions.”).

I am not a proponent of the terminal addition scenario, but technically, the terminal addition scenario does not specify how many genes would one need to observe being different across adults of different animal species for the hypothesis to be correct. The terminal addition hypothesis also does not say that homologous larvae don’t evolve and diverge across species, so how can we specify what level of transcriptome similarity we should expect?

ACTION: The referee is correct. Considering this and other comments, we have moderated the interpretation of our data under the intercalation/terminal addition scenarios and

following referee #3 suggestions, rephrased the Introduction (**lines 56-58**: “How these proposed mechanisms of co-option and innovation might translate to our current understanding of the genetic control of animal embryogenesis remains, however, unclear.”).

The data in this paper cannot rule out that a few key genes were the ones needed to activate genetic modules for making adult bodies work, and this occurred differently in different species.

ACTION: Our new genome-wide inter-species transcriptomic comparisons among three annelids with different life cycles do suggest this as a possibility (**Figure 2e-g; Extended Data Figure 4**). We identified 28 common transcription factors that consistently show distinct temporal expression dynamics across species of different developmental mode (i.e., indirect and direct) and larval type (planktotrophic and lecithotrophic). These are upstream regulators of body patterning (e.g., *Hox1* and *Hox4*, *gli*) and organogenesis (e.g., *myoD*, *pax6*), among other processes. Thus, this finding supports the referee’s hypothesis that there might be a small set of key genes controlling body plan evolution and that their different deployment in time and space might promote larval body plan and life cycle diversification (**lines 174-177**: “Therefore, although it is likely a combination of factors that explain transcriptional differences, heterochronic changes—sometimes involving just a few key genes related to a range of genetic programmes—correlate, and might account for, life cycle and larval differences in Annelida.”).

Additionally, the apparent similarity between adult stages could just be a byproduct of the fact that species with extremely divergent larvae are being compared. I don’t think there is any reason, a priori, to expect large scale similarity between drosophila, nematode, and annelid larvae, it is like comparing apples to oranges.

ACTION: We removed the statement comparing *O. fusiformis* with ecdysozoan larvae.

6) The authors use orthologous single-copy genes for the analysis in Figure 4. This analysis does not accommodate species-specific genes, so it cannot assess whether novel genetic modules were needed to make adult body plans, which could be an important aspect for the terminal addition scenario. This consideration is not mentioned in the paper.

ACTION: To address this point, we extended our phylostratigraphic analyses to two other annelid species (*C. teleta* and *D. gyrocoliatius*) and all metazoan species included in our comparative transcriptomic analyses (**Figure 5a; Extended Data Figure 8; Supplementary Figure 30**). The expression of younger (or species-specific) genes is not always higher at adult stages (e.g., species-specific genes dominate juvenile stages in *O. fusiformis*, but the blastula in *C. teleta*, or gastrulation in zebrafish; **lines 296-300**). Although we cannot rule out that just a few species-specific genes have an important role to define adult body plans, these analyses highlight that the overall developmental contribution of taxonomically restricted genes varies among taxa. Nevertheless, we have also generally modulated the interpretation of our findings under the terminal addition scenario.

7) The maximal similarity between O. fusiformis larva and other planktotrophic larvae could simply be driven by convergently-evolved functional properties, such as having to make cilia and to swim around to eat.

ACTION: To test this, we examined the GO annotation of genes with the highest transcriptomic similarity between the mitraria of *O. fusiformis* and its corresponding larval phases of maximal transcriptomic similarity in *C. teleta*, *C. gigas*, *S. purpuratus*, *C. hemisphaerica* and *N. vectensis* (**Extended Data Figure 9f, g**). As the referee suggests, these are largely involved in core cellular processes, and might just reflect convergently evolved functional properties of the larvae (as we now state in **lines 322-326**). However, a comparative transcriptomic analysis with just one-to-one transcription factor orthologs shows

that larval phases between these species are also stages of maximal similarity (**Extended Data Figure 9a, b, e**). This suggests that developmental and regulative programmes might also be contributing to larval transcriptomic similarities (**lines 326-328**).

So the pattern of similarity and differences in transcriptome profiles across species does not directly enable an assessment of homology. Furthermore, the heterochronic shifts in trunk patterning across annelid larvae could easily be driven by key upstream regulators that are different between these species. Thus the observations in this paper do not clearly support homology or convergence of annelid (or bilaterian) larvae. The authors do admit this possibility, so I recommend that they lead with the idea of being open to either scenario rather than showing preference to one (e.g. the claim “our study supports that the late deployment of trunk differentiation programmes is likely ancestral to Annelida”).

ACTION: We have removed the statement about homology and provide now an open discussion about each scenario. We also included a panel in Figure 5 (**Figure 5f**) depicting both possible scenarios (homology/convergence).

8) The authors also should consider homology with all of the complexity that is associated with this term. “larvae, which would then be homologous on the grounds of being largely anteriorly derived transitory structures” could mean that there is process homology in terms of anterior genetic modules being used across larvae; this is not the same as larvae being homologous. But stating that larvae themselves are homologous is saying that the ancestor of all bilaterians had a larval stage that became modified in descendant taxa. It is unclear what the authors mean by homology here.

ACTION: We now clarify this point in the Discussion (**lines 367-369**: “This ancestral temporal decoupling between head and trunk genetic programmes could thus have facilitated the evolution of larvae, which would then originally share the use of anterior genetic modules for their development (Fig. 5f).”).

ATAC-seq analysis and interpretations

The biphasic nature of ATAC-seq peaks and specifically the finding that the largest changes in peak accessibility occur in the larval stages is completely consistent with what was observed in the transcriptome data, and would be the null expectation for this work. Given that transcriptionally the larvae are the most different, corresponding differences in chromatin dynamics would be expected to underlie that. Other than yielding what is expected, and providing a dataset for future work, the ATACseq data are not adding to the biological inferences in this manuscript.

ACTION: To improve the biological inferences of the ATAC-seq data, we now include an equivalent time course of ATAC-seq for *C. teleta* (**new Figure 4** and associated Extended Data Figure 7 and Supplementary Figures 15–29). We reanalysed all ATAC-seq data and include new analyses to cluster predicted motifs and robustly annotate them to transcription factor classes (see below), as well as to detect transcription factor footprinting. This new comparative analysis of the ATAC-seq data in *O. fusiformis* and *C. teleta* reveals different dynamics of genome regulation in two annelids with different larval types and supports the finding that heterochronic shifts in similar regulatory modules (e.g., *Hox*) correlate with, and possibly underpin, life cycle transitions in Annelida (new section in **lines 223-283**).

The authors should explain if peak widening has been observed in other species/contexts and whether it is known to have any particular biological relevance.

ACTION: Given the new ATAC-seq data, we have removed this statement and analysis.

Additionally, the specificity of the TF motifs needs to be explained if the authors want to claim biological functions. For example, stating that “Consistent with our transcriptomic

dataset, motifs related to transcription factors involved in patterning anterior territories (e.g., PAX2/5/8 and PAX4/627), muscle and gut development (GATA28, FOXC28) and early neurogenesis (ATOH41) are amongst the most differentially accessible in open chromatin regions during embryogenesis, whilst ciliary band genes (OTX28), trunk related genes (e.g., NKX2.127) and most notably Hox genes appear in regulatory regions during larval competence and trunk patterning” requires that we know that the binding sites annotated as specific to FOXC or NKX2.1 in O. fusiformis are indeed likely to bind the specific orthologs that have roles in anterior and trunk patterning respectively. Usually, annotating binding sites in new model systems using vertebrate databases will yield specific names of motifs, but this does not mean that the sites will be regulated by the particular TF. For example, sites annotated could be bound by any number of other Fox TFs, most of which are NOT specific to the biological functions that FoxC performs. NKX2.1 sites could be bound by other Nkx homologs, which aren’t trunk patterning genes. Thus, these claims are a big leap. The authors have to either show functional associations with specific orthologs and sites.

ACTION: To address this point, we clustered the predicted motifs in *O. fusiformis* and *C. teleta* by sequence similarity into 141 clustered motif archetypes, of which 51 are common to both species, 43 are unique to *C. teleta* and 47 are unique to *O. fusiformis* (**lines 250-255**). To relate these motifs to transcription factors, we used four different databases (JASPAR2022, HOMER, CIS-BP and gimme vertebrate version 5.0; Supplementary Figure 22) and manually curated the annotation to validate the sequence similarity of the cluster to the consensus human sequence (as stated in a comprehensive, non-redundant transcription factor database of motif archetypes; <https://resources.altius.org/~jvierstra/projects/motif-clustering-v2.0beta/>). Those that passed manual curation were named according to the transcription factor class as in the TFclass database (<https://doi.org/10.1093/nar/gkx987>). As a result, we annotated 71 of the 141 archetypes, including 31 of the 51 that are common to *O. fusiformis* and *C. teleta*. Importantly, these common motifs confidently represent most of the main classes of transcription factors (e.g., TBOX, FOX, GATA, and subclasses of homeodomains), providing a comprehensive picture of the regulatory dynamics during the development of *O. fusiformis* and *C. teleta* (**new Figure 4d–i** and Supplementary Figures 23–28). In addition, we refer to the motif archetype itself (e.g., HOX/CDX/EVX; **lines 261-279**) throughout the text and not to the potential transcription factors that would bind to it.

Central claim

The Hox expression data convincingly show that trunk patterning genes are expressed late in development, lending support to the idea of a heterochronic shift in O. fusiformis. This inference does not rest on the genome scale work that the majority of the manuscript focuses on, and this shift was already known from another annelid, Urechis unicinctus. So the data in this paper add to the story, but don’t on their own, introduce a wholly new idea.

ACTION: To support our central claim with genome-scale data, we now include (i) comparative transcriptomic analyses in three annelid species with different life cycles (**new Figure 2 and Extended Data Figures 3 and 4**); (ii) quantitative temporal analysis of anterior and trunk/posterior genes as defined from adult tissue-specific transcriptomes (**Figure 3d and Extended Data Figure 6a–f**); and (iii) extensive comparative epigenomic analyses during larva development in *O. fusiformis* and *C. teleta* (**new Figure 4 and Extended Data Figure 7**). As detailed in previous points, all these new datasets and analyses back and strengthen our original claim that heterochronic shifts in transcriptional and regulatory programmes (and not only involved in trunk development) correlate and probably underpin larval body plan and life cycle diversification in Annelida, and likely Bilateria generally.

Although we use existing RNA-seq data for *U. unicinctus*, the temporal expression dynamics of *Hox* genes in this species were, to the best of our knowledge, unknown.

Minor comments:

line 168: “in line with the vastly enlarged pre-oral region of the mitraria that forms early on in the adult head” - I am not sure what it means for the pre-oral region of the mitraria to form in the adult head. Do you mean this region makes the adult head?

ACTION: We have removed this sentence.

line 172: “*Hox* genes, a conserved family of transcription factors involved in anterior-posterior trunk regionalisation in Bilateria³⁶, are among the most upregulated genes at these stages (Fig. 2g).” - it was not easy to find the ranks of all of the *Hox* genes in the list of DE genes. Only a small subset of these genes are highlighted in the volcano plot in Figure 2.

ACTION: We now show all differentially expressed *Hox* genes in the volcano plot (**Extended Data Figure 5b**), rather than only those with a $\log_2(\text{fold-change}) > 5$.

line 180: “*O. fusiformis* does not express *Hox* genes during embryogenesis” - the levels look low in heat maps, but were *in situ* hybridizations done in stages earlier than mitraria to confirm this? Transcription factors can have low but biologically meaningful expression despite overall low levels in transcriptome data.

ACTION: We now include images of whole mount *in situ* hybridisation for all *Hox* genes at the gastrula stage (**Extended Data 5f**) when there is no detectable expression.

Figure 2 legend: Given that Figure 2f is a schematic, the legend should make clear where the *in situ* hybridization data for these genes can be found. It should clarify “based on references XYZ”.

ACTION: We removed this panel from the revised version.

Referee #2 (Remarks to the Author):

The authors introduce the genome of the annelid *Owenia fusiformis*, as well as a comprehensive catalog of gene expression and chromatin accessibility profiles at different developmental time points. This is a high-quality genome, which will constitute a must-use in any future comparative genomics efforts and one that immediately places *Owenia* at the top of annelid model systems. I want to congratulate the authors for the effort.

The functional genomics datasets are also high quality and I have no concerns regarding data generation and analysis.

RESPONSE: Many thanks for the positive assessment.

These datasets are presented in the context of a discussion about the evolution of larval stages, proposing an intermediate solution to the intercalation (independent larval origins) versus terminal addition (ancient larval form, parallel evolution of adult forms). This represents my main concern with the manuscript: I find this interpretation confusing and even contradictory at times.

ACTION: Given this concern and those along similar lines by referee #1 and #3, we have streamlined the manuscript to focus the interpretations of our data on the gene expression and regulatory changes associated with different life cycles rather than on favouring/rejecting a particular traditional evolutionary scenario (e.g., **new panel Figure 1a, b**). As suggested below by this referee, we now expand and strengthen our data and conclusions on the role of gene expression heterochronies in life cycle and larval body plan evolution in Annelida and Bilateria generally (e.g., **new main Figures 2, 3 and 4**).

For example, I don't think the observed temporal gene expression dynamics is incompatible with an intercalation scenario. First, maybe the temporal sampling hasn't enough resolution to detect "co-opted genes" from adult to larval development, or simply these genes are few and not recovered in the gene module analysis (looking at extended data fig.4a it seems there are a number of such genes).

ACTION: As indicated above, we have now removed statements claiming our data rejects the intercalation scenario. Moreover, our new data and transcriptomic analyses support the referee's statement that a few genes might be having a major effect in life cycle transitions and evolution of larval body plans. For instance, we identified 28 transcription factors (including *Hox* genes) showing consistent temporal shifts between direct and indirect developers, and between planktotrophic and lecithotrophic larvae (**new Figure 2; lines 157-164**). Likewise, our new comparative epigenomic data in *O. fusiformis* and *C. teleta* suggest that temporal changes in *Hox* regulation—perhaps through the modulation a few conserved upstream regulators—might also underpin the changes in the onset of trunk development between planktotrophic and lecithotrophic annelid larvae (**new Figure 4; lines 269-275**).

The situation is similar with the ATAC-seq analysis. In fact, this data does suggest that many motifs are shared between mitraria larva and adult (Fig 3e).

ACTION: As detailed above, we now include an equivalent time course of ATAC-seq for *C. teleta* and analyse ATAC-seq data at the level of both motif accessibility and transcription factor footprinting (**new Figure 4 and Extended Data Figure 7**). There are indeed motifs that are accessible at both mitraria and adult/competent stages, albeit the proportion of those is smaller in *O. fusiformis* than in *C. teleta* (**Figure 4c; lines 236-241**). This is consistent with a more gradual shift in transcriptional and regulatory programmes in the lecithotrophic larva than in the planktotrophic one (as also supported by GO terms enrichment analyses of the genes regulated by each temporal cluster of co-regulated regulatory regions; **Supplementary Figure 21 and lines 241-245**).

Overall, it does seem that adult and larval stage share features, both at the gene expression as well as the accessibility level, and they are the result of a continuous gradient, rather than distinctive phases.

ACTION: We have removed our previous interpretation of our data according to two phases of development. As indicated in the previous point, our new data and analyses do show a more continuous gradient of transcriptional and regulatory activity during development, especially in lecithotrophic larvae and the direct developer. Yet, the analysis of adult transcriptomes in *O. fusiformis* and of genes with known spatial domains in anterior/trunk/posterior regions also reinforces that anterior and posterior/trunk development are temporally decoupled in the planktotrophic larva, while roughly co-occur in the lecithotrophic larva and the direct developer (**Figure 3d, e; Extended Data Figure 6; lines 201-221**).

Moving to cross-species analysis, the comparison between Capitella and Owenia seems to support the intercalation scenario, showing maximal divergence at the larval stages (line 258). But further comparison with other species points at opposite patterns, suggesting that these comparisons are confounded by differences in the cellular composition, tissue heterochronies, etc.

ACTION: As indicated in a previous point to referee #1, the analysis of the genes with more similar expression dynamics between the mitraria and other larval stages confirms that genes involved in core cellular processes (e.g., cellular components and complexes) contribute to transcriptional similarities between larval stages (**Extended Data Figure 9f, g**). However, the transcriptional similarity is also high at those larval phases based solely on one-to-one

transcription factor orthologs (**Extended Data Figure 9a, b, e**). We therefore conclude that both similarities/differences in core cellular process and regulatory programmes might underpin transcriptional similarities across metazoan larvae (**lines 322-328**).

The apparent opposing patterns between *C. teleta* and other metazoan lineages is because we define the relative Jensen-Shannon divergence (JSD) between the points of maximal transcriptional similarity relative to each pairwise comparison. Yet overall, when the normalised JSD value across all pairwise comparisons is considered, *Owenia fusiformis* has more similar transcriptional dynamics during development with *C. teleta* than with other metazoan taxa (as shown in the heatmaps in Extended Data Figure 9a and 10a). We now clarify, however, that this is relative similarity (**lines 313-315**: “In relative terms, global transcriptional dynamics between *O. fusiformis* and other major animal groups tend to be more dissimilar at early development [...]”).

In summary, I do not think the temporal gene expression and accessibility data provide much insight into the question of larval homology and the origin of bilaterian life cycles.

ACTION: We have now removed our statement on larval homology. Following referee #1 suggestion, we have also moderated our interpretation of the origin of bilaterian life cycles to keep our conclusions more open (**new Figure 5f panel; lines 357-369**).

In any case, the authors should adequately problematize the data in the context of the major intrinsic limitation of such bulk analyses: the presence of heterochronies and the absence of spatial/tissue resolution make it very difficult to interpret temporal expression/accessibility in terms of high or low similarity between adult and larval stages, as well as between species.

ACTION: We now refer to this limitation in the discussion (**lines 370-371**: “Regardless of the scenario, however, and despite the limitations that whole-embryo transcriptomic and epigenomic datasets without spatial resolution have, our study [...]”).

In contrast, I do find the results of the Hox expression pattern analysis very interesting and the interpretation of these results convincing, both in supporting the homology of adult bilaterian body plans, as well as in suggesting an important role for heterochronies in the evolution of larval structures.

ACTION: Our new comparative transcriptomic and epigenomic data expands on these initial findings, consolidating the presence of a heterochrony in *Hox* activity and trunk development between annelids with different life cycles (**new Figures 2, 3, and 4**). We also extend the potential role of transcriptional and regulatory heterochronies to other transcription factors (**Figure 2g**) and larval traits, such as the formation of larval-specific organs (e.g., development of defensive chaetae in free living planktotrophic larvae) and larval metabolism (e.g., early activation of autophagy in yolk-rich lecithotrophic larvae) (**Extended Data Figure 4i; Supplementary Figure 14**). We have streamlined our manuscript to focus on these key points.

Minor points

- I'm surprised the authors don't use the adult tissue transcriptomes in any of the analyses.

ACTION: Following the referee's suggestion, we used these adult transcriptomes to identify anterior and trunk/posterior genes and quantify the temporal transcriptional dynamics of these two classes during the development of *O. fusiformis* (**Figure 3d**). This allowed us to statistically demonstrate that the expression dynamics of trunk and posterior genes outweigh anterior genes from the mitraria stage onwards.

- The authors should provide the code for reproducing the analyses presented in the manuscript.

ACTION: We have assembled a public repository with all the code and files required to reproduce our analyses. It is available at <https://github.com/ChemaMD/OweniaGenome>.

Referee #3 (Remarks to the Author):

Liang et al. have undertaken a thorough analysis of the Owenia fusiformis genome, and its developmental regulation and expression. O. fusiformis is a member of a basal-branching annelid lineage. Overall, the results and analyses are impressive and high-quality, and the data appear to be robust. Together, they contribute to our understanding of the biology and evolution of annelids and lophotrochozoans, recognising the important caveat that O. fusiformis possesses a derived larval type restricted to members of family Oweniidae (the mitraria). This is in contrast to the trochophore larva that is present in most annelids and multiple closely-related phyla, including molluscs.

ACTION: Many thanks for the positive assessment of our work.

Regarding the morphological divergence of the larva of *O. fusiformis*, we amended the second paragraph of the Introduction to clarify that annelids show a variety of life cycles and larval types (often deviating from the archetypical trochophore, with Oweniidae larvae being a clear example of this; **lines 66-68**). We argue that it is indeed this diversity of life cycles and larval types that makes Annelida—and its apparently divergent lineages—attractive systems to understand how life strategies and larvae diversified (**lines 76-79**: “Therefore, Annelida, with its diversity of life cycles and larval forms, but generally conserved early embryogenesis and adult body plans, emerges as an excellent model to investigate how larval traits evolve, and thereby formulate and assess hypotheses on the origin of animal life cycles.”).

Despite having a derived larva and some odd developmental features – some oweniids were thought to undergo deuterostomous development, which was shown to be wrong by the corresponding author and colleagues (Martin-Duran et al. 2016 Nature Ecol Evol) - the analysis O. fusiformis development provides some important insights (e.g. Hox expression and gene regulation). However, these results do not necessarily contribute to the main proposition of the paper about bilaterian body plan evolution.

ACTION: To address these and other concerns raised by referee #1, we expanded our transcriptomic analyses to *C. teleta*, an annelid with a lecithotrophic larva, and *D. gyrociliatus*, an annelid with direct development. This has consolidated our original findings regarding trunk development and revealed a small set of 28 transcription factors whose expression dynamics temporally shift according to the mode of development and larval types (**new Figure 2 and updated Figure 3**). Moreover, new comparative epigenetic analyses reveal that temporal shifts in *Hox* regulation and *Hox* regulatory activity occur during the development of the lecithotrophic and planktotrophic larvae of *C. teleta* and *O. fusiformis*, respectively (**new Figure 4**). Together, these new analyses and others highlighted below, expand and solidify our original findings that heterochronic changes in trunk development correlate and might promote larval body plan evolution in Annelida and probably many other bilaterian lineages (**updated Figure 5 and discussion**).

The authors posit that changes in the timing of trunk formation in annelids, which occurs by the addition of new segments at the posterior end by teloblast growth, sheds light on the evolution of larvae and bilaterian life cycles.

ACTION: We now refer to the studies by Seaver *et al.* (<https://doi.org/10.1111/j.1525-142X.2005.05037.x>) with *C. teleta* and *Hydroides elegans* (an annelid with a more canonical planktotrophic trochophore), Prud'homme *et al.* (<https://doi.org/10.1016/j.cub.2003.10.006>) with *P. dumerilii* (lecithotrophic larva) and Wilson (with *O. fusiformis*). These demonstrate

that the onset of trunk formation in the annelid larva is likely through lateral growth and morphogenesis, with a posterior growth zone appearing only after metamorphosis in the juvenile (**lines 218-219**: “Therefore, trunk development, which initially occurs from lateral growth of the trunk rudiment^{22,30,41}, [...]”).

This study is built on the premise that analysis of one clade of bilaterians can address long-standing, conflicting hypotheses about the origin of bilaterian larvae and body plans – “intercalation” (larvae are an elaboration of embryonic development of a direct developer, promoting dispersal and habitat selection) vs “terminal addition” (sexually competent larval-like adults are ancestral to which a second life cycle phase was added that then became the sexually-competent adult). The authors state that in (1) the “intercalation” scenario the larval body plan evolves by co-opting adult gene regulatory networks that lead to adult-enriched gene expression profiles and (2) “terminal addition” scenario where that adult evolved secondarily by incorporating new genetic programmes. This statement sets-up the comparative analyses they undertake and is used to support their argument, which is the main premise of this manuscript. This construct however does not reflect current thinking about how gene regulatory networks and gene pleiotropy might manifest in these two scenarios. It is highly likely not to be as black-and-white as set-out in the manuscript. There is not a single way in which gene expression can manifest under these two scenarios. For instance, why is gene co-option invoked in one scenario and not the other? The addition of life cycles stages does not require the invention of new genes or gene networks (e.g. complex life cycles of parasitic platyhelminths or triphasic cnidarians with a medusa). The final paragraph of the paper recognising some of these caveats and thus undermines the proposition framework the authors established in the introduction and that runs through the manuscript.

ACTION: Given this referee’s concerns (like those by referee #1 and #2), and all new comparative analyses between three annelids with different life cycles (see points above), we have rewritten the manuscript to modulate the interpretation and justification of our analyses under traditional scenarios on larval evolution. In addition, we rephrased the initial question in the Introduction, to reflect the referee’s point that it is unclear how the traditionally proposed mechanisms for larva evolution would reflect under our current understanding of animal development (**lines 56-59**: “How these proposed mechanisms of co-option and innovation might translate to our current understanding of the genetic control of animal embryogenesis remains, however, unclear, and thus larval origins—and their importance to explain animal evolution—are still contentious.”).

As mentioned above, this is an impressive characterisation of an annelid genome and its developmental regulation. There are some interesting correlations between planktotrophy and lecithotrophy in relation to the expression of the Hox code. It would be interesting to see if this stood up after comparisons if other phyla, including molluscs.

ACTION: We now include a heatmap with the timing of activation of *Hox* genes in the metazoan species used for comparative transcriptomics analyses (**Extended Data Figure 10b**). This shows that three species with planktotrophic larvae (the annelids *O. fusiformis*, *U. unicinctus* and the sea urchin *S. purpuratus*) activate the *Hox* code after larval stages, while lineages with direct development or non-planktotrophic larva, such as amphioxus, activates *Hox* genes after gastrulation, as *C. teleta* does (lecithotrophic annelid larva). This does not apply to molluscan planktotrophic larva, however, since molluscs have apparently altered the normal deployment of *Hox* genes during development due to the evolution of a shell (<https://doi.org/10.1073/pnas.1907328117>).

However, the argument that this study provides critical insights into the evolution of body plans is less compelling. The author's case is not clearly articulated and the choice of taxon is questionable.

ACTION: As indicated above, we now clarify the choice of taxon, which together with the inclusion and thorough analyses of transcriptomic and epigenomic data for two additional annelids (**new Figures 2, 3, 4** and associated Extended Data Figures), extends and solidifies our original findings on the evolution of larval body plans and diversification of life cycles in Annelida.

It seems to answer such questions one needs to look at a deeper nodes. Cnidarians and even sponges have biphasic life cycles with ciliated larval forms suggesting that both "intercalation" and "terminal addition" theories have little relevance when trying to understand the evolution of bilaterian larval and adult body plans.

ACTION: We now include a sponge (*Amphimedon queenslandica*) and a hydrozoan (*Clytia hemisphaerica*) in our comparative transcriptomic analyses (**Extended Data Figure 9**). This confirms the transcriptomic similarities between the ciliated larva of *O. fusiformis* and cnidarian planulae, as well as the similarities of early/late development between *O. fusiformis* and the sponge. As indicated above, we balance the interpretation of our study under traditional evolutionary scenarios of animal larvae.

Put simply, different selective forces at different stages of the life cycle – coupled with developmental biases and contingencies – is enough to explain modern developmental and body plan diversity.

ACTION: We agree with the referee and hope the new data and interpretations better reflect this point. For instance, our comparative transcriptomic analyses between annelid larvae not only reveal heterochronic shifts in trunk formation, but also in the development of other structures (e.g., chaetae) or even the activation of metabolic pathways (e.g., autophagy for yolk consumption) (**Extended Data Figure 4i; lines 164-173**). This supports the referee's view that many selective forces (e.g., development of defensive structures in planktotrophic larvae with long free-living swimming stages) and developmental contingencies (e.g., yolk levels and energetic demands) shape larvae and life cycles. And among those, there are also temporal shifts in trunk development, which appear to recurrently associate with marine planktotrophic larvae in Annelida and many other bilaterian lineages.

Referee #4 (Remarks to the Author):

A. Summary of the key results

*This paper presents the chromosome-scale genome of *Owenia fusiformis* along with transcriptomic and epigenomic profiling of various life history stages of the species. By comparing with 22 other animal genes they show that *Owenia* has fewer gene family gains and losses, and retains more ancestral metazoan orthogroups than found in other annelid genomes known to date. The analysis of transcriptomes and regulatory genome also allowed them to assess gene regulatory events for the formation and development of the mitraria larva. *Owenia* is part of a clade with low species richness that is the sister group to the rest of Annelida. *Owenia* also has unusual larvae among annelids and does not have a 'classical' trochophore stage. *Owenia*'s phylogenetic position makes it a potentially important group to understand larval and life cycle evolution in Annelida and more inclusive animal groups such as Bilateria as a whole. It is known for *Owenia* that the embryos develop into an enlarged anterior domain that forms larval tissues and the adult head, with the posterior segmented region is developing closer to metamorphosis. The authors show that different genes and genomic regulatory elements control the development of its feeding larva and adult stage in*

Owenia. By using comparative transcriptomics, they show that *Owenia* shares most transcriptomic similarities at larval stages with bilaterian species with planktotrophic ciliated larvae and even the larvae of the cnidarian *Nematostella*, rather than to its closest relative (in the study) the annelid *Capitella teleta*, which has direct development.

B. Originality and significance.

The importance of this study is that the authors use their results to assess the currently dominant hypotheses concerning animal life cycle evolution; 1. The “intercalation” (co-option) hypothesis where larval stages were independently added to animal life cycles versus 2. the “terminal addition” (innovation) hypothesis where ancestral Bilateria resembled larvae. They reject them both, instead emphasizing the homology concept of ‘head larvae’ and the decoupling of head and trunk genetic programs. They suggest that this late deployment of trunk differentiation is likely ancestral for Annelida, and not necessarily related to maximal indirect development. They infer the decoupling of head and trunk genetic programs may have facilitated the evolution of larvae, which would then be homologous.

C. Data & methodology: validity of approach, quality of data, quality of presentation.

D. Data & methodology: validity of approach, quality of data, quality of presentation.

All seems to excellent. A huge amount of work drawn together to argue for a newer perspective on animal evolution.

RESPONSE: Many thanks for the positive appraisal.

E. Conclusions: robustness, validity, reliability.

F Suggested improvements: experiments, data for possible revision.

The conclusions draw to gather a range of evidence, underlain by the novel work here on *Owenia*. I think it deserves publication as is. While it would be ideal to have 1. The sister group to Annelida and 2. more data for a range of annelids, especially in the paraphyletic grade with respect to the Sedentaria/Errantia clade, they do present results for *Chaetopterus* and argue you it to be a form of head larvae. This may be a bit of a stretch since the anterior region then becomes a series of segments (See Irvine *et al.* 1999). Nevertheless, it is not a classical trochophore and so there may well be the underlying homology there they argue for.

ACTION: We agree that more data in other annelids and resolving the sister group to Annelida will improve our understanding of larval evolution and help to test our proposed scenarios. We now moderate similarities with *Chaetopterus* larva, and reference a recent work by Helm *et al* (<https://doi.org/10.1007/s13127-022-00553-z>) uncovering further similarities between Chaetopteridae and Oweniida larvae (**lines 340-342:** “This occurs in other feeding annelid larvae⁴² (Extended Data Fig. 5f), and likely in Chaetopteriformia⁴⁹⁻⁵² too, and thus the late differentiation of the adult trunk might be an ancestral trait to Annelida (Fig. 5e).”). We also added the reference to Irvine *et al.* 1999.

G. References: appropriate credit to previous work.

H. Clarity and context: lucidity of abstract/summary, appropriateness of abstract, introduction and conclusions.

Citations are fine. It is a complex paper but is very well written.

Greg Rouse

Referee #5 (Remarks to the Author):

*In this study, Liang et al. characterize the chromosome-level genome of *Owenia fusiformis* and explore stage-specific gene expression through transcriptomics and in situ hybridization, gene regulation through ATAC-seq, and gene family evolution. Their gene expression*

findings support that axial and body patterning (among some other processes) take place during an early developmental transcriptional phase, while adult trunk formation largely occurs in a second post-embryonic transcriptional phase during O. fusiformis life cycle. In light of these results, the case is made that heterochronic shifts in trunk development rather than co-option and innovation in genetic programs have driven bilaterian life cycle evolution. The work also provides insights into evolution of bilaterian genome organization and gene family gain/loss along evolutionary timescales. The manuscript is well-written (but see some comments about organization below).

I am very excited about this manuscript and its results. In my opinion, the manuscript is thorough, well-written, and will have an important impact on understanding of animal evolution. I am very familiar with all of the bioinformatic methods employed (except WGCNA; I'm also not knowledgeable about in situ hybridization) and the methodologies used here are exemplary (but see my requests for clarification in the "Minor points" below).

RESPONSE: Many thanks for the positive appraisal.

Major points:

The second paragraph of the results section is a lengthy report of / discussion on results related to synteny. This is interesting but of little importance to the take-home messages highlighted in the abstract of the paper. I feel that the authors are burying the lead and that this text should be reduced and combined with the first paragraph on more general genome characterization or moved towards the end of the results.

ACTION: This is now changed as suggested (**lines 109-113:** "At a high-order genomic organisation, *O. fusiformis* has globally retained the ancestral bilaterian linkage, exhibiting chromosomal fusions that are present in molluscs and even nemertean (Fig. 1h; Extended Data Fig. 1h, i), and less lineage-specific chromosomal rearrangements than other annelids (Extended Data Fig. 1h).").

*Related to the discussion of novel genes – what proportion of the predicted genes/transcripts have direct evidence from 1) the *Owenia* transcriptomes and 2) proteomes of other annelids and /or public databases like the input proteomes used for the Metazoa odb10 used here in BUSCO?*

ACTION: We added those values for both *O. fusiformis* and *C. teleta* to Methods (**lines 746-756:** "Moreover, 31,678 out of the 31,903 (99.29%) of the filtered transcripts are supported by RNA-seq data and 80.69% of the transcripts have a significant BLAST match (*e*-value cut-off < 0.001) to a previously annotated annelid gene (database containing non-redundant proteomes of the high-quality annelid genomes of *C. teleta*, *D. gyrocoliatum*, *E. andreei*, *L. luymesii*, *P. echinospica*, *R. pachyptila* and *S. benedictii*). A similar functional annotation approach was followed to re-annotate the genome of *C. teleta* with the new RNA-seq data, using as starting assembly the soft masked version available at Ensembl Metazoa. This resulted in 41,221 transcripts, 39,814 of which have RNA-seq support (96.59%). Additionally, 80.47% of the transcripts have a significant BLAST match (*e*-value cut-off < 0.001) to other well-annotated annelid genomes (see above).").

The code availability statement indicates that no custom code was used in this study, but I would argue that making the series of commands used to conduct the analyses presented herein is code and providing the series of exact commands used as well as key input/output files would improve the reproducibility and clarity about exactly what was done in this paper.

ACTION: Following the referee's suggestion, and as replied in a previous point, we built a public repository with all code and datasets required to reproduce our findings at <https://github.com/ChemaMD/OweniaGenome>.

Minor points:

A non-trivial portion of the genome is not encompassed in the pseudomolecules/"chromosomes" – I assume that all of these scaffolds were included in downstream analyses?

ACTION: Yes, they were.

*The taxonomic authority should be listed after the first use of *Owenia fusiformis* (arguably other scientific names too but given the focus on this species, at least *Owenia* should have the authority mentioned).*

ACTION: Changed as suggested (**line 82:** we did not add it to the abstract given the word number limit in that section).

Line 90: 10X genomics "read clouds" is vague (and the "g" in genomics should be capitalized).

ACTION: Changed (**line 93:** "[...] 10x Genomics linked-reads, optical mapping [...]").

Line 163: I use "expressed" when referring to mRNA and "localized" when referring to protein. The meaning here is clear from context, but I thought I would note this.

ACTION: Changed as suggested (**line 108:** "[...] which is asymmetrically expressed around the blastopore lip of the gastrula [...]").

Line 198: Please clarify "mostly abundant within gene bodies" – within introns?

ACTION: Clarified (**lines 230-231:** "[...] mostly abundant within gene bodies (69.69% and 51.53%, respectively; from transcription start to end sites) [...]").

Line 243: I think "on" should be corrected to "in"

ACTION: We removed that part of the sentence.

Lines 395-396: The methods for filtering the repeat-masked genome to recover genes is not adequately described.

ACTION: We now describe this in more detail (**lines 709-713:** "Briefly, we used DIAMOND v.0.9.22⁷⁴ with an *e*-value cut-off of 1e-10 to identify sequences in the *de novo* repeat library with significant similarity to protein coding genes in *C. teleta* that are not transposable elements. Sequences with a significant hit were manually inspected to verify they were not transposable elements and if so, they were manually removed from the *de novo* repeat library.").

Figure 5C – Amphinomida is clade including Amphinomidae and a second family (Euphosinidae) – I suggest using this more inclusive name.

ACTION: Changed as suggested throughout text, figures and tables. We also modified Chaetopteridae to Chaetopteriformia.

Supplementary tables 8-10: I would have liked to see at least 3 replicates per transcriptome.

RESPONSE: Although our study generally relies on two replicates per time point, these show high correlation values (**Supplementary Fig. 2**), ensuring the statistical robustness of all transcriptomic analyses.

Reviewer Reports on the First Revision:

Referees' comments:

Referee #1 (Remarks to the Author):

I am satisfied by the authors' responses to my comments. I appreciate their willingness to reassess their statements and to strengthen their work with more analyses.

Referee #2 (Remarks to the Author):

The authors addressed my concerns (and very similar ones from Reviewer 1) by substantially transforming (or rather reducing) the article claims, which are now more generic/descriptive.

I must admit I am not very convinced by the reframing in terms of hypothesis testing, but at least the obvious limitations of the data in addressing some of the questions at hand (homology claims, larval evolutionary scenarios, etc.) is now made explicit.

The interpretation of the cross-species comparisons does not always correspond with the data shown in Extended Data Fig. 9a, e.g. *O. fusiformis* larval stages are not particularly more similar to *C. hemisphaerica* (and the trends are not even consistent with another cnidarian, *N. vectensis*).

The inclusion of additional annelid datasets is much welcome and, overall, this work will constitute an excellent resource for annelid genomics.

Referee #3 (Remarks to the Author):

The revised manuscript includes many additional comparative data and analyses, which further strengthen the original expression and regulation results (but places the genome analyses further from the central theme of the manuscript).

The authors also undertook an extensive revision of the manuscript taking into account the substantive comments and suggestions of the reviewers. Of particular importance is the considerable de-emphasis on the long-standing and conflicting theories about the role of larvae in bilaterian life cycle evolution. Three of us struggled to see how the results presented could resolve these theories. For the most part, reference to these have been removed or reduced, and there is now a stronger emphasis on the relationship between gene expression to the evolution of planktotrophy and lecithotrophy, and the role of heterochronic changes. I think this improves the focus and consistency of the manuscript, particularly in relation to the new comparative gene expression and regulation analyses.

However, I still feel that there is a disconnect between the Introduction, which still has an emphasis on "intercalation" vs "terminal addition" theories, and the results presented and the final Discussion, which barely explores this subject. It is unclear why these theories need to be in the manuscript at

all, except perhaps as a minor discussion point. Indeed, it is questionable whether the title to the current version accurately reflects the content.

With the revision a number of typos and odd expressions have crept in. The authors should undertake further editing and proofreading.

Referee #5 (Remarks to the Author):

I found the original version of this manuscript to be robust and potentially of broad interest. I am satisfied with the revisions made to the manuscript in light of my feedback. Other reviewers raised some non-trivial concerns and had significant suggestions for improvement. In my opinion these issues were also appropriately considered and the manuscript was revised and improved accordingly.

Author Rebuttals to First Revision:

Response to Reviewers

We wish to thank the four referees for their positive appraisals of our revised manuscript. We are pleased to provide a new version that we hope addresses the comments raised and complies with the editorial requirements.

Below we provide a detailed point-by-point response to each of the referees' concerns.

Referee #1 (Remarks to the Author):

I am satisfied by the authors' responses to my comments. I appreciate their willingness to reassess their statements and to strengthen their work with more analyses.

RESPONSE: Many thanks.

Referee #2 (Remarks to the Author):

The authors addressed my concerns (and very similar ones from Reviewer 1) by substantially transforming (or rather reducing) the article claims, which are now more generic/descriptive.

I must admit I am not very convinced by the reframing in terms of hypothesis testing, but at least the obvious limitations of the data in addressing some of the questions at hand (homology claims, larval evolutionary scenarios, etc.) is now made explicit.

RESPONSE: With shortening the manuscript, we have removed the sentences regarding hypothesis testing. We still mention the limitations of our datasets, as requested (line 265).

*The interpretation of the cross-species comparisons does not always correspond with the data shown in Extended Data Fig. 9a, e.g. *O. fusiformis* larval stages are not particularly more similar to *C. hemisphaerica* (and the trends are not even consistent with another cnidarian, *N. vectensis*).*

RESPONSE: As depicted in Extended Data 9b and summarised in Extended Data Fig. 9c, d, e, the larval phases of the two cnidarians are the stages of maximal transcriptomic similarity with the larva of *O. fusiformis*. However, the normalised values of transcriptomic similarity do vary between cnidarians, and thus the heatmaps are different. This is however likely a technical issue because (i) the time points of the cnidarian time-courses are not identical between species; and (ii) the *C. hemisphaerica* dataset also includes adult stages and specialised zooids.

The inclusion of additional annelid datasets is much welcome and, overall, this work will constitute an excellent resource for annelid genomics.

RESPONSE: Many thanks.

Referee #3 (Remarks to the Author):

The revised manuscript includes many additional comparative data and analyses, which further strengthen the original expression and regulation results (but places the genome analyses further from the central theme of the manuscript).

The authors also undertook an extensive revision of the manuscript taking into account the substantive comments and suggestions of the reviewers. Of particular importance is the considerable de-emphasis on the long-standing and conflicting theories about the role of larvae in bilaterian life cycle evolution. Three of us struggled to see how the results presented could resolve these theories. For the most part, reference to these have been

removed or reduced, and there is now a stronger emphasis on the relationship between gene expression to the evolution of planktotrophy and lecithotrophy, and the role of heterochronic changes. I think this improves the focus and consistency of the manuscript, particularly in relation to the new comparative gene expression and regulation analyses.

RESPONSE: Many thanks for the positive appraisal.

However, I still feel that there is a disconnect between the Introduction, which still has an emphasis on “intercalation” vs “terminal addition” theories, and the results presented and the final Discussion, which barely explores this subject. It is unclear why these theories need to be in the manuscript at all, except perhaps as a minor discussion point. Indeed, it is questionable whether the title to the current version accurately reflects the content.

RESPONSE: With shortening the text and as suggested by this referee, we have streamlined the Introduction, removed the description of the “intercalation” and “terminal addition” theories, and just briefly discussed these scenarios in the Discussion (lines 252–255). We have, however, retained the original title.

With the revision a number of typos and odd expressions have crept in. The authors should undertake further editing and proofreading.

RESPONSE: We hope these are now solved.

Referee #5 (Remarks to the Author):

I found the original version of this manuscript to be robust and potentially of broad interest. I am satisfied with the revisions made to the manuscript in light of my feedback. Other reviewers raised some non-trivial concerns and had significant suggestions for improvement. In my opinion these issues were also appropriately considered and the manuscript was revised and improved accordingly.

RESPONSE: Many thanks.